# Muon in Associative Memory Learning: Training Dynamics and Scaling Laws

**Kaifei Wang** [* 1]  **Binghui Li** [* 2]  **Han Zhong** [3 4]  **Pinyan Lu** [5]  **Liwei Wang** [2 4 6]

## Abstract

Muon updates matrix parameters via the matrix sign of the gradient and has shown strong empirical gains, yet its dynamics and scaling behavior remain unclear in theory. We study Muon in a linear associative memory model with softmax retrieval and a hierarchical frequency spectrum over query–answer pairs, with and without label noise. In this setting, we show that Gradient Descent (GD) learns frequency components at highly imbalanced rates, leading to slow convergence bottlenecked by low-frequency components. In contrast, the Muon optimizer mitigates this imbalance, leading to faster and more uniform progress. Specifically, in the noiseless case, Muon achieves an exponential speedup over GD; in the noisy case with a power-law frequency spectrum, we derive Muon's scaling law and demonstrate its superior scaling efficiency over GD. Furthermore, we show that Muon can be interpreted as an implicit matrix preconditioner arising from adaptive task alignment and block-symmetric gradient structure. In contrast, the preconditioner with coordinate-wise sign operator could match Muon under oracle access to unknown task representations, which is infeasible for SignGD in practice. Experiments on synthetic long-tail classification and LLaMA-style pre-training corroborate the theory.

## 1. Introduction

Modern Large Language Models (LLMs) (Brown et al., 2020; Touvron et al., 2023; Liu et al., 2024) are trained at massive scale, where optimization efficiency directly translates into compute and data efficiency. While adaptive gradient methods such as Adam (Kingma, 2014) and its decoupled weight decay variant AdamW (Loshchilov & Hutter, 2017) have long served as the standard, the Muon optimizer (Jordan et al., 2024) has recently emerged as a promising alternative, offering significant gains in the optimization of matrix-valued parameters. Crucially, empirical evidence (Jordan et al., 2024; Liu et al., 2025; Xie et al., 2025) demonstrates that Muon is particularly effective in the regime of large-scale training, achieving superior performance and computational efficiency compared to canonical first-order baselines such as SGD (Robbins & Monro, 1951), Adam, and AdamW. However, despite this empirical success, the theoretical underpinnings of Muon remain largely obscure. This creates a pronounced disparity between its widespread practical utility and our formal understanding of its convergence properties and optimization dynamics.

Most existing theoretical studies have predominantly concentrated on deriving convergence bounds under standard stochastic optimization frameworks (Li & Hong, 2025; Lau et al., 2025; Shen et al., 2025; Kovalev, 2025; Pethick et al., 2025; Chang et al., 2025). Yet, such static guarantees fail to capture the intricate optimization trajectory. To date, there is a scarcity of theoretical analysis focused on the training dynamics, which is crucial for demystifying how Muon's distinctive spectral normalization actively shapes the learning process. Furthermore, while neural scaling laws (Hestness et al., 2017; Kaplan et al., 2020; Hoffmann et al., 2022; Bi et al., 2024) act as the governing principles for modern large-scale training, the scaling behavior specific to Muon remains unexplored territory.

To bridge this gap, we provide a theoretical characterization of Muon's training dynamics through the lens of associative memory learning (Willshaw et al., 1969; Longuet-Higgins et al., 1970; Hopfield, 1982; Hopfield & Tank, 1985; Kohonen, 2009)—a tractable proxy that faithfully models the information retrieval and pattern matching capabilities of modern Transformer architectures (Geva et al., 2021; Dai et al., 2022; Meng et al., 2022a; Bietti et al., 2023; Ca-

---

[*]Equal contribution  [1]School of EECS, Peking University [2]Center for Machine Learning Research, Peking University [3]Antai College of Economics and Management, Shanghai Jiao Tong University [4]Center for Data Science, Peking University [5]Key Laboratory of Interdisciplinary Research of Computation and Economics, Shanghai University of Finance and Economics [6]State Key Laboratory of General Artificial Intelligence, Peking University. Correspondence to: Binghui Li <libinghui@pku.edu.cn>, Han Zhong <hanzhong.work@gmail.com>, Pinyan Lu <lu.pinyan@mail.shufe.edu.cn>, Liwei Wang <wanglw@pku.edu.cn>.

*Proceedings of the 43$^{rd}$ International Conference on Machine Learning*, Seoul, South Korea. PMLR 306, 2026. Copyright 2026 by the author(s).

bannes et al., 2023; 2024; Nichani et al., 2024; Fang et al., 2024; Smart et al., 2025; Wang et al., 2025b). Formally, we consider a knowledge set characterized by orthogonal embeddings and a hierarchical frequency structure. Under this setting, we analyze a linear softmax model for associative memory, trained to minimize the cross-entropy loss over knowledge items. Specifically, our main contributions are summarized as follows:

**1. Dynamics and acceleration.** In the noiseless associative memory model (Section 4), we show that GD learns different frequency components at highly imbalanced rates and is bottlenecked by low-frequency classes. In contrast, Muon equalizes progress across various frequency groups and achieves an exponential speedup. Under label noise (Section 5.1), we characterize Muon's three-phase training dynamics and show an $\Omega(C)$-fold speedup, where $C$ denotes the knowledge-group size.

**2. Scaling laws.** When group frequencies follow a power law (Section 5.2), $\widetilde{p}_i \propto i^{-\beta}$ for some constant $\beta > 1$, we derive Muon's optimization scaling law and show that its loss decays as $\tilde{\mathcal{O}}(1/T^2)$, whereas GD admits a lower bound of $\tilde{\Omega}(1/T^{1-1/\beta})$. This yields a substantially steeper scaling exponent for Muon.

**3. Mechanism and connection to SignGD.** In Section 6, we provide a preconditioning perspective in which Muon acts as an implicit matrix preconditioner: the matrix-sign update implicitly aligns with the underlying task representations and exploits a block-symmetric gradient structure. By contrast, coordinate-wise SignGD can match Muon under oracle access to the unknown task representations; SignGD in the original coordinates fails to exploit the latent structure.

**4. Experiments.** Experiments on synthetic imbalanced classification and LLaMA-style pre-training corroborate our theory, showing improved long-tail learning and stronger scaling efficiency consistent with our predictions.

Here we use standard Big-$\mathcal{O}/\tilde{\mathcal{O}}$ and Big-$\Omega/\tilde{\Omega}$ asymptotic notations; formal definitions are deferred to Section 3.

## 2. Related Work

**Associative memory.** The associative memory model traces its origins to the neural computation literature, where it was initially developed to characterize biological information storage mechanisms (Willshaw et al., 1969; Longuet-Higgins et al., 1970). It has since become a cornerstone of neural network design (Hopfield, 1982; Hopfield & Tank, 1985) and knowledge representation (Kohonen, 2009). Following the success of transformers, recent studies have begun reinterpreting them as associative memories (Geva et al., 2021; Dai et al., 2022; Meng et al., 2022a; Bietti et al., 2023; Cabannes et al., 2023; 2024; Nichani et al., 2024; Fang et al., 2024; Smart et al., 2025; Wang et al., 2025b).

Notably, Wang et al. (2025b) investigate Muon in heavy-tailed associative memory, demonstrating its effectiveness on tail classes. While their work highlights these capabilities empirically, we advance the theoretical understanding by establishing provable global convergence guarantees and deriving explicit scaling laws for Muon in noisy regimes.

**The Muon optimizer.** Jordan et al. (2024) recently proposed Muon, an optimizer that updates weights along the direction of the orthogonalized gradient, rather than the raw stochastic gradient. Empirically, Muon consistently surpasses Adam across various scales and architectures, including dense Transformers and Mixture-of-Experts (MoEs) (Jordan et al., 2024; Liu et al., 2025; Wen et al., 2025). Despite Muon's widespread adoption, the theoretical mechanisms underlying its effectiveness remain unexplored. To address this gap, we provide a theoretical analysis of Muon within the framework of associative memory.

**Theoretical analysis of Muon.** Bernstein & Newhouse (2024) characterize Muon as steepest descent with respect to the matrix operator norm. Subsequently, a growing body of literature (Li & Hong, 2025; Lau et al., 2025; Shen et al., 2025; Kovalev, 2025; Pethick et al., 2025; Chang et al., 2025) has focused on establishing convergence guarantees within the classical stochastic optimization framework. Vasudeva et al. (2025) study the generalization benefits of Muon in a Gaussian mixture setup. Finally, concurrent work by Ma et al. (2026) considers matrix factorization and in-context learning of linear transformers, highlighting the preconditioning benefits of spectral orthogonalization in Muon. We focus on the training dynamics of Muon in associative memory learning, establishing its global convergence guarantees and deriving its scaling laws.

**Theory of Scaling laws.** A series of studies have sought to theoretically explain the scaling behaviors of the training of large language models. In particular, a line of research (Bordelon et al., 2024; Paquette et al., 2024; Lin et al., 2024; Bordelon et al., 2025; Li et al., 2025; Wang et al., 2026; Li et al., 2026b) has analyzed scaling laws by tracking the one-pass SGD training dynamics using linear regression models. Most recently, Kunstner & Bach (2025) investigated the scaling laws of SignGD within a linear bigram model, while Kim et al. (2026) and Li et al. (2026a) characterized those of SignSGD. We extend this theoretical frontier to Muon, analyzing its scaling behavior under associative memory learning and proving its superior scaling efficiency relative to GD. To the best of our knowledge, this is the first theoretical analysis to characterize the scaling laws of Muon.

## 3. Preliminaries

In this section, we present notations used throughout the paper and introduce our theoretical framework.

**Notations.** Throughout this paper, letters denote scalars and bold letters denote vectors and matrices. We use $[N]$ to denote the set of $\{1, 2, \ldots, N\}$ for an integer $N$. The notation $\asymp$ or $\Theta$ indicates equivalence up to a constant factor, and $\lesssim$ or $\mathcal{O}$ (resp. $\gtrsim$ or $\Omega$) indicates inequality up to a constant factor. We use $\widetilde{\Theta}$, $\widetilde{\mathcal{O}}$ and $\widetilde{\Omega}$ to hide logarithmic factors. We also use $\gg$ or $\omega$ (resp. $\ll$ or $o$) to indicate that the left-hand side is significantly larger (resp. smaller) than the right-hand side, such that the smaller term is negligible in our analysis. We use $\omega_K(1)$ (resp. $o_K(1)$) to indicate the factor goes to $\omega(1)$ (resp. $o(1)$) as $K \to \infty$. In addition, we use $\|\cdot\|_2$ to denote the $\ell_2$ norm for vectors and the spectral norm for matrices, while $\|\cdot\|_F$ denotes the Frobenius norm for matrices. We use $\|\cdot\|_\infty$ to denote the $\ell_\infty$ norm for vectors, and use $\|\cdot\|_{\max}$ to denote the entrywise max norm for matrices, namely $\|\mathbf{X}\|_{\max} = \max_{i,j} |X_{i,j}|$. Finally, $\mathbf{I}_n \in \mathbb{R}^{n \times n}$ denotes the identity matrix, $\mathbf{1}_n \in \mathbb{R}^n$ denotes the all-ones vector, and $\mathbf{J}_n \in \mathbb{R}^{n \times n}$ denotes the all-ones matrix.

### 3.1. Associative memory learning

Associative memory learning provides a framework for modeling knowledge storage and retrieval through query-answer associations. We use it as an analytically tractable proxy for studying the optimization dynamics of knowledge learning in modern large-scale training.

**Knowledge and embeddings.** We consider a set of $K$ atomic knowledge items, where the $j$-th item consists of a query of subject-relation pair $\mathcal{Q}_j = (\mathcal{S}_j, \mathcal{R}_j)$ and a corresponding ground-truth answer $\mathcal{A}_j$. For example, regarding the factual knowledge "Paris is the capital of France", we have subject $\mathcal{S}_j$ = "Paris", relation $\mathcal{R}_j$ = "capital_of", and ground-truth answer $\mathcal{A}_j$ = "France". Let $\mathbf{E}_j \in \mathbb{R}^d$ and $\widetilde{\mathbf{E}}_j \in \mathbb{R}^d$ denote the embeddings of the subject-relation pair query $\mathcal{Q}_j$ and the answer $\mathcal{A}_j$, respectively, where $d$ denotes the embedding dimension.

**Assumption 3.1** (Orthogonal equinorm embeddings)**.** We assume: (1) $\|\mathbf{E}_j\|_2 = \|\widetilde{\mathbf{E}}_j\|_2 = 1$ for all $j \in [K]$; (2) $\mathbf{E}_i \perp \mathbf{E}_j$ and $\widetilde{\mathbf{E}}_i \perp \widetilde{\mathbf{E}}_j$ for all $1 \leq i < j \leq K$.

This strict orthogonality condition can be relaxed to a nearly-orthogonal case, which is widely adopted in the context of representation learning theory (Allen-Zhu & Li, 2020; Bietti et al., 2023; Li et al., 2024; Li & Li, 2024; 2025; Wang et al., 2025b; Vasudeva et al., 2025) and has been empirically observed in real-world knowledge learning practice (Geva et al., 2021; Dai et al., 2022; Meng et al., 2022a;b; Fang et al., 2024; Wang et al., 2025b). It requires that $d \geq K$ to ensure that there exist $K$ orthogonal vectors in the Euclidean space $\mathbb{R}^d$. For simplicity, we assume $d = K$ throughout this paper. Moreover, our techniques and results remain applicable in the absence of the equinormality condition.

**Knowledge structure.** We suppose that the knowledge follows a distribution $\mathcal{D}$, and let $p_j$ denote the frequency of the $j$-th knowledge item. We assume that the knowledge items can be partitioned into $M$ groups, each containing $C$ items (where we assume that $C = K/M$ is an integer), and all items within the same group share an identical frequency, which is formally presented as the following assumption.

**Assumption 3.2** (Hierarchical frequency)**.** There exist a decreasing positive constant sequence $\widetilde{p}_1 > \widetilde{p}_2 > \cdots > \widetilde{p}_M > 0$ such that $\sum_{i=1}^M \widetilde{p}_i = 1$ and $p_j = \frac{\widetilde{p}_i}{C}$ for all item index $j \in \{(i-1)C+1, (i-1)C+2, \ldots, iC\}$ and $i \in [M]$.

This assumption means that knowledge manifests as a hierarchical spectrum of frequencies, which is a premise consistently supported by empirical evidence in the study of large-scale language modeling tasks (Rosch et al., 1976; Ferrer i Cancho & Sole, 2001; Petersen et al., 2012; Michaud et al., 2023; Kandpal et al., 2023; An et al., 2025). Intuitively, this partitions factual associations $(\mathcal{Q}_j, \mathcal{A}_j)$ into discrete tiers of popularity: common facts like $((\text{Paris}, \text{capital\_of}), \text{France})$ occupy the high-frequency "head" groups, whereas specialized knowledge like $((\text{Thulium}, \text{boiling\_point}), 1950°\text{C})$ falls into the low-frequency "tail" groups. This discretization mirrors the frequency gaps between ubiquitous and rare information found in massive pre-training corpora. Technically, the GD analysis does not strictly rely on this assumption.

**Parametric regimes.** We consider the following regimes:

- **In the noiseless setting and the noisy setting except for the scaling law case** (Sections 4 and 5.1): We focus on the regime where the total number of knowledge items $K$ is sufficiently large and the number of groups $M$ is much smaller than the group size $C$; specifically, $1 \ll K$ and $M \ll C$ (equivalently, $M \ll \sqrt{K}$). In this regime, $K, M, C$ are considered fixed constants, while $t$ goes to infinity to obtain the asymptotic loss. The dependence on $K$ is kept explicit in the bounds to show task-number effects.

- **In the scaling law case** (Section 5.2): The relation $1 \ll K$ and $M \ll C$ are maintained, while $K, M, T$ are considered to go to infinity jointly. The specific constraints on the growth rates of $K, M, T$ are given in Section 5.2.

**Linear softmax model.** We use a linear softmax model to store knowledge. Specifically, we define a memory weight matrix $\mathbf{W} \in \mathbb{R}^{K \times K}$ that yields the following softmax probability:

$$\widehat{p}_{i|j}(\mathbf{W}) = \frac{\exp(\widetilde{\mathbf{E}}_i^\top \mathbf{W} \mathbf{E}_j)}{\sum_{k=1}^K \exp(\widetilde{\mathbf{E}}_k^\top \mathbf{W} \mathbf{E}_j)},$$

which denotes the predicted conditional probability of associating the embedding of answer $\mathcal{A}_i$ (i.e., $\widetilde{\mathbf{E}}_i$) with the

embedding of query $\mathcal{Q}_j$ (i.e., $\mathbf{E}_j$), effectively measuring the strength of the memory link between the two entities.

## 3.2. Training

**Label noise.** We consider a label-noise model where the noise level is characterized by $\alpha \in [0, 1)$. Here $\alpha = 0$ recovers the noiseless case and $\alpha \in (0, 1)$ corresponds to the noisy case. Specifically, the observed label matches the ground-truth index with probability $1 - \alpha$, and is sampled uniformly at random from all $K$ possible items with probability $\alpha$. It induces a conditional probability as follows:

$$p_{i|j} = \begin{cases} 1 - \alpha + \frac{\alpha}{K}, & \text{if } i = j \\ \frac{\alpha}{K}, & \text{if } i \neq j \end{cases}, \tag{1}$$

where $p_{i|j}$ represents the probability of assigning answer $\mathcal{A}_i$ to query $\mathcal{Q}_j$. It also results in a noisy knowledge distribution $\mathcal{D}_\alpha$ ($\mathcal{D}_0 = \mathcal{D}$) over query-answer space $\{\mathcal{Q}_j\}_{j=1}^K \times \{\mathcal{A}_i\}_{i=1}^K$.

**Cross-entropy loss.** Our goal is to learn the associative memory through the following cross-entropy (CE) loss:

$$\mathcal{L}(\mathbf{W}) = \mathbb{E}_{(\mathcal{Q}_j, \mathcal{A}_i) \sim \mathcal{D}_\alpha} \left[ -\log \widehat{p}_{i|j}(\mathbf{W}) \right],$$

which is the de facto standard loss function used in real-world classification tasks, particularly in LLM pre-training.

**Optimizers.** We consider the following two optimizers.

- **Gradient descent (GD):** the weight is updated by $\mathbf{W}_{t+1} = \mathbf{W}_t - \eta \nabla \mathcal{L}(\mathbf{W}_t)$.

- **Muon:** To focus the analysis on the preconditioning mechanism of Muon, we omit its momentum term and therefore simplify the update to $\mathbf{W}_{t+1} = \mathbf{W}_t - \eta \, \mathrm{msgn}\big(\nabla \mathcal{L}(\mathbf{W}_t)\big)$, where the matrix-sign operator $\mathrm{msgn}(\cdot)$ is defined as $\mathrm{msgn}(\mathbf{X}) = \mathbf{U} \, \mathrm{sgn}(\mathbf{\Sigma}) \mathbf{V}^\top$ based on the singular value decomposition (SVD) of $\mathbf{X} = \mathbf{U}\mathbf{\Sigma}\mathbf{V}^\top$.

This momentum-free variant coincides with Spectral GD (SpecGD) (Carlson et al., 2015a;b; Fan et al., 2025; Vasudeva et al., 2025) and therefore our theory directly applies to SpecGD and sheds light on matrix-sign preconditioners. Here, $\eta > 0$ denotes the constant learning rate, and both optimizers adopt the zero initialization, i.e., $\mathbf{W}_0 = \mathbf{0}_{K \times K}$.

## 3.3. Optimal loss and gradient structure

**Optimization Target.** Since softmax is invariant to adding the same constant to all logits for a fixed query, the loss depends only on logit differences, making the minimizer generally non-unique in weight space. The following two propositions characterize the optimization targets in the noiseless and noisy regimes, respectively.

**Proposition 3.3** (Noiseless case). *When $\alpha = 0$, the cross-entropy loss satisfies $\inf_{\mathbf{W}} \mathcal{L}(\mathbf{W}) = 0$. Moreover, for any sequence $\{\mathbf{W}^{(m)}\}$, $\mathbf{W}^{(m)} \in \mathbb{R}^{K \times K}$, $\mathcal{L}(\mathbf{W}^{(m)}) \to 0 \iff \widetilde{\mathbf{E}}_j^\top \mathbf{W}^{(m)} \mathbf{E}_j - \widetilde{\mathbf{E}}_i^\top \mathbf{W}^{(m)} \mathbf{E}_j \to +\infty, \, \forall j \in [K], \, \forall i \neq j$.*

**Proposition 3.4** (Noisy case). *When $0 < \alpha < 1$, the minimum value of $\mathcal{L}(\mathbf{W})$ is $\mathcal{L}^* := -(1 - \alpha + \frac{\alpha}{K}) \log(1 - \alpha + \frac{\alpha}{K}) - \frac{\alpha(K-1)}{K} \log \frac{\alpha}{K}$. Moreover, all global minimizers $\mathbf{W}^*$ induce the same predicted conditional distribution such that $\widehat{p}_{i|j}(\mathbf{W}^*) = p_{i|j}$, for all $1 \leq i, j \leq K$.*

The proofs of Propositions 3.3 and 3.4 are deferred to Appendix B.5. Viewed through the lens of logit gaps, label noise turns the problem from one whose infimum is approached only at infinite margins into one whose minimum is attained at finite margins.

**Gradient structure.** The following proposition, proved in Appendix B.1, characterizes the structure of the gradient.

**Proposition 3.5** (Gradient decomposition). *The gradient admits the decomposition:*

$$\nabla \mathcal{L}(\mathbf{W}) = \sum_{1 \leq i, j \leq K} \underbrace{p_j}_{query\ frequency} \underbrace{\big(\widehat{p}_{i|j}(\mathbf{W}) - p_{i|j}\big)}_{prediction\ residual} \underbrace{\widetilde{\mathbf{E}}_i \mathbf{E}_j^\top}_{association}.$$

This decomposition implies that the gradient is aligned with the embedding association $\widetilde{\mathbf{E}}_i \mathbf{E}_j^\top$ and weighted by the query frequency $p_j$. Crucially, the residual term $\widehat{p}_{i|j}(\mathbf{W}) - p_{i|j}$ acts as an automatic mechanism to correct prediction errors.

# 4. Warmup: Analysis of Noiseless Case

In this section, we analyze the noiseless case (i.e., $\alpha = 0$ in Equation (1)) as a warmup. In this setting, the loss infimum is approached only as the correct-vs-incorrect logit gaps diverge to infinity for each query (Proposition 3.3). As a consequence, using a larger learning rate typically accelerates the growth of these gaps and hence yields faster convergence. To decouple algorithmic effects from trivial learning rate scaling, we use a fixed constant-order learning rate $\eta = 1$ throughout the noiseless analysis.

We first give the convergence analysis for the GD dynamics.

**Theorem 4.1.** *Under GD, for sufficiently large training time $t \gtrsim 1$, the sub-task CE loss for the $j$-th knowledge item satisfies $\mathcal{L}_j^{GD}(t) \asymp 1/(p_j t)$. As a consequence, the total CE loss scales as $\mathcal{L}^{GD}(t) \asymp K/t$.*

We defer the full proof to Appendix C.1. Theorem 4.1 shows that the sub-task loss for the $j$-th knowledge item converges at a rate inversely proportional to its frequency $p_j$, implying that low-frequency items exhibit slower convergence and are learned less sufficiently within a finite training duration. The total loss converges at a rate of $\mathcal{O}(1/t)$.

In contrast to GD, we show that Muon achieves a linear convergence rate, as established in the following theorem.

**Theorem 4.2.** *Under Muon, for any training time $t > 0$, each sub-task CE loss exhibits identical convergence behavior, specifically, $\mathcal{L}_j^{Muon}(t) \approx Ke^{-(1+o_K(1))t}$, which also leads to a total CE loss $\mathcal{L}^{Muon}(t) \approx Ke^{-(1+o_K(1))t}$.*

The full proof is deferred to Appendix C.2. Theorem 4.2 demonstrates that Muon treats distinct frequency components uniformly, i.e., $\mathcal{L}_{j_1}^{\text{Muon}}(t)/\mathcal{L}_{j_2}^{\text{Muon}}(t) \to 1$ as $K \to \infty$ for any $1 \leq j_1 \neq j_2 \leq K$ and $t \geq 0$. Furthermore, Muon achieves exponential acceleration compared to the polynomial convergence of GD. Specifically, to attain a loss precision of $\mathcal{L}(t) \leq \epsilon$, the time complexity of GD is $\mathcal{O}(1/\epsilon)$, whereas Muon improves this complexity to $\mathcal{O}(\log(1/\epsilon))$.

We now give an intuitive explanation for this speedup. As established in Proposition 3.5, under GD the gradient is proportional to the prediction residual $\widehat{p}_{i|j}(\mathbf{W}_t) - p_{i|j}$. Hence $\|\nabla\mathcal{L}(\mathbf{W}_t)\|_F \to 0$ as $t \to \infty$, leading to a convergence slowdown. Moreover, by Proposition 3.5, the association term $\widetilde{\mathbf{E}}_j\mathbf{E}_j^\top$ is weighted by the knowledge frequency $p_j$, so the effective step size along the $j$-th component scales as $p_j$. Consequently, GD learns different frequency components at highly imbalanced speeds: $\mathcal{L}_j^{\text{GD}}(t) \approx 1/(p_j t)$, so low-frequency items converge much more slowly.

In contrast, Muon largely removes this frequency-induced imbalance by applying the matrix-sign operator to the gradient, which normalizes the update spectrum and yields near-isotropic progress across components (see Section 6.1 for details).

*Remark* 4.3 (Effect of normalization). Since normalization can increase the effective step size and thus affect convergence speed, we also consider normalized GD (NGD), whose update is $\mathbf{W}_{t+1} = \mathbf{W}_t - \eta\frac{\nabla\mathcal{L}(\mathbf{W}_t)}{\|\nabla\mathcal{L}(\mathbf{W}_t)\|_F}$. NGD is known to accelerate GD in linearly separable classification (Nacson et al., 2019; Deora et al., 2024). In our associative-memory setting, however, global normalization couples all columns in the task-representation space, yielding $M$ coupled nonlinear systems; the noisy case further introduces oscillations. These effects make a sharp theoretical analysis substantially more involved and beyond the scope of this work. We instead compare NGD empirically in both noiseless and noisy cases. While NGD is faster than GD, its learning process remains more imbalanced and slower than Muon. This suggests that the acceleration of Muon is not solely due to normalization. Details and results are provided in Appendix G.3.

# 5. Analysis of Noisy Case and Scaling Laws

Having established the theoretical analysis in the noiseless setting, we now move to a more realistic regime by studying the training dynamics and scaling behavior of Muon under label noise (i.e., $\alpha \in (0,1)$ in Equation (1)).

## 5.1. Training dynamics

**Theorem 5.1.** *Under Muon, the loss dynamics for the $j$-th knowledge learning sub-task satisfies:*

$$\mathcal{L}_j^{Muon}(t) \lesssim \begin{cases} Ke^{-\eta(1+o_K(1))t} + \eta t, & t \leq T_j^*; \\ \eta^2 + \mathcal{L}_j^*, & t > T_j^*, \end{cases}$$

*where $T_j^* = \Theta\left(\frac{\log K}{\eta}\right)$ denotes the corresponding critical time for the $j$-th knowledge item and the irreducible sub-task loss $\mathcal{L}_j^* = \mathcal{L}^*$ (defined in Proposition 3.4).*

We defer the full proof to Appendix D.2. Theorem 5.1 establishes that, under label noise, learning dynamics of Muon related to the $j$-th sub-task exhibit **two phases**: *(1) descent phase*, in which the loss decreases and decomposes into an exponential convergence term, analogous to the noiseless setting, and an additional noise accumulation term $\eta t$ induced by label noise injection; and *(2) oscillation phase*, during which the loss fluctuates at a scale of $\mathcal{L}_j^* + \mathcal{O}(\eta^2)$. The two-phase pattern also induces a trade-off in the choice of learning rate $\eta$: increasing the learning rate accelerates the decent phase but simultaneously leads to a larger oscillation magnitude in the second phase.

**Theorem 5.2.** *Under Muon, the total loss satisfies:*

$$\mathcal{L}^{Muon}(t) \lesssim \begin{cases} Ke^{-\eta(1+o_K(1))t} + \eta t, & t \lesssim \frac{\log K}{\eta}; \\ Ke^{-\eta(1+o_K(1))t} + \eta t + \eta^2 + \mathcal{L}^*, & t \approx \frac{\log K}{\eta}; \\ \eta^2 + \mathcal{L}^*, & t \gtrsim \frac{\log K}{\eta}. \end{cases}$$

This result is a direct corollary of applying Theorem 5.1 to all sub-tasks. All critical times share a same order since the decent phase task-space update is dominated by the $\mathbf{I}_K$ term, while $o_K(1)$ block matrices introduce lower-order differences and therefore lead to a short mixed phase. Indeed, the total loss exhibits **three phases**: (1) all sub-tasks are in the descent phase; (2) a mixed phase during which some sub-tasks have entered the oscillation while others are still in the descent phase. (3) all sub-tasks are in the oscillation phase. Substituting $\mathcal{L}^*$, we have the excess risk at the end of the phase-1 is $\mathcal{O}(M^2 \log K/K)$. The detailed computation is provided in Appendix D.2.2.

Next, we show the following lower bound for the GD loss dynamics. Before this, we introduce the concept of linear stability that is widely used to determine the appropriate range for the learning rate in GD dynamics (Wu et al., 2018; 2022; Damian et al., 2022).

**Definition 5.3** (Linear stability (Strogatz, 2001)). For a discrete dynamical system $\mathbf{x}_{t+1} = \mathbf{x}_t - \mathbf{F}(\mathbf{x}_t)$, where $x_t \in \mathbb{R}^n$ and $\mathbf{F} : \mathbb{R}^n \to \mathbb{R}^n$ is a differentiable mapping, let $\mathbf{x}^*$ be a fixed point of this stochastic dynamics. Consider the

linearized dynamical system: $\widetilde{\mathbf{x}}_{t+1} = \widetilde{\mathbf{x}}_t - \mathbf{J}^*(\widetilde{\mathbf{x}}_t - \mathbf{x}^*)$, where $\mathbf{J}^* = \nabla \mathbf{F}(\mathbf{x}^*) \in \mathbb{R}^{n \times n}$ denotes the Jacobian matrix at the fixed point. We say $\mathbf{x}^*$ is linearly stable if there exists a constant $C^* > 0$ such that $\|\widetilde{\mathbf{x}}_t - \mathbf{x}^*\|_2 \le C^* \|\widetilde{\mathbf{x}}_0 - \mathbf{x}^*\|_2$.

**Proposition 5.4** (Stability at minimizer). *The global minimizer $\mathbf{W}^*$ in Proposition 3.4 is linearly stable for GD with learning rate $\eta$, if and only if $\eta p_1 \lesssim 1$.*

The proof is provided in Appendix B.3.

**Theorem 5.5.** *Under the stability condition in Proposition 5.4, for sufficiently large training time $t \gg \frac{\log \log K}{\eta p_j}$, the loss dynamics for the $j$-th knowledge satisfies $\mathcal{L}_j^{GD}(t) - \mathcal{L}_j^* \gtrsim e^{-\eta p_j t}(\log K)^2$.*

See the full proof in Appendix D.1. Theorem 5.5 suggests that the convergence of GD is slower than exponential decay, with an exponent proportional to the knowledge frequency $p_j$. Moreover, according to the monotonicity of sub-task losses, i.e., $\mathcal{L}_1^{GD}(t) \le \mathcal{L}_2^{GD}(t) \le \cdots \le \mathcal{L}_K^{GD}(t)$, we know that the total loss satisfies: $\mathcal{L}^{GD}(t) = \sum_{j=1}^K p_j \mathcal{L}_j^{GD}(t) \ge \mathcal{L}_1^{GD}(t) \ge \mathcal{L}^* + \Omega(e^{-\eta p_1 t}(\log K)^2)$. To achieve an excess risk of $\mathcal{O}(M^2 \log K/K)$, GD requires $\Omega(C \log(K/M^2)/\eta)$ steps while Muon only needs $\mathcal{O}(\log K/\eta)$. This implies a speedup by a factor of $C$ (the group size), when treating the group number $M$ as a relatively fixed constant. See the detailed derivation in Appendix D.3.

## 5.2. Scaling laws

We now analyze the scaling behavior of Muon and GD in a large-scale training setup by letting $K$ and $M$ go to infinity jointly with $T$. Under this setting, we consider a proportional regime in which the total training time scales as $T \asymp M^\beta$ (Kunstner & Bach, 2025; Li et al., 2025). Specifically, we assume $cM^\beta \le T \le M^\beta$ for some $0 < c < 1$, and $(\log K)^{\frac{1}{\beta}} \le M \ll K^{\frac{1}{2}}$. We also assume that the probability mass of the $i$-th group $\widetilde{p}_i$ follows a power-law decay.

**Assumption 5.6** (Power-decay frequency spectrum). We assume that $\widetilde{p}_i \propto i^{-\beta}$ holds for some constant $\beta > 1$.

This assumption is a standard convention in scaling law theory (Bahri et al., 2024; Kunstner & Bach, 2025; Yan et al., 2025), where knowledge frequency is modeled as a power-law decay, a behavior reminiscent of Zipf's law in natural language distributions (Zipf, 2013; 2016).

**Theorem 5.7** (Scaling law for GD). *Under the stability condition in Proposition 5.4, $\mathcal{L}^{GD}(T) - \mathcal{L}^* \gtrsim \frac{\log K}{T^{1-1/\beta}}$.*

We defer the detailed proof to Appendix E.1. Similar to Theorem 5.5, we prove that $\mathcal{L}_j^{GD}(T) - \mathcal{L}_j^* \gtrsim e^{-\eta p_j T} \log K$ under the proportional regime $T \asymp M^\beta$, implying $\mathcal{L}^{GD}(T) - \mathcal{L}^* \gtrsim (\log K) \sum_{j=1}^K p_j e^{-\eta p_j T}$. Under the stability condition $\eta p_1 \lesssim 1$, the sum term $\sum_{j=1}^K p_j e^{-\eta p_j T} \gtrsim$

$\sum_{i=1}^M \widetilde{p}_i e^{-\widetilde{p}_i T} \approx \int_1^M z^{-\beta} e^{-z^{-\beta} T} \, \mathrm{d}z \approx T^{-(1-\frac{1}{\beta})}$. The key insight behind Theorem 5.7 is that while each sub-task converges exponentially, the aggregate excess risk exhibits a power-law decay $\widetilde{\Omega}(T^{-(1-1/\beta)})$, a result of task accumulation and heavy-tailed distribution of knowledge frequency.

**Theorem 5.8** (Scaling law for Muon). *Let $\eta = \Theta\left(\frac{\log K}{T}\right)$, then $\mathcal{L}^{Muon}(T) - \mathcal{L}^* \lesssim \left(\frac{\log K}{T}\right)^2$.*

The proof is provided in Appendix E.2. We set the learning rate to $\eta = \Theta\left(\frac{\log K}{T}\right)$ to optimally balance the trade-off between the descent rate in the first phase and the oscillation magnitude in the second phase. Theorem 5.8 demonstrates that Muon achieves a superior scaling efficiency of $\widetilde{\mathcal{O}}(T^{-2})$, which significantly outperforms GD's lower bound $\widetilde{\Omega}(T^{-(1-1/\beta)})$ by mitigating the effect of the frequency gap between distinct knowledge groups, which underscores the acceleration advantage of Muon in large-scale training.

## 6. Unveiling Muon via a Preconditioning View

In this section, we study Muon and SignGD (a canonical simplified variant of Adam) through a unified preconditioning lens, focusing on the difference between Muon's matrix sign operation and SignGD's coordinate-wise sign operator.

To facilitate our discussion, we work in the task representation space. Specifically, we define the weight matrix and gradient in the task-representation space as

$$\widehat{\mathbf{W}} = \widetilde{\mathbf{E}}^\top \mathbf{W} \mathbf{E}, \quad \mathbf{G}_t = \widetilde{\mathbf{E}}^\top \nabla \mathcal{L}(\mathbf{W}_t) \mathbf{E}. \quad (2)$$

### 6.1. Muon as a matrix-gradient preconditioner

With notations defined in (2), we can rewrite Muon as

$$\widehat{\mathbf{W}}_{t+1} = \widehat{\mathbf{W}}_t - \eta \, \mathrm{msgn}(\mathbf{G}_t).$$

By Proposition 3.5 and algebra, we have

$$\mathrm{msgn}(\mathbf{G}_t) = \mathrm{msgn}(\mathbf{P} - \widehat{\mathbf{P}}_t) \quad (3)$$

where $\mathbf{P}, \widehat{\mathbf{P}}_t \in \mathbb{R}^{K \times K}$ are defined as $(\mathbf{P})_{i,j} = p_j p_{i|j}$ and $(\widehat{\mathbf{P}}_t)_{i,j} = p_j \widehat{p}_{i|j}(\mathbf{W}_t)$, respectively. Furthermore, we control the term $\mathrm{msgn}(\mathbf{P} - \widehat{\mathbf{P}}_t)$ by the following proposition.

**Proposition 6.1** (Approximate identity in the descent phase). *Under the standing assumptions and zero initialization, for any Muon iterate in the global descent phase defined in Section 5.1, we have*

$$\| \mathrm{msgn}(\mathbf{P} - \widehat{\mathbf{P}}_t) - \mathbf{I}_K \|_{\max} = o_K(1).$$

*In the noiseless case, this conclusion holds for any finite time $t \ge 0$.*

Equation (3) and Proposition 6.1 demonstrate that the spectral normalized gradient approximates the identity matrix during the descent phase, i.e., $\mathrm{msgn}(\mathbf{G}_t) \approx \mathbf{I}_K$ and $\widehat{\mathbf{W}}_t \approx t\,\mathbf{I}_K$. Thus, Muon (i) preconditions the update to identify the desired task representations and roughly optimize in these directions, and (ii) makes a near-isotropic update across task-representation directions.

**Proof sketch of Proposition 6.1** The full proof is provided in Appendix C.2 and Appendix D.2 for the noiseless case and the noisy case, respectively. The key observation is that, starting from $\mathbf{W}_0 = \mathbf{0}$, Muon updates preserve the block symmetry induced by the frequency groups, which can be proved by induction. Hence, at every step, the prediction residual in the task-representation space admits the decomposition $\mathbf{P} - \widehat{\mathbf{P}}_t = \mathbf{R}_t^+ - \mathbf{R}_t^-$, where $\mathbf{R}_t^+$ is block diagonal with $M$ blocks and each diagonal block is proportional to $\mathbf{I}_C$, while $\mathbf{R}_t^-$ is a block-wise constant matrix containing $M^2$ blocks and each block is proportional to $\mathbf{J}_C$.

For each group, the solution space of $\mathbf{z}^\top \mathbf{1}_C = 0$ yields a $(C-1)$-dimensional within-group contrast singular subspace. Let $\mathcal{S}_i := \{\mathbf{x} \in \mathbb{R}^K \mid \mathbf{x}_{C(i-1)+1:Ci}^\top \mathbf{1}_C = 0 \text{ and } x_j = 0 \text{ for } j \notin [C(i-1)+1, Ci]\}$ for $i \in [M]$. On each $\mathcal{S}_i$, the block-wise constant term $\mathbf{R}_t^-$ vanishes, while $\mathbf{R}_t^+$ acts as a scalar multiple of the identity. Therefore, $\mathcal{S} := \mathcal{S}_1 \oplus \cdots \oplus \mathcal{S}_M$ is an $M(C-1)$-dimensional singular subspace of $\mathbf{P} - \widehat{\mathbf{P}}_t$. The remaining singular directions lie in the $M$-dimensional block-mean subspace $\mathcal{S}^\perp = \mathrm{span}\{\mathbf{e}_i \otimes \mathbf{1}_C\}_{i=1}^M$, where $\mathbf{e}_i$ is the $i$-th standard basis vector in $\mathbb{R}^M$. Thus, any unit singular vector in the $\mathcal{S}^\perp$ is constant within each group, so each of its entries has magnitude at most $\frac{1}{\sqrt{C}}$.

Consider a matrix $\mathbf{X} \in \mathbb{R}^{C \times (C-1)}$ where the columns form an orthonormal basis for the solution space of $\mathbf{x}^\top \mathbf{1}_C = 0$. We have $\mathbf{X}\mathbf{X}^\top = \mathbf{I}_C - \frac{1}{C}\mathbf{J}_C$. Since the contribution from the remaining $M$ singular directions is at most $M/C$, we obtain $\|\mathrm{msgn}(\mathbf{P} - \widehat{\mathbf{P}}_t) - \mathbf{I}_K\|_{\max} \leq \frac{1}{C} + \frac{M}{C} = o_K(1)$.

*Remark* 6.2. With the frequency hierarchy, only the $M$ block-mean singular directions evolve nontrivially during training, while the remaining $M(C-1)$ within-group contrast directions are fixed by symmetry. The matrix sign operator therefore plays two roles: it preconditions the aligned gradient against frequency disparities across different groups, and it dilutes the variation of this evolving part over the $C$ coordinates within each group, making the update close to $\mathbf{I}_K$.

### 6.2. Comparison between msgn and sgn operators

To better understand the matrix sign operator $\mathrm{msgn}$ in Muon, we compare it with the coordinate-wise sign operator $\mathrm{sgn}$ in SignGD, which is a simplified version of Adam.

**TRA-SignGD.** We introduce Task-Representation Aligned SignGD (TRA-SignGD), an idealized variant of SignGD whose updates depend on the *unknown* matrices $\mathbf{E}$ and $\widetilde{\mathbf{E}}$. Intuitively, TRA-SignGD first maps the gradient into the task-representation space, applies the coordinate-wise sign there, and then maps the signed update back to the original parameter space. TRA-SignGD is introduced purely as an analytical device to compare the preconditioning via sign mapping, rather than as a practical optimizer or a proposal for improving optimization. Concretely, the update rule is

$$\mathbf{W}_{t+1} = \mathbf{W}_t - \eta\,\widetilde{\mathbf{E}}\,\mathrm{sgn}\left(\widetilde{\mathbf{E}}^\top \nabla_{\mathbf{W}_t}\mathcal{L}(\mathbf{W}_t)\mathbf{E}\right)\mathbf{E}^\top.$$

Under the notations in (2), the update rule of TRA-SignGD can be rewritten as:

$$\widehat{\mathbf{W}}_{t+1} = \widehat{\mathbf{W}}_t - \eta\,\mathrm{sgn}(\mathbf{G}_t). \tag{4}$$

With TRA-SignGD, the optimization of different sub-tasks are independent. For each sub-task, zero initialization and symmetry preservation keep the predicted probabilities of all incorrect classes identical throughout training. Viewed through the logit gap, each sub-task reduces to a one-dimensional optimization problem with a strictly convex objective, so taking the coordinate-wise sign in the aligned basis always moves toward the optimum. Moreover, learning is balanced across sub-tasks because the update speed is the same along all aligned coordinates. We establish in the following theorem that SignGD, when equipped with task-representation alignment, can achieve performance comparable to Muon. The formal statement and proof are provided in Appendix F.

**Theorem 6.3** (Informal)**.** *The results of Theorems 5.1, 5.2, and 5.8, derived for Muon with a learning rate of $2\eta$, also hold for TRA-SignGD with a learning rate of $\eta$.*

However, strictly implementing TRA-SignGD is infeasible as task representations are typically unknown. When applying vanilla SignGD directly in the original parameter coordinates, it fails to exploit the latent structure, so its per-subtask behavior can no longer be guaranteed.

**SignGD in the original coordinates.** Viewed in the task-representation space, the update of vanilla SignGD is

$$\widehat{\mathbf{W}}_{t+1} = \widehat{\mathbf{W}}_t - \eta\widetilde{\mathbf{E}}^\top\,\mathrm{sgn}(\widetilde{\mathbf{E}}\mathbf{G}_t\mathbf{E}^\top)\mathbf{E}.$$

When the task-representation basis is misaligned with the original coordinate basis, the update of vanilla SignGD can be roughly viewed as applying the aligned gradient in an arbitrary coordinate system, performing the sign operation, and rotating it back. The high-frequency sub-tasks dominate the direction after rotation due to their large gradients, while the directional signals of the low-frequency sub-tasks are submerged, so their updates may fail to align with the correct

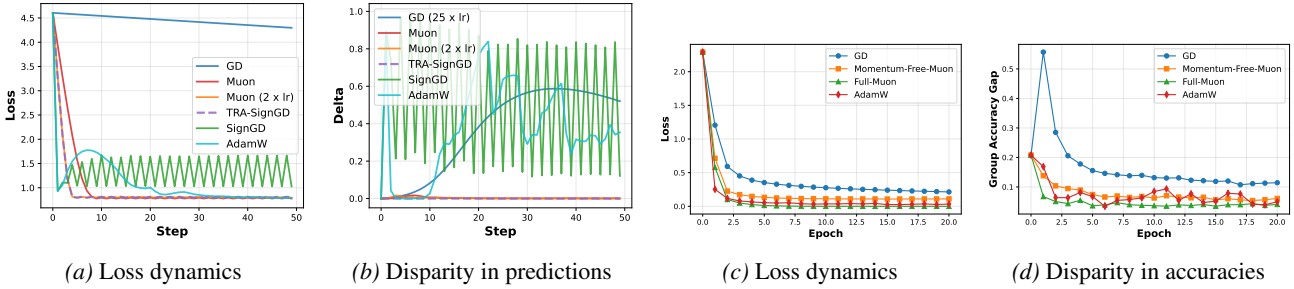

*(a)* Loss dynamics  *(b)* Disparity in predictions  *(c)* Loss dynamics  *(d)* Disparity in accuracies

*Figure 1.* (a)(b): Numerical simulations; (c)(d): Synthetic imbalanced classification.

direction, causing ineffective learning. When the number of low-frequency sub-tasks is large enough that their loss dominates the total loss, the optimization with SignGD will oscillate at a high loss. Moreover, the misalignment also makes the effective update norm in the task-representation space uncontrolled, which can be large and further leads to large-amplitude oscillations. Both phenomena are observed in our numerical simulations, as shown in Figure 1a and 1b.

This observation may offer a possible rationale for the empirical effectiveness of matrix-sign preconditioning reported in large-scale model training (Jordan et al., 2024; Liu et al., 2025; Wen et al., 2025). Prior work has discovered optimization heterogeneity across blocks in Transformers and shown that exploiting such heterogeneity can improve training efficiency (Zhang et al., 2024b;c;a; Wang et al., 2025a; Li et al., 2026a). However, focusing on each sub-block, even if there is some block-diagonal structure viewed in another basis (such as task representation space), SignGD/Adam fails to capture it since the basis is typically inaccessible. Muon, on the other hand, can adapt to such structures and may provide effective preconditioning.

## 7. Experiments

In this section, we validate our theory across increasingly realistic settings, gradually relaxing the assumptions made in our theoretical analysis. We start with controlled numerical simulations of the linear softmax associative memory model, then turn to the imbalanced MNIST classification (LeCun et al., 2002). Finally, we run LLaMA pre-training in multiple data budgets to measure optimization scaling laws and test whether Muon's data-efficiency advantage persists in realistic LLM training. Although our goal is not to directly compare Muon and AdamW, we also report AdamW results in the numerical simulations and synthetic imbalanced classification experiments to provide additional practical context.

### 7.1. Numerical simulations

**Settings.** We conduct numerical simulations of the linear softmax associative memory model. To verify the dynamics

predicted by our theory, we compare the performance of five optimizers: GD, Muon, TRA-SignGD, and SignGD, AdamW ($\beta_1 = 0.9$, $\beta_2 = 0.999$, weight decay = 0.01). We consider the following problem setting: $K = 100$, $M = C = 10$, the total steps $T = 50$, the base learning rate $\eta = 0.75$, and the noise level $\alpha = 0.1$. To quantify imbalance, we define the maximal probability gap as $\Delta_t = \max_{j \in [K]} \widehat{p}_{j|j}(\mathbf{W}_t) - \min_{j \in [K]} \widehat{p}_{j|j}(\mathbf{W}_t)$. Due to the slow convergence of GD, we additionally employ a larger learning rate of $25\eta$ when measuring the disparity, allowing for a meaningful visual comparison of the imbalanced learning within the same timeframe. See more details in Appendix G.1.

**Results.** The results are presented in Figures 1a and 1b. Evidently, Muon outperforms GD by effectively mitigating imbalanced learning. Notably, TRA-SignGD exhibits a loss trajectory nearly identical to that of Muon scaled by a factor of 2 in learning rate (Theorem 6.3), whereas vanilla SignGD fails to capture low-frequency knowledge, resulting in a higher probability gap and larger oscillations as analyzed in Section 6.2. As for AdamW, it exhibits pronounced oscillations in its loss trajectory and large nonsystematic fluctuations in the maximal probability gap, suggesting that it does not achieve balanced learning across groups.

### 7.2. Synthetic imbalanced classification

**Settings.** To empirically test whether our theoretical conclusions continue to hold beyond the idealized setting, we consider a nonlinear imbalanced MNIST classification task. We relax the strict orthogonal-embedding assumption (viewing the image inputs as the analogue of embeddings in our theoretical setting) and employ a two-layer MLP with ReLU activation to extend the analysis beyond the linear regime. We partition the training dataset into groups with varying retention rates to simulate an imbalanced distribution, while maintaining a balanced test dataset to ensure an unbiased evaluation. We compare the performance of SGD against the momentum-free Muon. We also report optimization dynamics for full-Muon and AdamW to provide additional practical context. All optimizers share a fixed learning rate of 0.005. Performance is evaluated by the training loss and

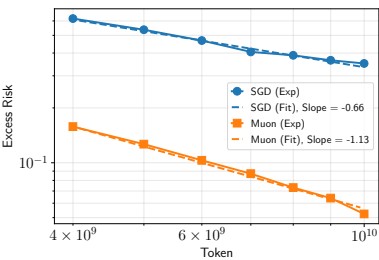

*Figure 2.* Data scaling behavior in language model pre-training. To extract the scaling envelope, we sweep over learning rates for each data budget and plot the optimal validation loss.

the group accuracy gap defined as the maximum disparity between group accuracies. See more details in Appendix G.2.

**Results.** As illustrated in Figures 1c and 1d, GD is bottlenecked by imbalanced learning arising from distinct group frequencies. In contrast, momentum-free Muon overcomes this spectral disparity, resulting in accelerated convergence. Compared to AdamW, full Muon achieves better accuracy on the low-frequency group and more balanced learning, while its performance on the high-frequency group is similar, resulting in a smaller group-accuracy gap and a lower training loss. These results empirically support that our theoretical insights generalize beyond the idealized associative memory learning and hold robustly in more realistic regimes.

### 7.3. Language model pre-training

**Settings.** To verify optimization scaling laws in realistic language modeling settings, we conduct extensive pre-training experiments using the LLaMA architecture (Touvron et al., 2023). We utilize a model configuration with approximately 100M parameters, trained on a 10B-token corpus with a constant learning rate, batch size 512, and sequence length 2048. By maintaining a fixed batch size for both SGD and Muon, the total number of training steps scales linearly with the data budget. For each budget, we sweep learning rates over $\{0.000125, 0.00025, 0.0005, 0.001, 0.002, 0.004, 0.008\}$ and report the optimal validation loss. We fit the resulting data scaling curves to the power-law form $\mathcal{L}(D) = \mathcal{L}^* + aD^{-\gamma}$, where $\mathcal{L}^*$ denotes the irreducible risk and $\mathcal{L}(D) - \mathcal{L}^*$ represents the excess risk.

**Results.** Figure 2 illustrates the data scaling laws for SGD and Muon. Aligned with our theoretical findings from linear associative memory learning (Theorem 5.7 and Theorem 5.8), we observe a distinct power-law decay in the optimal excess risk with respect to the data budget. Crucially, Muon exhibits a steeper scaling slope (faster scaling rate) compared to SGD. This confirms that the data efficiency advantage of Muon generalizes beyond the simplified linear softmax associative memory learning regime and holds robustly in LLM pre-training.

## 8. Conclusions and Limitations

In this work, we analyze the Muon optimizer within a linear associative memory framework and show that it mitigates the frequency-dependent optimization imbalance that bottlenecks GD under hierarchical data. Our analysis identifies implicit task alignment and matrix-gradient preconditioning as the key mechanisms: Muon adaptively aligns the update with task representations and promotes more uniform progress across frequency components. These mechanisms yield an exponential speedup in the noiseless case and improved scaling laws under label noise, and are supported empirically by synthetic experiments and LLM pre-training results. More broadly, our results shed light on a practical bottleneck in modern LLM training, where heavy-tailed language distributions and heterogeneous data sources can cause optimization to be dominated by frequent patterns while underrepresented components are learned much more slowly. This perspective suggests that Muon is particularly worth considering when training data are highly imbalanced or rare-frequency components matter, while also leaving several limitations and future directions discussed below.

**Orthogonal assumption and initialization.** Our theoretical analysis relies on orthogonal task embeddings, and the behavior under relaxed assumptions is only partially verified by experiments, leaving a theoretical gap. Our theory also relies on zero initialization. We provide additional numerical simulations under random initialization, examining loss dynamics, prediction disparity, and task-space update matrices to analyze alignment behavior. With small random initialization, Muon remains close to the zero-initialization case; with larger random initialization, it remains largely diagonal and balanced but becomes noisier. GD consistently shows clear frequency-dependent disparities. More details are provided in Appendix G.4.

**Gap between associative memory model and practical LLM training.** Our theory focuses on a linear associative memory model, a standard and widely adopted abstraction for rigorously analyzing neural network dynamics (Wang et al., 2025b; Kunstner & Bach, 2025; Kunstner et al., 2024; Zhong et al., 2025), but it still differs from the transformer setting. Future work can extend the analysis to more realistic deep-network and LLM training settings, studying how the alignment mechanism identified here arises and what structures Muon captures in practice.

**Limited scope of preconditioning analysis.** Our preconditioning analysis focuses on sign-style methods and removes the influence of momentum. Covariance-based preconditioners, such as Shampoo (Gupta et al., 2018) and SOAP (Vyas et al., 2025), are not covered. Extending the alignment perspective to these methods and to full practical Muon variants is an important future direction.

## Acknowledgment

Binghui Li is supported by the Elite Ph.D. Program in Applied Mathematics at Peking University. Liwei Wang is supported by National Science and Technology Major Project (2022ZD0114902) and National Science Foundation of China (NSFC92470123, NSFC62276005). This work is supported by the State Key Laboratory of General Artificial Intelligence.

## Impact Statement

This paper presents work whose goal is to advance the field of machine learning. There are many potential societal consequences of our work, none of which we feel must be specifically highlighted here.

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

## Appendix Contents

# A. Notations

The notations used in this appendix are consistent with those in the main paper. For clarity, we summarize all notations here:

## A.1. Setting and Model

We consider a linear associative memory model. $K$ is a positive integer, representing the number of pairs of queries and answers to be memorized. We assume $K$ is sufficiently large.

Each query and answer is represented as a $d$-dimensional vector. As we mentioned in the main paper, we set $d = K$ for simplicity. $\mathbf{E}_j \in \mathbb{R}^K$ is the embedding of the $j$-th query and $\widetilde{\mathbf{E}}_j \in \mathbb{R}^K$ is the embedding of the $j$-th answer. Denote $\mathbf{E} \in \mathbb{R}^{K \times K}$ and $\widetilde{\mathbf{E}} \in \mathbb{R}^{K \times K}$ as the embedding matrices of queries and answers, respectively.

$$\mathbf{E} = [\mathbf{E}_1, \mathbf{E}_2, \ldots, \mathbf{E}_K], \quad \widetilde{\mathbf{E}} = [\widetilde{\mathbf{E}}_1, \widetilde{\mathbf{E}}_2, \ldots, \widetilde{\mathbf{E}}_K].$$

The weight matrix of the linear associative memory model is denoted as $\mathbf{W} \in \mathbb{R}^{K \times K}$. $\mathbf{W}_t$ is the weight matrix at time step $t$ during training. $\widehat{\mathbf{W}}_t$ is defined as $\widehat{\mathbf{W}}_t = \widetilde{\mathbf{E}}^\top \mathbf{W}_t \mathbf{E}$, representing the weight matrix in the embedding space.

The predicted probability of the $i$-th answer given the $j$-th query at time step $t$ is denoted as $\widehat{p}_{t,i|j}$. It is defined as

$$\widehat{p}_{t,i|j} = \mathrm{softmax}(\widetilde{\mathbf{E}}^\top \mathbf{W}_t \mathbf{E}_j)_i = \frac{\exp(\widetilde{\mathbf{E}}_i^\top \mathbf{W}_t \mathbf{E}_j)}{\sum_{l=1}^K \exp(\widetilde{\mathbf{E}}_l^\top \mathbf{W}_t \mathbf{E}_j)}.$$

**Assumptions about the dataset:** We assume $K$ pairs can be divided into several groups, where pairs in the same group share the same probability in the dataset. $M$ is a positive integer, representing the number of groups. To simplify the analysis, we assume the size of each group is equal, which requires that $K$ is divisible by $M$. The size of each group is denoted as $C$, satisfying $C = K/M$. We assume $C$ is sufficiently large, which means $K$ is much larger than $M^2$.

The probability of the $j$-th query in the dataset is denoted as $p_j$. Without loss of generality, we assume the $i$-th group comprises pairs indexed from $(i-1)C + 1$ to $iC$, $i = 1, 2, \cdots, M$. $\widetilde{p}_i$ denotes the sum of probabilities of all pairs in the $i$-th group. Without loss of generality, we assume $\widetilde{p}_1 \geq \widetilde{p}_2 \geq \cdots \geq \widetilde{p}_M$.

The conditional probability that the $i$-th answer is observed given the $j$-th query in the dataset is denoted as $p_{i|j}$.

The noise level is denoted as $\alpha \in [0, 1)$, representing the probability that the observed label is incorrect. When the noise level is $\alpha$, the observed label is correct with probability $1 - \alpha$ and is uniformly randomly assigned to one of $K$ pairs with probability $\alpha$. In other words, the conditional probability $p_{i|j}$ can be expressed as:

$$p_{i|j} = \begin{cases} 1 - \alpha + \frac{\alpha}{K}, & \text{if } i = j \\ \frac{\alpha}{K}, & \text{if } i \neq j \end{cases}.$$

We refer to the case where $\alpha = 0$ as the noiseless case, while the case where $\alpha > 0$ is referred to as the noisy case.

**Scores:** To facilitate the analysis of optimization dynamics, we define $s^+$ and $s^-$ as the exponentiated logits (scores) for correct and incorrect class predictions, respectively.

Let $s^+_{t,i|j}$ denote the score at time $t$ for a correct prediction in the $i$-th group, given the $j$-th query as input. This score is well-defined only when the input query $j$ belongs to group $i$:

$$s^+_{t,i|j} = \exp(\widetilde{\mathbf{E}}_j^\top \mathbf{W}_t \mathbf{E}_j).$$

Similarly, we let $s^-_{t,i|j}$ represent the score for an incorrect prediction in group $i$ under input query $j$. We further demonstrate that all incorrect labels within the same group share a uniform prediction score. Thus, the score for an arbitrary incorrect entry in group $i$ is defined as:

$$s^-_{t,i|j} = \exp(\widetilde{\mathbf{E}}_l^\top \mathbf{W}_t \mathbf{E}_j),$$

where $l$ is any index such that $l$ belongs to group $i$ and $l \neq j$.

## A.2. Loss Function and Gradient

The cross-entropy loss at time step $t$ is denoted as $\mathcal{L}(\mathbf{W}_t)$. The gradient of the loss function with respect to $\mathbf{W}_t$ is denoted as $\nabla_{\mathbf{W}_t}\mathcal{L}(\mathbf{W}_t)$. Let $\mathbf{G}_t = \widetilde{\mathbf{E}}^\top \nabla_{\mathbf{W}_t}\mathcal{L}(\mathbf{W}_t)\mathbf{E}$ be the gradient in the task representation space.

As the memory task can be divided into $K$ sub-tasks based on the $K$ queries, the total loss can be viewed as the sum of the cross-entropy losses related to each sub-task. The cross-entropy loss related to the $i$-th sub-task is denoted as $\mathcal{L}_i(\mathbf{W}_t)$. The total loss can be expressed as:

$$\mathcal{L}(\mathbf{W}_t) = \sum_{i=1}^{K} p_i \mathcal{L}_i(\mathbf{W}_t).$$

The optimal value of the loss function is denoted as $\mathcal{L}^*$.

## A.3. Optimizer

Throughout this work, we assume zero initialization for the weight matrix that

$$\mathbf{W}_0 = \mathbf{0}. \tag{5}$$

For each optimizer discussed below, we employ a fixed learning rate $\eta > 0$. Note that while $\eta$ is constant throughout the training trajectory of a single model, its value may vary across different optimizers.

We consider Gradient Descent (GD), Adam, and Muon optimizers in our classification task. To facilitate our analysis, we employ simplified versions of Adam and Muon by omitting momentum. Furthermore, for Adam, we also set $\beta_2 = 0$ and omit the numerical stability epsilon, reducing the optimizer to Sign Gradient Descent (SignGD).

- **Gradient Descent (GD):** The weight matrix is updated directly using the gradient:

$$\mathbf{W}_{t+1} = \mathbf{W}_t - \eta \nabla_{\mathbf{W}_t}\mathcal{L}(\mathbf{W}_t).$$

- **SignGD:** The update depends only on the sign of the gradient:

$$\mathbf{W}_{t+1} = \mathbf{W}_t - \eta \operatorname{sgn}\left(\nabla_{\mathbf{W}_t}\mathcal{L}(\mathbf{W}_t)\right).$$

- **Muon:** The update is based on the matrix sign function of the gradient:

$$\mathbf{W}_{t+1} = \mathbf{W}_t - \eta \operatorname{msgn}\left(\nabla_{\mathbf{W}_t}\mathcal{L}(\mathbf{W}_t)\right).$$

Let the SVD of the gradient be $\nabla_{\mathbf{W}_t}\mathcal{L}(\mathbf{W}_t) = \mathbf{U}\mathbf{\Sigma}\mathbf{V}^\top$. The matrix sign function is defined as $\operatorname{msgn}\left(\nabla_{\mathbf{W}_t}\mathcal{L}(\mathbf{W}_t)\right) = \mathbf{U}\operatorname{sgn}(\mathbf{\Sigma})\mathbf{V}^\top$. The update rule of Muon can be expressed as:

$$\mathbf{W}_{t+1} = \mathbf{W}_t - \eta \, \mathbf{U}\operatorname{sgn}(\mathbf{\Sigma})\mathbf{V}^\top.$$

Using $\widehat{\mathbf{W}}_t$ and $\mathbf{G}_t$, the update rules of these optimizers can be rewritten as:

- **GD:**

$$\widehat{\mathbf{W}}_{t+1} = \widehat{\mathbf{W}}_t - \eta \mathbf{G}_t. \tag{6}$$

- **SignGD:**

$$\widehat{\mathbf{W}}_{t+1} = \widehat{\mathbf{W}}_t - \eta \widetilde{\mathbf{E}}^\top \operatorname{sgn}\left(\widetilde{\mathbf{E}}\mathbf{G}_t\mathbf{E}^\top\right)\mathbf{E}.$$

- **Muon:**

$$\widehat{\mathbf{W}}_{t+1} = \widehat{\mathbf{W}}_t - \eta \operatorname{msgn}(\mathbf{G}_t). \tag{7}$$

In the following analysis, we primarily focus on the updates of $\widehat{\mathbf{W}}_t$.

### A.4. Auxiliary Vectors, Matrices, and Functions

We define $\mathbf{1}_C \in \mathbb{R}^C$ as the $C$-dimensional vector of all ones. Let $\mathbf{I}_C \in \mathbb{R}^{C \times C}$ be the $C \times C$ identity matrix, and $\mathbf{J}_C \in \mathbb{R}^{C \times C}$ be the $C \times C$ all-ones matrix.

We denote the set of all discrete probability distributions over $K$ outcomes by $\Delta^{K-1}$. For a distribution $P \in \Delta^{K-1}$, $P = (p_1, p_2, \cdots, p_K)$, its entropy is defined as:

$$H(P) = -\sum_{i=1}^{K} p_i \log p_i. \tag{8}$$

For two distributions $Q_1, Q_2 \in \Delta^{K-1}$, $Q_1 = (q_{1,1}, q_{1,2}, \cdots, q_{1,K})$, $Q_2 = (q_{2,1}, q_{2,2}, \cdots, q_{2,K})$. The Kullback-Leibler (KL) divergence is defined as:

$$D_{\mathrm{KL}}(Q_1 \parallel Q_2) = \sum_{i=1}^{K} q_{1,i} \log(q_{1,i}/q_{2,i}). \tag{9}$$

## B. Supporting Propositions

The supporting propositions that will be used in the proof are presented in this section.

### B.1. Supporting Propositions of Gradient

We denote the probability of predicting the $i$-th answer given that the input is the $j$-th query at time step $t$ as:

$$\widehat{p}_{t,i|j} = \frac{\exp(\widetilde{\mathbf{E}}_i^\top \mathbf{W}_t \mathbf{E}_j)}{\sum_{l=1}^{K} \exp(\widetilde{\mathbf{E}}_l^\top \mathbf{W}_t \mathbf{E}_j)}.$$

The cross-entropy loss and its gradient are analyzed in both noiseless and noisy cases.

#### B.1.1. NOISELESS CASE

The cross-entropy loss in the noiseless case is:

$$\mathcal{L}(\mathbf{W}_t) = -\sum_{j=1}^{K} p_j \log \widehat{p}_{t,j|j}. \tag{10}$$

The gradient of the loss function with respect to $\mathbf{W}_t$ is:

$$\begin{aligned}
\nabla_{\mathbf{W}_t} \mathcal{L}(\mathbf{W}_t) &= -\sum_{j=1}^{K} p_j \left( (1 - \widehat{p}_{t,j|j}) \widetilde{\mathbf{E}}_j \mathbf{E}_j^\top - \sum_{i \neq j} \widehat{p}_{t,i|j} \widetilde{\mathbf{E}}_i \mathbf{E}_j^\top \right) \\
&= -\widetilde{\mathbf{E}} \cdot (\mathbf{P} - \widehat{\mathbf{P}}_t) \cdot \mathbf{E}^\top,
\end{aligned} \tag{11}$$

where $\mathbf{P} = \mathrm{diag}(p_1, p_2, \ldots, p_K)$ is the ground-truth joint probability matrix, and $\widehat{\mathbf{P}}_t \in \mathbb{R}^{K \times K}$ is the predicted joint probability matrix at time step $t$ with entries $(\widehat{\mathbf{P}}_t)_{i,j} = p_j \widehat{p}_{t,i|j}$.

#### B.1.2. NOISY CASE

We consider label noise with noise level $\alpha \in [0, 1)$, where the observed label is correct with probability $1 - \alpha$ and is uniformly randomly assigned to one of the $K$ possible values with probability $\alpha$.

Denoting the probability that the $j$-th query is matched with the $i$-th answer in the dataset as $p_{i|j}$, we have:

$$p_{i|j} = \begin{cases} 1 - \alpha + \frac{\alpha}{K}, & \text{if } i = j \\ \frac{\alpha}{K}, & \text{if } i \neq j \end{cases}.$$

The cross-entropy loss in the noisy case is:

$$\mathcal{L}(\mathbf{W}_t) = -\sum_{j=1}^{K} p_j \left( \sum_{i=1}^{K} p_{i|j} \log \widehat{p}_{t,i|j} \right). \tag{12}$$

The gradient of the loss function with respect to $\mathbf{W}_t$ is:

$$
\begin{aligned}
\nabla_{\mathbf{W}_t} \mathcal{L}(\mathbf{W}_t) &= -\sum_{j=1}^{K} p_j \sum_{i=1}^{K} \left( p_{i|j} (\widetilde{\mathbf{E}}_i \mathbf{E}_j^\top - \sum_{l=1}^{K} \widehat{p}_{t,l|j} \widetilde{\mathbf{E}}_l \mathbf{E}_j^\top) \right) \\
&= -\sum_{i,j} p_j (p_{i|j} - \hat{p}_{t,i|j}) \widetilde{\mathbf{E}}_i \mathbf{E}_j^\top \\
&= -\widetilde{\mathbf{E}} \cdot (\mathbf{P}' - \widehat{\mathbf{P}'}_t) \cdot \mathbf{E}^\top, \tag{13}
\end{aligned}
$$

where $\mathbf{P}' \in \mathbb{R}^{K \times K}$, $\widehat{\mathbf{P}'}_t \in \mathbb{R}^{K \times K}$ satisfy:

$$(\mathbf{P}')_{i,j} = p_j p_{i|j}, \quad (\widehat{\mathbf{P}'}_t)_{i,j} = p_j \widehat{p}_{t,i|j}.$$

## B.2. Supporting Propositions of SVD

**Proposition B.1.** *Let $\mathbf{Q} = \mathbf{A} + \mathbf{B} \in \mathbb{R}^{CM \times CM}$ be a matrix satisfying:*

$\mathbf{A} = \mathrm{diag}(a_1 \mathbf{1}_C^\top, a_2 \mathbf{1}_C^\top, \ldots, a_M \mathbf{1}_C^\top)$ *is a diagonal matrix, where the first $C$ elements on the diagonal are $a_1$, the second $C$ elements are $a_2$ and so on, up to $a_M$. $a_i > 0$ for all $i \in [M]$.*
$\mathbf{B}$ *is a block matrix with $M \times M$ blocks and each block is a $C \times C$ matrix, defined as:*

$$
\mathbf{B} = \begin{pmatrix}
b_{1,1} \mathbf{J}_C & b_{1,2} \mathbf{J}_C & \ldots & b_{1,M} \mathbf{J}_C \\
b_{2,1} \mathbf{J}_C & b_{2,2} \mathbf{J}_C & \ldots & b_{2,M} \mathbf{J}_C \\
\vdots & \vdots & \ddots & \vdots \\
b_{M,1} \mathbf{J}_C & b_{M,2} \mathbf{J}_C & \ldots & b_{M,M} \mathbf{J}_C
\end{pmatrix},
$$

*where $\mathbf{J}_C$ is a $C \times C$ matrix with all entries being 1.*

*Then, the matrix sign of the matrix $\mathbf{Q}$ satisfies:*

$$\mathrm{msgn}(\mathbf{Q}) = \mathbf{I}_{CM} + \mathbf{B}',$$

*where $\mathbf{I}_{CM}$ is the identity matrix in $\mathbb{R}^{CM \times CM}$ and $\mathbf{B}'$ shares the same block structure as $\mathbf{B}$ (a block matrix with $M \times M$ blocks and each block is a $C \times C$ matrix). Furthermore, each entry in $\mathbf{B}'$ is $\mathcal{O}(M/C)$, as $C/M \to \infty$.*

*Proof of Proposition B.1.* We prove Lemma B.2 first to analyze the SVD structure of $\mathbf{Q}$, then use it to complete the proof of Proposition B.1.

**Lemma B.2.** *Let $\mathbf{Q} = \mathbf{A} + \mathbf{B} \in \mathbb{R}^{CM \times CM}$ be a matrix defined as in Proposition B.1. The singular value decomposition (SVD) of $\mathbf{Q}$ can be characterized by the direct sum of $M + 1$ mutually orthogonal subspaces:*

$$\mathbb{R}^{CM} = \mathcal{S}^\perp \oplus \mathcal{S}_1 \oplus \mathcal{S}_2 \oplus \cdots \oplus \mathcal{S}_M.$$

*For each $i \in \{1, 2, \cdots, M\}$, $\mathcal{S}_i$ is a $(C - 1)$-dimensional subspace defined as:*

$$\mathcal{S}_i = \{\mathbf{x} \in \mathbb{R}^{CM} \mid \mathbf{x}_j = 0 \text{ for } j \notin [C(i-1)+1, Ci], \text{ and } \mathbf{x}_{C(i-1)+1:Ci}^\top \mathbf{1}_C = 0\},$$

*$\mathcal{S}_i$ is the singular subspace associated with the singular value $a_i$ of $\mathbf{Q}$. $\mathcal{S}^\perp$ is a $M$-dimensional space defined as:*

$$\mathcal{S}^\perp = \mathrm{span}\{\mathbf{y}_1, \ldots, \mathbf{y}_M\}, \text{ where } \mathbf{y}_i = \frac{1}{\sqrt{C}} (\mathbf{0}, \ldots, \mathbf{1}_C^\top, \ldots, \mathbf{0})^\top \text{ is the } i\text{-th block-unit vector.}$$

*Let $\mathbf{u}_1, \mathbf{u}_2, \cdots, \mathbf{u}_M$ be the left singular vectors of $\mathbf{Q}$ in $\mathcal{S}^\perp$ associated with singular values $\sigma_1 \geq \sigma_2 \geq \cdots \geq \sigma_M$ respectively, and $\mathbf{v}_1, \mathbf{v}_2, \cdots, \mathbf{v}_M$ be the corresponding right singular vectors. The entries of the singular vectors $\mathbf{u}_i$ and $\mathbf{v}_i$ satisfy*

$$|(\mathbf{u}_i)_j| \leq \frac{1}{\sqrt{C}}, \quad |(\mathbf{v}_i)_j| \leq \frac{1}{\sqrt{C}} \quad j = 1, 2, \cdots, CM, i = 1, 2, \cdots, M.$$

*Proof of Lemma B.2.* First, we show that $\mathcal{S}_i$ is a singular subspace of $\mathbf{Q}$ associated with singular value $a_i$ for each $i \in \{1, 2, \cdots, M\}$. As $\mathbf{B}$ is a block matrix with each block being a constant matrix, we have:

$$\mathbf{Bx} = \mathbf{B}^\top \mathbf{x} = \mathbf{0}, \quad \forall \mathbf{x} \in \mathcal{S}_i.$$

Thus, we have:

$$\mathbf{Qx} = \mathbf{Q}^\top \mathbf{x} = \mathbf{Ax} = a_i \mathbf{x}.$$

This indicates $\mathcal{S}_i$ is both the left and right singular subspace of $\mathbf{Q}$ associated with singular value $a_i$.

Next, we consider the singular vectors in $\mathcal{S}^\perp$.

The application of $\mathbf{Q}$ in $\mathcal{S}^\perp$ is determined by a $M \times M$ induced matrix $\tilde{\mathbf{Q}}$ defined as:

$$\tilde{\mathbf{Q}} = \begin{pmatrix} \mathbf{y}_1^\top \mathbf{Q}\mathbf{y}_1 & \mathbf{y}_1^\top \mathbf{Q}\mathbf{y}_2 & \cdots & \mathbf{y}_1^\top \mathbf{Q}\mathbf{y}_M \\ \mathbf{y}_2^\top \mathbf{Q}\mathbf{y}_1 & \mathbf{y}_2^\top \mathbf{Q}\mathbf{y}_2 & \cdots & \mathbf{y}_2^\top \mathbf{Q}\mathbf{y}_M \\ \vdots & \vdots & \ddots & \vdots \\ \mathbf{y}_M^\top \mathbf{Q}\mathbf{y}_1 & \mathbf{y}_M^\top \mathbf{Q}\mathbf{y}_2 & \cdots & \mathbf{y}_M^\top \mathbf{Q}\mathbf{y}_M \end{pmatrix}.$$

$$\tilde{\mathbf{Q}}_{i,j} = \begin{cases} a_i + Cb_{i,i} & \text{if } i = j \\ Cb_{i,j} & \text{if } i \neq j \end{cases}.$$

The singular values of $\tilde{\mathbf{Q}}$ are the same as the singular values of $\mathbf{Q}$ in $\mathcal{S}^\perp$. Let $\tilde{\mathbf{u}}_1, \tilde{\mathbf{u}}_2, \cdots, \tilde{\mathbf{u}}_M$ and $\tilde{\mathbf{v}}_1, \tilde{\mathbf{v}}_2, \cdots, \tilde{\mathbf{v}}_M$ be the left and right singular vectors, corresponding to singular values $\sigma_1 \geq \sigma_2 \geq \cdots \geq \sigma_M$ respectively. The following equations hold:

$$\mathbf{u}_i = \sum_{j=1}^{M} (\tilde{\mathbf{u}}_i)_j \mathbf{y}_j, \quad \mathbf{v}_i = \sum_{j=1}^{M} (\tilde{\mathbf{v}}_i)_j \mathbf{y}_j, \quad i = 1, 2, \cdots, M.$$

As $\|\tilde{\mathbf{u}}_i\|_2 = 1$ and $\|\tilde{\mathbf{v}}_i\|_2 = 1$, we have $|(\tilde{\mathbf{u}}_i)_j| \leq 1$ and $|(\tilde{\mathbf{v}}_i)_j| \leq 1$ for all $j = 1, 2, \cdots, M$. Thus, we have

$$|(\mathbf{u}_i)_j| \leq \frac{1}{\sqrt{C}}, \quad |(\mathbf{v}_i)_j| \leq \frac{1}{\sqrt{C}} \quad j = 1, 2, \cdots, CM, i = 1, 2, \cdots, M.$$

$\square$

Assume the SVD of $\mathbf{Q}$ is $\mathbf{Q} = \mathbf{U}\boldsymbol{\Sigma}\mathbf{V}^\top$. We define a new matrix $\mathbf{X} \in \mathbb{R}^{C \times (C-1)}$, the columns of which form an orthonormal basis of the subspace $\{\mathbf{x} \in \mathbb{R}^C \mid \mathbf{x}^\top \mathbf{1}_C = 0\}$. With Lemma B.2, the matrix $\mathbf{U}$ and $\mathbf{V}$ can be expressed as:

$$\mathbf{U} = \begin{pmatrix} \mathbf{X} & \mathbf{0} & \cdots & \mathbf{0} & & & & \\ \mathbf{0} & \mathbf{X} & \cdots & \vdots & \mathbf{u}_1 & \mathbf{u}_2 & \cdots & \mathbf{u}_M \\ \vdots & \vdots & \ddots & \vdots & & & & \\ \mathbf{0} & \mathbf{0} & \cdots & \mathbf{X} & & & & \end{pmatrix},$$

$$\mathbf{V} = \begin{pmatrix} \mathbf{X} & \mathbf{0} & \cdots & \mathbf{0} & & & & \\ \mathbf{0} & \mathbf{X} & \cdots & \vdots & \mathbf{v}_1 & \mathbf{v}_2 & \cdots & \mathbf{v}_M \\ \vdots & \vdots & \ddots & \vdots & & & & \\ \mathbf{0} & \mathbf{0} & \cdots & \mathbf{X} & & & & \end{pmatrix}.$$

Then, the matrix sign of $\mathbf{Q}$ can be expressed as:

$$\mathrm{msgn}(\mathbf{Q}) = \mathbf{U}\,\mathrm{sgn}(\boldsymbol{\Sigma})\mathbf{V}^\top = \begin{pmatrix} \mathbf{XX}^\top & \mathbf{0} & \cdots & \mathbf{0} \\ \mathbf{0} & \mathbf{XX}^\top & \cdots & \vdots \\ \vdots & \vdots & \ddots & \vdots \\ \mathbf{0} & \mathbf{0} & \cdots & \mathbf{XX}^\top \end{pmatrix} + \sum_{i=1}^{M} \mathrm{sgn}(\sigma_i)\mathbf{u}_i\mathbf{v}_i^\top.$$

According to Lemma B.2, the entries of the singular vectors $\mathbf{u}_i$ and $\mathbf{v}_i$ satisfy $|(\mathbf{u}_i)_j| \leq \frac{1}{\sqrt{C}}$ and $|(\mathbf{v}_i)_j| \leq \frac{1}{\sqrt{C}}$ for all $j = 1, 2, \cdots, CM$, $i = 1, 2, \cdots, M$. Thus, the absolute value of each entry in the matrix $\sum_{i=1}^{M} \mathrm{sgn}(\sigma_i)\mathbf{u}_i\mathbf{v}_i^\top$ is upper bounded by $\frac{M}{C}$. Then we analyze the term $\mathbf{XX}^\top$.

**Lemma B.3.** *We define $\mathbf{X}$ as above. The following equation holds: $\mathbf{XX}^\top = \mathbf{I}_C - \frac{1}{C}\mathbf{J}_C$.*

*Proof of Lemma B.3.* We construct a $C \times C$ matrix $\mathbf{X}'$, defined as $\mathbf{X}' = \left(\mathbf{X}, \quad \frac{1}{\sqrt{C}}\mathbf{1}_C\right)$. $\mathbf{X}'$ is an orthogonal matrix, thus we have:

$$\mathbf{I}_C = \mathbf{X}'\mathbf{X}'^\top = \mathbf{XX}^\top + \frac{1}{C}\mathbf{J}_C.$$

Thus we have $\mathbf{XX}^\top = \mathbf{I}_C - \frac{1}{C}\mathbf{J}_C$. $\qquad\square$

With Lemma B.3, we can express the matrix sign of $\mathbf{Q}$ as:

$$\mathrm{msgn}(\mathbf{Q}) = \mathbf{I}_{CM} - \begin{pmatrix} \frac{1}{C}\mathbf{J}_C & \mathbf{0} & \cdots & \mathbf{0} \\ \mathbf{0} & \frac{1}{C}\mathbf{J}_C & \cdots & \vdots \\ \vdots & \vdots & \ddots & \vdots \\ \mathbf{0} & \mathbf{0} & \cdots & \frac{1}{C}\mathbf{J}_C \end{pmatrix} + \sum_{i=1}^{M} \mathrm{sgn}(\sigma_i)\mathbf{u}_i\mathbf{v}_i^\top.$$

Thus, we complete the proof of Proposition B.1. $\qquad\square$

**Proposition B.4.** *Let $\mathbf{Q} = \mathbf{A} + \mathbf{B} \in \mathbb{R}^{CM \times CM}$ be defined as in Proposition B.1, but removing the restriction $a_i > 0$. Then, the matrix sign of $\mathbf{Q}$ satisfies:*

$$\mathrm{msgn}(\mathbf{Q}) = \begin{pmatrix} \mathrm{sgn}(a_1)I_C & \mathbf{0} & \cdots & \mathbf{0} \\ \mathbf{0} & \mathrm{sgn}(a_2)I_C & \cdots & \vdots \\ \vdots & \vdots & \ddots & \vdots \\ \mathbf{0} & \mathbf{0} & \cdots & \mathrm{sgn}(a_M)I_C \end{pmatrix} + \mathbf{B}',$$

*where $\mathbf{I}_C$ is the identity matrix in $\mathbb{R}^{C \times C}$ and $\mathbf{B}'$ maintains the same block structure as $\mathbf{B}$ with entries of order $\mathcal{O}(M/C)$.*

*Proof of Proposition B.4.* Without the restriction $a_i > 0$, Lemma B.2 still holds. The only difference is that for each $i \in \{1, 2, \cdots, M\}$, $\mathcal{S}_i$ is the singular subspace associated with the singular value $|a_i|$ of $\mathbf{Q}$. We use the same notations as in the proof of Proposition B.1. Thus, the matrix $U$ and $V$ can be expressed as:

$$\mathbf{U} = \begin{pmatrix} \mathrm{sgn}(a_1)\mathbf{X} & \mathbf{0} & \cdots & \mathbf{0} & & & & \\ \mathbf{0} & \mathrm{sgn}(a_2)\mathbf{X} & \cdots & \vdots & \mathbf{u}_1 & \mathbf{u}_2 & \cdots & \mathbf{u}_M \\ \vdots & \vdots & \ddots & \vdots & & & & \\ \mathbf{0} & \mathbf{0} & \cdots & \mathrm{sgn}(a_M)\mathbf{X} & & & & \end{pmatrix},$$

$$\mathbf{V} = \begin{pmatrix} \mathbf{X} & \mathbf{0} & \cdots & \mathbf{0} & & & & \\ \mathbf{0} & \mathbf{X} & \cdots & \vdots & \mathbf{v}_1 & \mathbf{v}_2 & \cdots & \mathbf{v}_M \\ \vdots & \vdots & \ddots & \vdots & & & & \\ \mathbf{0} & \mathbf{0} & \cdots & \mathbf{X} & & & & \end{pmatrix}.$$

Then, the matrix sign of $\mathbf{Q}$ can be expressed as:

$$\text{msgn}(\mathbf{Q}) = \begin{pmatrix} \text{sgn}(a_1)(\mathbf{I}_C - \frac{1}{C}\mathbf{J}_C) & \mathbf{0} & \cdots & \mathbf{0} \\ \mathbf{0} & \text{sgn}(a_2)(\mathbf{I}_C - \frac{1}{C}\mathbf{J}_C) & \cdots & \vdots \\ \vdots & \vdots & \ddots & \vdots \\ \mathbf{0} & \mathbf{0} & \cdots & \text{sgn}(a_M)(\mathbf{I}_C - \frac{1}{C}\mathbf{J}_C) \end{pmatrix} + \sum_{i=1}^{M} \text{sgn}(\sigma_i)\mathbf{u}_i\mathbf{v}_i^{\top}.$$

According to Lemma B.2, the absolute value of each entry in the matrix $\sum_{i=1}^{M} \text{sgn}(\sigma_i)\mathbf{u}_i\mathbf{v}_i^{\top}$ is upper bounded by $\frac{M}{C}$. Thus, we complete the proof of Proposition B.4. $\qquad\square$

### B.3. Supporting Proposition for Linear Stability

*Proof of Proposition 5.4.* We work in the task-representation space

$$\widehat{\mathbf{W}}_t = \widetilde{\mathbf{E}}^{\top}\mathbf{W}_t\mathbf{E}.$$

Under GD, the columns of $\widehat{\mathbf{W}}_t$ evolve independently. Thus it suffices to first study the first column. Its update is

$$(\widehat{\mathbf{W}}_{t+1})_{:,1} = (\widehat{\mathbf{W}}_t)_{:,1} - \eta p_1 \left( \text{softmax}((\widehat{\mathbf{W}}_t)_{:,1}) - p_{\cdot|1} \right),$$

where $p_{\cdot|1} = \left(1 - \alpha + \frac{\alpha}{K}, \frac{\alpha}{K}, \ldots, \frac{\alpha}{K}\right)^{\top}$. Let $(\widehat{\mathbf{W}}^*)_{:,1}$ be a fixed point satisfying $\text{softmax}((\widehat{\mathbf{W}}^*)_{:,1}) = p_{\cdot|1}$. Such fixed points are not unique because the softmax function is invariant under adding a constant multiple of $\mathbf{1}_K$ to the logits. However, the Jacobian of the gradient part at any such fixed point is the same:

$$\mathbf{J}_1^* = \eta p_1 \left( \text{diag}(p_{\cdot|1}) - p_{\cdot|1}p_{\cdot|1}^{\top} \right).$$

Therefore the linearized GD map around this fixed point is $\mathbf{I} - \mathbf{J}_1^*$. The matrix $\text{diag}(p_{\cdot|1}) - p_{\cdot|1}p_{\cdot|1}^{\top}$ has eigenvalues $0, \alpha\left(1 - \alpha + \frac{\alpha}{K}\right), \frac{\alpha}{K}$ (with multiplicity $K - 2$). Consequently, the eigenvalues of the linearized GD map $\mathbf{I} - \mathbf{J}_1^*$ are $1, 1 - \eta p_1 \alpha\left(1 - \alpha + \frac{\alpha}{K}\right), 1 - \eta p_1 \frac{\alpha}{K}$ (with multiplicity $K - 2$).

We use the standard power-bounded notion of linear stability: the fixed point is linearly stable if there exists a constant $C > 0$ such that

$$\|(\mathbf{I} - \mathbf{J}_1^*)^t\|_2 \le C, \qquad \forall t \ge 0.$$

Since $\mathbf{J}_1^*$ is symmetric, this condition is equivalent to requiring every eigenvalue of $\mathbf{I} - \mathbf{J}_1^*$ to have absolute value at most one. Thus, linear stability is equivalent to

$$\left|1 - \eta p_1 \alpha\left(1 - \alpha + \frac{\alpha}{K}\right)\right| \le 1 \quad \text{and} \quad \left|1 - \eta p_1 \frac{\alpha}{K}\right| \le 1.$$

When $\alpha \in (0, 1)$ is treated as a fixed constant, this is equivalent to $\eta p_1 \lesssim 1$.

For a general column $j$, the same argument applies after replacing $p_1$ by $p_j$ and permuting the coordinates of $p_{\cdot|1}$. Thus the $j$-th column is linearly stable if and only if $\eta p_j \lesssim 1$ Since $p_1 \ge p_2 \ge \cdots \ge p_K$, all columns are linearly stable if and only if the largest-frequency column is linearly stable. Therefore the global minimizer is linearly stable for GD if and only if $\eta p_1 \lesssim 1$, which proves both the "if" and the "only if" directions. $\qquad\square$

### B.4. Proof for the Noiseless Case (Proposition 3.3)

*Proof of Proposition 3.3.* When $\alpha = 0$, the problem reduces to a classification task with no label noise. The cross-entropy loss function $\mathcal{L}(\mathbf{W})$ is given by

$$\mathcal{L}(\mathbf{W}) = -\sum_{j=1}^{K} p_j \log\left(\frac{\exp(\widetilde{\mathbf{E}}_j^{\top}\mathbf{W}\mathbf{E}_j)}{\sum_{l=1}^{K} \exp(\widetilde{\mathbf{E}}_l^{\top}\mathbf{W}\mathbf{E}_j)}\right) = \sum_{j=1}^{K} p_j \log\left(1 + \sum_{i \ne j} \exp\left(\widetilde{\mathbf{E}}_i^{\top}\mathbf{W}\mathbf{E}_j - \widetilde{\mathbf{E}}_j^{\top}\mathbf{W}\mathbf{E}_j\right)\right).$$

Since each term is nonnegative, we have $\mathcal{L}(\mathbf{W}) \geq 0$. Moreover, there exists a sequence of weights whose correct-vs-incorrect logit gaps diverge to $+\infty$ for all $j \in [K]$ and $i \neq j$, along which the loss converges to zero. Hence, $\inf_{\mathbf{W}} \mathcal{L}(\mathbf{W}) = 0$.

We now prove the equivalence. First, suppose that for a sequence $\{\mathbf{W}^{(m)}\}$, $\widetilde{\mathbf{E}}_j^\top \mathbf{W}^{(m)} \mathbf{E}_j - \widetilde{\mathbf{E}}_i^\top \mathbf{W}^{(m)} \mathbf{E}_j \to +\infty$, for all $j \in [K], i \neq j$. Then $\exp\left(\widetilde{\mathbf{E}}_i^\top \mathbf{W}^{(m)} \mathbf{E}_j - \widetilde{\mathbf{E}}_j^\top \mathbf{W}^{(m)} \mathbf{E}_j\right) \to 0$, for all $j \in [K], \forall i \neq j$. Therefore, each logarithmic term in $\mathcal{L}(\mathbf{W}^{(m)})$ converges to zero, and thus $\mathcal{L}(\mathbf{W}^{(m)}) \to 0$.

Conversely, suppose $\mathcal{L}(\mathbf{W}^{(m)}) \to 0$. Since $p_j > 0$ and each logarithmic term is nonnegative, for every $j \in [K]$ we must have

$$\log\left(1 + \sum_{i \neq j} \exp\left(\widetilde{\mathbf{E}}_i^\top \mathbf{W}^{(m)} \mathbf{E}_j - \widetilde{\mathbf{E}}_j^\top \mathbf{W}^{(m)} \mathbf{E}_j\right)\right) \to 0.$$

This implies $\sum_{i \neq j} \exp\left(\widetilde{\mathbf{E}}_i^\top \mathbf{W}^{(m)} \mathbf{E}_j - \widetilde{\mathbf{E}}_j^\top \mathbf{W}^{(m)} \mathbf{E}_j\right) \to 0$. Since every term in the sum is nonnegative, each term must converge to zero. Hence, for all $j \in [K]$ and $i \neq j$,

$$\widetilde{\mathbf{E}}_j^\top \mathbf{W}^{(m)} \mathbf{E}_j - \widetilde{\mathbf{E}}_i^\top \mathbf{W}^{(m)} \mathbf{E}_j \to +\infty.$$

This completes the proof. $\qquad\square$

### B.5. Proof for the Noisy Case (Proposition 3.4)

*Proof of Proposition 3.4.* Denote $P_{\cdot|j}$ as a probability distribution in the dataset given the $j$-th query. Denote $\widehat{P}_{\cdot|j}$ as the predicted probability distribution given the $j$-th query.

According to Equation (12), the loss function can be represented with KL divergence between $P_{\cdot|j}$ and $\widehat{P}_{\cdot|j}$ and entropy of $P_{\cdot|j}$:

$$\mathcal{L}(\mathbf{W}) = -\sum_{j=1}^{K} p_j \left(\sum_{i=1}^{K} p_{i|j} \log \widehat{p}_{t,i|j}\right) = \sum_{j=1}^{K} p_j \left(D_{\mathrm{KL}}(P_{\cdot|j} \parallel \widehat{P}_{\cdot|j}) + H(P_{\cdot|j})\right),$$

where $D_{\mathrm{KL}}(\cdot)$ is defined in Equation (9) and $H(\cdot)$ is the entropy defined in Equation (8). Since KL divergence is nonnegative, we have $\mathcal{L}(\mathbf{W}) \geq \sum_{j=1}^{K} p_j H(P_{\cdot|j})$. The equality holds when $\widehat{P}_{\cdot|j} = P_{\cdot|j}$ for all $j \in [K]$, i.e., $\widehat{p}_{t,i|j} = p_{i|j}$ for all $i, j \in [K]$.

According to the assumption of label noise, $P_{\cdot|j}$ for all $j \in [K]$ is equal to the same distribution. Therefore, we have:

$$H(P_{\cdot|j}) = -\left(1 - \alpha + \frac{\alpha}{K}\right) \log\left(1 - \alpha + \frac{\alpha}{K}\right) - \frac{\alpha(K-1)}{K} \log\left(\frac{\alpha}{K}\right), \quad \text{for all } j \in [K].$$

Then, the minimum value of the loss function, $\mathcal{L}^*$, is given by:

$$\mathcal{L}^* = \sum_{j=1}^{K} p_j H(P_{\cdot|j}) = -\left(1 - \alpha + \frac{\alpha}{K}\right) \log\left(1 - \alpha + \frac{\alpha}{K}\right) - \frac{\alpha(K-1)}{K} \log\left(\frac{\alpha}{K}\right).$$

And the minimum value is attained when $\widehat{p}_{i|j}(\mathbf{W}) = p_{i|j}$ for all $i, j \in [K]$, meaning that all that all global minimizers $\mathbf{W}^*$ induce the same predicted conditional distribution $\widehat{p}_{i|j}(\mathbf{W}^*) = p_{i|j}$ for all $1 \leq i, j \leq K$. $\qquad\square$

## C. Proof for the noiseless case

### C.1. Analysis of Gradient Descent (GD)

*Proof of Theorem 4.1.* $\mathcal{L}(\mathbf{W}_t)$ can be divided into $K$ terms based on different sub-tasks, the loss related to sub-task $j$ is only determined by the $j$-th column of $\widehat{\mathbf{W}}_t$. According to the update rule of GD (6), each column of the weight matrix $\widehat{\mathbf{W}}_t$ is updated independently. We analyze the update of the first column $(\widehat{\mathbf{W}}_t)_1$ and the loss related to the first sub-task $\mathcal{L}_1(\mathbf{W}_t)$ as an example.

**Lemma C.1.** *Assume the weight matrix is initialized as $\mathbf{W}_0 = \mathbf{0}$ and optimized via GD. The weight matrix $\widehat{\mathbf{W}}_t$ satisfies:*

$$(\widehat{\mathbf{W}}_t)_{i_1,j} = (\widehat{\mathbf{W}}_t)_{i_2,j}, \quad \forall i_1, i_2 \neq j, \quad \forall j \in [K].$$

The proof of Lemma C.1 is presented in Appendix C.1.1. With Lemma C.1, we have $(\widehat{\mathbf{W}}_t)_{i,1} = (\widehat{\mathbf{W}}_t)_{j,1}$ for all $i, j \neq 1$. The predicted probability can be expressed as:

$$\widehat{p}_{t,1|1} = \frac{\exp((\widehat{\mathbf{W}}_t)_{1,1})}{\exp((\widehat{\mathbf{W}}_t)_{1,1}) + (K-1)\exp((\widehat{\mathbf{W}}_t)_{2,1})},$$

$$\widehat{p}_{t,i|1} = \frac{\exp((\widehat{\mathbf{W}}_t)_{2,1})}{\exp((\widehat{\mathbf{W}}_t)_{1,1}) + (K-1)\exp((\widehat{\mathbf{W}}_t)_{2,1})}, \quad \forall i \neq 1.$$

The update rule of GD can be expressed as:

$$(\widehat{\mathbf{W}}_{t+1})_{1,1} = (\widehat{\mathbf{W}}_t)_{1,1} + \eta \cdot p_1(1 - \widehat{p}_{t,1|1}),$$

$$(\widehat{\mathbf{W}}_{t+1})_{i,1} = (\widehat{\mathbf{W}}_t)_{i,1} - \eta \cdot p_1\widehat{p}_{t,i|1}, \quad \forall i \neq 1.$$

We denote $\Delta_t = (\widehat{\mathbf{W}}_{t+1})_{1,1} - (\widehat{\mathbf{W}}_{t+1})_{2,1}$. Given the initialization $\mathbf{W}_0 = \mathbf{0}$, we have the initial condition $\Delta_0 = 0$. Then the update rule of $\Delta_t$ is:

$$\Delta_{t+1} = \Delta_t + \eta \cdot p_1\big(1 - \widehat{p}_{t,1|1} + \widehat{p}_{t,2|1}\big).$$

As $\widehat{p}_{t,1|1} + (K-1)\widehat{p}_{t,2|1} = 1$, we have:

$$\Delta_{t+1} = \Delta_t + \eta \cdot p_1 K\widehat{p}_{t,2|1} = \Delta_t + \eta \cdot p_1 \frac{K}{e^{\Delta_t} + (K-1)}.$$

Denote $e^{\Delta_t}$ as $u_t$, the iteration above can be expressed as:

$$u_{t+1} = u_t \cdot \exp\left(\frac{\eta p_1 K}{u_t + K - 1}\right).$$

As we consider $\eta = 1$ in this case, the iteration of $u_t$ can be expressed as:

$$u_{t+1} = u_t \cdot \exp\left(\frac{p_1 K}{u_t + K - 1}\right).$$

As $t \to \infty$, $\Delta_t \to \infty$, leading $u_t \to \infty$. Thus, we have:

$$u_{t+1} \approx u_t \cdot \left(1 + \frac{p_1 K}{u_t + K - 1} + \left(\frac{p_1 K}{u_t + K - 1}\right)^2\right) \tag{14}$$

$$= u_t + p_1 K \cdot \frac{u_t}{u_t + K - 1} + \frac{u_t(p_1 K)^2}{(u_t + K - 1)^2}$$

$$\approx u_t + p_1 K. \tag{15}$$

Equation (14) uses $e^x \approx 1 + x + x^2$ as $x \to 0$. Equation (15) applies $\frac{u_t}{u_t+K-1} \to 1$ as $u_t \to \infty$ and $\frac{u_t(p_1 K)^2}{(u_t+K-1)^2} \to 0$ as $u_t \to \infty$.

As $u_0 = e^0 = 1$, we have:

$$u_t = 1 + p_1 Kt \approx p_1 Kt, \quad \text{as } t \to \infty.$$

Thus, we have:

$$e^{\Delta_t} = u_t \approx p_1 Kt \quad \text{as } t \to \infty.$$

Then $\mathcal{L}_1(\mathbf{W}_T)$ can be expressed with $\Delta_T$ as:

$$
\begin{aligned}
\mathcal{L}_1(\mathbf{W}_T) &= -\log \widehat{p}_{T,1|1} \\
&= -\log \frac{\exp((\widehat{\mathbf{W}}_T)_{1,1})}{\exp((\widehat{\mathbf{W}}_T)_{1,1}) + (K-1)\exp((\widehat{\mathbf{W}}_T)_{2,1})} \\
&= -\log \frac{1}{1 + (K-1)\exp(-\Delta_T)} \\
&\approx (K-1)e^{-\Delta_T} \tag{16} \\
&\approx \frac{(K-1)}{p_1 K T}, \quad \text{as } T \to \infty \tag{17} \\
&\approx \frac{1}{p_1 T}, \quad \text{for sufficiently large } K. \tag{18}
\end{aligned}
$$

In Equation (16), we use the Taylor expansion $\log(1+x) \approx x$ for small $x$, noting that $(K-1)e^{-\Delta_T} \to 0$ as $\Delta_T \to \infty$. In Equation (17), we substitute $\Delta_T \approx \log(p_1 K T)$.

Similarly, we can analyze other columns of the weight matrix $\widehat{\mathbf{W}}_T$ and obtain the same convergence rate of $\mathcal{L}_i(\mathbf{W}_T)$,

$$
\mathcal{L}_i(\mathbf{W}_T) \approx \frac{1}{p_i T}, \quad i = 1, 2, \cdots, K.
$$

Combining the loss related to all sub-tasks, we have:

$$
\mathcal{L}(\mathbf{W}_T) = \sum_{i=1}^{K} p_i \mathcal{L}_i(\mathbf{W}_T) \approx \frac{K}{T}, \quad \text{as } T \to \infty.
$$

$\square$

### C.1.1. PROOF OF LEMMA C.1

*Proof of Lemma C.1.* We prove the lemma by induction.
**Base Case:** According to the initialization (5), we have $\widehat{\mathbf{W}}_0 = \mathbf{0}$. At time step $t = 0$, we have $(\widehat{\mathbf{W}}_t)_{i_1,j} = (\widehat{\mathbf{W}}_t)_{i_2,j} = 0, \quad \forall i_1, i_2 \neq j, \quad \forall j \in [K]$.
**Induction Step:** Assume at time step $t$, the weight matrix $\widehat{\mathbf{W}}_t$ satisfies Lemma C.1. We have:

$$
\widehat{p}_{t,i_1|j} = \frac{\exp((\widehat{\mathbf{W}}_t)_{i_1,j})}{\sum_{l=1}^{K} \exp((\widehat{\mathbf{W}}_t)_{l,j})} = \widehat{p}_{t,i_2|j}.
$$

According to the update rule of GD (6) and gradient of loss (11), we have:

$$
\begin{aligned}
(\widehat{\mathbf{W}}_{t+1})_{i_1,j} &= (\widehat{\mathbf{W}}_t)_{i_1,j} - \eta \cdot p_j \widehat{p}_{t,i_1|j} \quad \text{if } i_1 \neq j. \\
(\widehat{\mathbf{W}}_{t+1})_{i_2,j} &= (\widehat{\mathbf{W}}_t)_{i_2,j} - \eta \cdot p_j \widehat{p}_{t,i_2|j} \quad \text{if } i_2 \neq j.
\end{aligned}
$$

Thus, we have $(\widehat{\mathbf{W}}_{t+1})_{i_1,j} = (\widehat{\mathbf{W}}_{t+1})_{i_2,j}, \quad \forall i_1, i_2 \neq j, \quad \forall j \in [K]$. $\square$

### C.2. Analysis of Muon

*Proof of Theorem 4.2.* Equivalently, the total loss $\mathcal{L}(\mathbf{W}_t)$ can be divided into $K$ terms based on different sub-tasks, the loss related to sub-task $j$ is only determined by the $j$-th column of $\widehat{\mathbf{W}}_t$. $\widehat{\mathbf{W}}_t$ maintains a special structure during the optimization process via Muon, leading to similar updates for different columns of $\widehat{\mathbf{W}}_t$. We will state the special structure in Lemma C.2.

**Lemma C.2.** *Assume the weight matrix is initialized as $\mathbf{W}_0 = \mathbf{0}$ and optimized via Muon. At any time step $t$, the weight matrix $\widehat{\mathbf{W}}_t \in \mathbb{R}^{K \times K}$ can be partitioned into $M^2$ sub-blocks $\{\mathbf{B}_{i,j}\}_{i,j=1}^{M}$, where each block $\mathbf{B}_{i,j} \in \mathbb{R}^{C \times C}$ exhibits the following structure:*

- **Diagonal Blocks** $(i = j)$**:** Each block $(\mathbf{B}_t)_{i,i}$ is a symmetric-constant matrix:

$$(\mathbf{B}_t)_{i,i} = \omega_{i,t}\mathbf{I}_C + \mu_{i,t}(\mathbf{J}_C - \mathbf{I}_C) = \begin{pmatrix} \omega_{i,t} & \mu_{i,t} & \cdots & \mu_{i,t} \\ \mu_{i,t} & \omega_{i,t} & \cdots & \mu_{i,t} \\ \vdots & \vdots & \ddots & \vdots \\ \mu_{i,t} & \mu_{i,t} & \cdots & \omega_{i,t} \end{pmatrix}.$$

- **Off-diagonal Blocks** $(i \neq j)$**:** Each block $(\mathbf{B}_t)_{i,j}$ is a rank-1 constant matrix:

$$(\mathbf{B}_t)_{i,j} = \gamma_{i,j,t}\mathbf{J}_C = \begin{pmatrix} \gamma_{i,j,t} & \gamma_{i,j,t} & \cdots & \gamma_{i,j,t} \\ \gamma_{i,j,t} & \gamma_{i,j,t} & \cdots & \gamma_{i,j,t} \\ \vdots & \vdots & \ddots & \vdots \\ \gamma_{i,j,t} & \gamma_{i,j,t} & \cdots & \gamma_{i,j,t} \end{pmatrix}.$$

where $\omega_{i,t} = (\widehat{\mathbf{W}}_t)_{C(i-1)+1,C(i-1)+1}$, $\mu_{i,t} = (\widehat{\mathbf{W}}_t)_{C(i-1)+2,C(i-1)+1}$, and $\gamma_{i,j,t} = (\widehat{\mathbf{W}}_t)_{C(i-1)+1,C(j-1)+1}$.

The proof of Lemma C.2 is presented in Appendix C.2.1. According to Lemma C.2, the update of different columns of the weight matrix $\widehat{\mathbf{W}}_t$ are similar. We analyze the loss related to the first column of the weight matrix $\widehat{\mathbf{W}}_t$ as an example.

As Lemma C.2 indicates, the weight matrix $\widehat{\mathbf{W}}_t$ maintains a special structure during the optimization process via Muon. Thus, $\mathbf{P} - \widehat{\mathbf{P}}_t$ (defined in Equation (11)) also maintains a similar structure that it can be rewritten as the sum of a diagonal block matrix and a block-wise constant matrix. We apply SVD to $\mathbf{P} - \widehat{\mathbf{P}}_t$ and use the similar notations as in Proposition B.1. Define the last $M$ left singular vectors of $\mathbf{P} - \widehat{\mathbf{P}}_t$ as $\{\mathbf{u}_{t,i}\}_{i=1}^M$ and the last $M$ right singular vectors as $\{\mathbf{v}_{t,i}\}_{i=1}^M$. The update of the first column of the weight matrix $\widehat{\mathbf{W}}_t$ is:

$$\begin{cases} (\widehat{\mathbf{W}}_t)_{1,1} = (\widehat{\mathbf{W}}_{t-1})_{1,1} + \eta\Big(1 - \frac{1}{C} + \sum_{i=1}^M (\mathbf{u}_{t-1,i}\mathbf{v}_{t-1,i}^\top)_{1,1}\Big), \\ (\widehat{\mathbf{W}}_t)_{C,1} = (\widehat{\mathbf{W}}_{t-1})_{C,1} + \eta\Big(-\frac{1}{C} + \sum_{i=1}^M (\mathbf{u}_{t-1,i}\mathbf{v}_{t-1,i}^\top)_{1,1}\Big), \\ (\widehat{\mathbf{W}}_t)_{iC,1} = (\widehat{\mathbf{W}}_{t-1})_{iC,1} + \eta\Big(\sum_{i=1}^M (\mathbf{u}_{t-1,i}\mathbf{v}_{t-1,i}^\top)_{iC,1}\Big), \quad i \neq 1. \end{cases}$$

*Remark* C.3. Using Proposition B.1, each entry in the matrix $\sum_{i=1}^M (\mathbf{u}_{t-1,i}\mathbf{v}_{t-1,i}^\top)$ is of order $\mathcal{O}(M/C) = \mathcal{O}\Big(\frac{M^2}{K}\Big)$. As $M^2 \ll K$, these entries are relatively small compared to 1. We have:

$$\widehat{\mathbf{W}}_t - \widehat{\mathbf{W}}_{t-1} \approx \mathbf{I}_K. \tag{19}$$

Specifically, as mentioned in Proposition 6.1, we have:

$$\| \operatorname{msgn}(\mathbf{P} - \widehat{\mathbf{P}}_t) - \mathbf{I}_K \|_{\max} = o_K(1).$$

As $\widehat{\mathbf{W}}_t$ is initialized as $\mathbf{0}$, summing up with the time step, we have:

$$\begin{cases} (\widehat{\mathbf{W}}_t)_{1,1} = \eta\Big(1 - \frac{1}{C} + \sum_{t'=0}^{t-1}\sum_{i=1}^M (\mathbf{u}_{t',i}\mathbf{v}_{t',i}^\top)_{1,1}\Big), \\ (\widehat{\mathbf{W}}_t)_{C,1} = \eta\Big(-\frac{1}{C} + \sum_{t'=0}^{t-1}\sum_{i=1}^M (\mathbf{u}_{t',i}\mathbf{v}_{t',i}^\top)_{1,1}\Big), \\ (\widehat{\mathbf{W}}_t)_{iC,1} = \eta\Big(\sum_{t'=0}^{t-1}\sum_{i=1}^M (\mathbf{u}_{t',i}\mathbf{v}_{t',i}^\top)_{iC,1}\Big), \quad i \neq 1. \end{cases} \tag{20}$$

According to Lemma C.2, $\widehat{\mathbf{W}}_t$ maintains the special structure, we introduce $s_{t,i|j}^+$ and $s_{t,i|j}^-$ to simplify the notation. Remark that the first index $i \in [M]$ is the group index and the last index $j \in [K]$ is the query index. These scores are defined as:

$$s_{t,i|j}^+ = \exp\Big((\widehat{\mathbf{W}}_t)_{j,j}\Big), \quad \text{denoting the score for a true prediction, } i \text{ is the group includes } j$$

$$s_{t,i|j}^- = \exp\Big((\widehat{\mathbf{W}}_t)_{l,j}\Big), \quad \text{denoting the score for a false prediction, } l \text{ belongs to group } i, l \neq j.$$

where the indices $i$ and $j$ denote that the predicted answer is in group $i$, given that the input is the $j$-th query.

The loss $\mathcal{L}_1(\mathbf{W}_t)$ can be expressed using $s_{t,1|1}^+$ and $s_{t,l|1}^-$ as:

$$\mathcal{L}_1(\mathbf{W}_t) = -\log\left(\frac{s_{t,1|1}^+}{s_{t,1|1}^+ + (C-1)s_{t,1|1}^- + \sum_{l=2}^{M} Cs_{t,i|1}^-}\right). \tag{21}$$

According to Equation (20), all scores related to the first column of $\widehat{\mathbf{W}}_t$ can be expressed by $s_{t,1|1}^+$. We have:

$$s_{t,1|1}^- = s_{t,1|1}^+ \cdot \exp(-\eta t), \tag{22}$$

$$s_{t,i|1}^- = s_{t,1|1}^+ \cdot \exp\left(-\eta t + \frac{\eta t}{C} + \eta \sum_{t'=0}^{t-1}\sum_{l=1}^{M}((\mathbf{u}_{t',l}\mathbf{v}_{t',l}^\top)_{iC,1} - (\mathbf{u}_{t',l}\mathbf{v}_{t',l}^\top)_{1,1})\right), \quad i \neq 1. \tag{23}$$

Denote $\tilde{s}_{t,i} = \sum_{l=1}^{M}((\mathbf{u}_{t',l}\mathbf{v}_{t',l}^\top)_{iC,1} - (\mathbf{u}_{t',l}\mathbf{v}_{t',l}^\top)_{1,1})$. With Equation (22) and Equation (23), $\mathcal{L}_1(\mathbf{W}_t)$ can be expressed as:

$$\mathcal{L}_1(\mathbf{W}_t) = \log\left(1 + (C-1)\exp(-\eta t) + C\cdot\sum_{l=2}^{M}\exp\left(-\eta t + \frac{\eta t}{C} + \eta\sum_{t'=0}^{t-1}\tilde{s}_{t,l}\right)\right).$$

As we consider $\eta = 1$ in this case, we have:

$$\mathcal{L}_1(\mathbf{W}_t) = \log\left(1 + (C-1)\exp(-t) + C\cdot\sum_{l=2}^{M}\exp\left(-t + \frac{t}{C} + \sum_{t'=0}^{t-1}\tilde{s}_{t,l}\right)\right). \tag{24}$$

As $|(\mathbf{u}_{t',l}\mathbf{v}_{t',l}^\top)_{iC,1}| \leq 1$, for all $t', l, i$, we have:

$$|\tilde{s}_{t,l}| \leq \sum_{l=1}^{M}|(\mathbf{u}_{t,l}\mathbf{v}_{t,l}^\top)_{iC,1} - (\mathbf{u}_{t,l}\mathbf{v}_{t,l}^\top)_{1,1}|$$

$$\leq \frac{2M}{C} = \frac{2M^2}{K}, \quad \text{as } K = CM. \tag{25}$$

Substituting the upper bound of $|\tilde{s}_{t,l}|$ in Equation (25) into the expression of $\mathcal{L}_1(\mathbf{W}_t)$ in Equation (24), we have:

$$\mathcal{L}_1(\mathbf{W}_t) \leq \log\left(1 + (C-1)\exp(-t) + C\cdot\sum_{l=2}^{M}\exp\left(-t + \frac{t}{C} + \frac{2M^2t}{K}\right)\right)$$

$$\leq \log\left(1 + (C-1)\exp(-t) + C(M-1)\exp\left(-t + \frac{3M^2t}{K}\right)\right) \tag{26}$$

$$\leq \log\left(1 + K\exp\left(-t + \frac{3M^2t}{K}\right)\right) \tag{27}$$

$$\approx K\cdot\exp\left(-t + \frac{3M^2t}{K}\right), \quad \text{as } t \to \infty. \tag{28}$$

In Equation (26), we relax $2M^2+1$ to $3M^2$ (for $M \geq 1$). In Equation (27), we upper bound $\exp(-t)$ with $\exp\left(-t+\frac{3M^2t}{K}\right)$ and relax $CM-1$ to $K$. As $M^2 \ll K$, the term $\exp\left(-t+\frac{3M^2t}{K}\right) \to 0$ as $t \to \infty$. In Equation (28), we use $\log(1+x) \approx x$ as $x \to 0$.

Similarly, we can analyze other columns of the weight matrix $\widehat{\mathbf{W}}_t$ and obtain the same convergence rate of $\mathcal{L}_i(\mathbf{W}_t)$. Combining the loss related to all sub-tasks, we have:

$$\mathcal{L}(\mathbf{W}_t) = \sum_{i=1}^{K} p_i\mathcal{L}_i(\mathbf{W}_t) \lesssim K\cdot\exp\left(-t + \frac{3M^2t}{K}\right), \quad \text{as } t \to \infty.$$

For the lower bound, the estimation is similar. Combining Equation (25) and Equation (24), we have:

$$\mathcal{L}_1(\mathbf{W}_t) \geq \log\left(1 + (C-1)\exp(-t) + C \cdot \sum_{l=2}^{M} \exp\left(-t + \frac{t}{C} - \frac{2M^2 t}{K}\right)\right)$$

$$\geq \log\left(1 + K\exp\left(-t - \frac{2M^2 t}{K}\right)\right) \tag{29}$$

$$\approx K \cdot \exp\left(-t - \frac{2M^2 t}{K}\right), \quad \text{as } t \to \infty. \tag{30}$$

In Equation (29), we relax $\exp(-t)$ and $\exp(-t + \frac{t}{C} - \frac{2M^2 t}{K})$ to $\exp\left(-t - \frac{2M^2 t}{K}\right)$. We also approximate $K-1$ to $K$ as $K$ is sufficiently large. In Equation (30), we use $\log(1+x) \approx x$ as $x \to 0$.

Combining the loss related to all sub-tasks, we have:

$$\mathcal{L}(\mathbf{W}_t) \gtrsim K \cdot \exp\left(-t - \frac{2M^2 t}{K}\right), \quad \text{as } t \to \infty.$$

Combining the upper and lower bounds, we have:

$$\mathcal{L}(\mathbf{W}_t) \approx Ke^{-(1+o_K(1))t}.$$

$\square$

### C.2.1. PROOF OF LEMMA C.2

*Proof of Lemma C.2.* We prove the lemma by induction.
**Base Case:** At time step $t = 0$, we have $\widehat{\mathbf{W}}_0 = \mathbf{0}$. Thus, each block $\mathbf{B}_{i,j}$ satisfies the structure in Lemma C.2.
**Induction Step:** Assume Lemma C.2 holds from time step $0$ to $t$. We analyze the update from time step $t$ to $t+1$.
According to Equation (11) loss function, we have:

$$\text{msgn}(\nabla_{\mathbf{W}_t}\mathcal{L}(\mathbf{W}_t)) = -\widetilde{\mathbf{E}} \cdot \text{msgn}(\mathbf{P} - \widehat{\mathbf{P}}_t) \cdot \mathbf{E}^\top.$$

Then the update of $\widehat{\mathbf{W}}_t$ according to update rule of Muon (7) can be expressed as:

$$\widehat{\mathbf{W}}_{t+1} = \widehat{\mathbf{W}}_t + \text{msgn}(\mathbf{P} - \widehat{\mathbf{P}}_t).$$

$\widehat{p}_{t,i|j}$ can be expressed with $\widehat{\mathbf{W}}_t$ as:

$$\widehat{p}_{t,i|j} = \frac{\exp((\widehat{\mathbf{W}}_t)_{i,j})}{\sum_{l=1}^{K} \exp((\widehat{\mathbf{W}}_t)_{l,j})}.$$

As Lemma C.2 holds at time step $t$, $\widehat{p}_{t+1,i|j}$ satisfies similar equivalence properties as $\widehat{\mathbf{W}}_t$:

- **True Prediction:** For any $i, j$, if $i, j$ belong to the same group, we have $\widehat{p}_{t,i|i} = \widehat{p}_{t,j|j}$.

- **False Prediction:** For any $i_1, i_2, j_1, j_2$, if $i_1, i_2$ belong to a same group and $j_1, j_2$ belong to a same group, $i_1 \neq j_1$, $i_2 \neq j_2$, we have $\widehat{p}_{t,i_1|j_1} = \widehat{p}_{t,i_2|j_2}$.

Thus, $(\mathbf{P} - \widehat{\mathbf{P}}_t)$ can be decomposed into the difference between a diagonal matrix and a block matrix:

$$\mathbf{P} - \widehat{\mathbf{P}}_t = \mathbf{R}_t^+ - \mathbf{R}_t^-, \quad \text{where}$$

$$\mathbf{R}_t^+ = \begin{pmatrix} p_C(1 - \widehat{p}_{t,C|C} + \widehat{p}_{t,C-1|C})\mathbf{I}_C & \mathbf{0} & \cdots & \mathbf{0} \\ \mathbf{0} & p_{2C}(1 - \widehat{p}_{t,2C|2C} + \widehat{p}_{t,2C-1|2C})\mathbf{I}_C & \cdots & \vdots \\ \vdots & \vdots & \ddots & \vdots \\ \mathbf{0} & \mathbf{0} & \cdots & p_K(1 - \widehat{p}_{t,K|K} + \widehat{p}_{t,K-1|K})\mathbf{I}_C \end{pmatrix}, \tag{31}$$

$$\mathbf{R}_t^- = \begin{pmatrix} p_C\widehat{p}_{t,C-1|C}\mathbf{J}_C & p_C\widehat{p}_{t,2C|C}\mathbf{J}_C & \cdots & p_C\widehat{p}_{t,K|C}\mathbf{J}_C \\ p_{2C}\widehat{p}_{t,C|2C}\mathbf{J}_C & p_{2C}\widehat{p}_{t,2C-1|2C}\mathbf{J}_C & \cdots & p_{2C}\widehat{p}_{t,K|2C}\mathbf{J}_C \\ \vdots & \vdots & \ddots & \vdots \\ p_K\widehat{p}_{t,C|K}\mathbf{J}_C & p_K\widehat{p}_{t,2C|K}\mathbf{J}_C & \cdots & p_K\widehat{p}_{t,K-1|K}\mathbf{J}_C \end{pmatrix}. \tag{32}$$

Then the structure of $\widehat{\mathbf{W}}_{t+1}$ is suitable for setting in Proposition B.1. Using Proposition B.1, we have $\mathrm{msgn}(\mathbf{P} - \widehat{\mathbf{P}}_t)$ can be expressed as sum of an identity matrix and a block matrix sharing the same structure as $\mathbf{B}_t'$. Thus, each block $\mathbf{B}_{i,j}$ in $\widehat{\mathbf{W}}_{t+1}$ satisfies the structure in Lemma C.2. $\qquad\square$

## D. Proof for the Noisy Case

### D.1. Analysis of Gradient Descent (GD)

We prove Lemma D.1, Lemma D.2 before completing the total proof of Theorem 5.5.

**Lemma D.1.** *(General Lemma C.1) Assume the weight matrix is initialized as* $\mathbf{W}_0 = \mathbf{0}$ *and optimized via GD. The weight matrix* $\widehat{\mathbf{W}}_t$ *satisfies:*

$$(\widehat{\mathbf{W}}_t)_{i_1,j} = (\widehat{\mathbf{W}}_t)_{i_2,j}, \quad \forall i_1, i_2 \neq j, \quad \forall j \in [K].$$

*Proof of Lemma D.1.* The proof is similar to that of Lemma C.1. $\qquad\square$

**Lemma D.2.** *Consider the following discrete update:*

$$\begin{cases} x_{t+1} - x_t = \frac{A}{e^{x_t}+B} - D, & A > 0, B > 0, D > 0. \\ x_0 = 0, \quad A - BD > D, \quad A < 4B. \end{cases}$$

*Denote the fixed point of the update as* $x^*$*, satisfying* $x^* = \log\left(\frac{A-BD}{D}\right)$*. Assume* $x_t \leq x^*$ *for all* $t$ *throughout the iterations, we have:*

$$x^* - x_t \geq x^* \cdot \left(1 - \frac{A}{4B}\right)^t.$$

*Proof of Lemma D.2.* Define an auxiliary function $f : \mathbb{R} \to \mathbb{R}$ as $f(x_t) = \frac{A}{e^{x_t}+B} - D$. According to Mean Value Theorem, we have:

$$f(x^*) - f(x_t) = f'(\xi_t)(x^* - x_t), \quad \text{for some } \xi_t \in (x_t, x^*).$$

Substituting $f(x_t) = x_{t+1} - x_t$, we have:

$$x^* - x_{t+1} = (1 + f'(\xi_t))(x^* - x_t).$$

We analyze $f'(x)$ as follows:

$$f'(x) = -\frac{Ae^x}{(e^x + B)^2} = -\frac{A}{e^x + 2B + B^2 e^{-x}} \geq -\frac{A}{4B}.$$

Thus, we have:

$$x^* - x_{t+1} \geq \left(1 - \frac{A}{4B}\right) \cdot (x^* - x_t) \quad \text{as } A < 4B.$$

Combining the above inequality recursively, we have:

$$x^* - x_t \geq (x^* - x_0) \cdot \left(1 - \frac{A}{4B}\right)^t = x^*\left(1 - \frac{A}{4B}\right)^t.$$

$\qquad\square$

*Proof of Theorem 5.5.* According to the update rule of GD in (6) and loss function in (13), each column of the weight matrix $\widehat{\mathbf{W}}_t$ is updated independently. Similar to the noiseless case, we analyze the update of the first column of $\widehat{\mathbf{W}}_t$ and the loss related to the first task $\mathcal{L}_1(\mathbf{W}_t)$ as an example.

With Lemma D.1, we have $(\widehat{\mathbf{W}}_t)_{i,1} = (\widehat{\mathbf{W}}_t)_{j,1}$ for all $i, j \neq 1$. Similar to the noiseless case, the predicted probability can be expressed as:

$$\widehat{p}_{t,1|1} = \frac{\exp((\widehat{\mathbf{W}}_t)_{1,1})}{\exp((\widehat{\mathbf{W}}_t)_{1,1}) + (K-1)\exp((\widehat{\mathbf{W}}_t)_{2,1})},$$

$$\widehat{p}_{t,i|1} = \frac{\exp((\widehat{\mathbf{W}}_t)_{2,1})}{\exp((\widehat{\mathbf{W}}_t)_{1,1}) + (K-1)\exp((\widehat{\mathbf{W}}_t)_{2,1})}, \quad \forall i \neq 1.$$

The update rule of GD can be expressed as:

$$(\widehat{\mathbf{W}}_{t+1})_{1,1} = (\widehat{\mathbf{W}}_t)_{1,1} + \eta \cdot p_1\left(1 - \alpha + \frac{\alpha}{K} - \widehat{p}_{t,1|1}\right),$$

$$(\widehat{\mathbf{W}}_{t+1})_{i,1} = (\widehat{\mathbf{W}}_t)_{i,1} + \eta \cdot p_1\left(\frac{\alpha}{K} - \widehat{p}_{t,i|1}\right), \quad \forall i \neq 1.$$

We use the same notation $\Delta_t$ in Appendix C.1, define $\Delta_t = (\widehat{\mathbf{W}}_t)_{1,1} - (\widehat{\mathbf{W}}_t)_{2,1}$. The update rule of $\Delta_t$ is:

$$\Delta_{t+1} = \Delta_t + \eta \cdot p_1\left(1 - \alpha - \widehat{p}_{t,1|1} + \widehat{p}_{t,2|1}\right).$$

As $\widehat{p}_{t,1|1} + (K-1)\widehat{p}_{t,2|1} = 1$, we have:

$$\Delta_{t+1} = \Delta_t + \eta p_1 K \cdot \widehat{p}_{t,2|1} - \alpha\eta p_1$$

$$= \Delta_t + \frac{\eta p_1 K}{e^{\Delta_t} + (K-1)} - \alpha\eta p_1.$$

Solving the fixed-point equation $\frac{K}{e^{\Delta^*}+K-1} = \alpha$ gives

$$\Delta^* = \log\left(K \cdot \frac{\left(1 - \alpha + \frac{\alpha}{K}\right)}{\alpha}\right).$$

By Proposition 5.4, linear stability requires $\eta p_1 \lesssim 1$. We assume $\eta p_1 \leq 1$ in the following analysis.

Define $f : \mathbb{R} \to \mathbb{R}$, as $f(\Delta_t) = \frac{\eta p_1 K}{e^{\Delta_t}+(K-1)} - \alpha\eta p_1$.
We analyze $f'(x)$ as follows:

$$f'(x) = -\frac{\eta p_1 K e^x}{(e^x + (K-1))^2} \geq -\frac{p_1 K}{4(K-1)}.$$

As $\widehat{\mathbf{W}}_t$ is initialized as $\mathbf{0}$, we have the initial condition $\Delta_0 = 0$. Then using Lemma D.2, we have:

$$\Delta^* - \Delta_t \geq \Delta^* \cdot \left(1 - \frac{K\eta p_1}{4(K-1)}\right)^t, \quad \text{as } \eta p_1 \leq 1,$$

$$\overline{\approx} \Delta^*\left(1 - \frac{1}{4}\eta p_1\right)^t, \quad \text{as } K \text{ is large enough.}$$

For $x \leq 1/4$. we have $(1-x) \geq e^{-2x}$. Then $\Delta^* - \Delta_t$ can be further bounded as:

$$\Delta^* - \Delta_t \geq \Delta^* \cdot e^{-\frac{1}{2}\eta p_1 t}.$$

Predicted probability can be expressed using $\Delta_t$ as:

$$\widehat{p}_{t,1|1} = \frac{e^{\Delta_t}}{e^{\Delta_t} + (K-1)}, \qquad \widehat{p}_{t,i|1} = \frac{1}{e^{\Delta_t} + (K-1)}, \quad \forall i \neq 1.$$

Then $\mathcal{L}_1(\mathbf{W}_T)$ can be expressed using $\Delta_T$ as:

$$\mathcal{L}_1(\mathbf{W}_T) = -\left(1 - \alpha + \frac{\alpha}{K}\right)\log\widehat{p}_{T,1|1} - \frac{\alpha}{K}\sum_{l=2}^{K}\log\widehat{p}_{T,l|1}$$

$$= -\left(1 - \alpha + \frac{\alpha}{K}\right)\log\frac{e^{\Delta_T}}{e^{\Delta_T} + (K-1)} - \frac{\alpha}{K}\sum_{l=2}^{K}\log\frac{1}{e^{\Delta_T} + (K-1)}$$

$$= -\left(1 - \alpha + \frac{\alpha}{K}\right)\Delta_T + \log(e^{\Delta_T} + (K-1)).$$

According to Proposition 3.4, the optimal $\mathcal{L}^*$ is the entropy of data distribution. It is achieved when the predicted distribution matches the data distribution, that:

$$\widehat{p}_{1|1} = \left(1 - \alpha + \frac{\alpha}{K}\right), \qquad \widehat{p}_{i|1} = \frac{\alpha}{K}, \quad \forall i \neq 1.$$

When $\mathcal{L}(\mathbf{W}_t)$ achieved its optimal value, $\Delta_t$ reaches its fixed point $\Delta^*$. We analyze the convergence rate of $\mathcal{L}_1(\mathbf{W}_T)$ to $\mathcal{L}_1^*$ as follows:

$$\mathcal{L}_1(\mathbf{W}_T) - \mathcal{L}_1^* = \left(1 - \alpha + \frac{\alpha}{K}\right)(\Delta^* - \Delta_T) + \log\left(\frac{e^{\Delta_T} + (K-1)}{e^{\Delta^*} + (K-1)}\right)$$

$$= \left(1 - \alpha + \frac{\alpha}{K}\right)(\Delta^* - \Delta_T) + \log\left(1 + \frac{e^{\Delta^*}(e^{\Delta_T - \Delta^*} - 1)}{e^{\Delta^*} + (K-1)}\right).$$

Consider an auxiliary function $g : \mathbb{R} \to \mathbb{R}$, defined as $g(x) = \left(1 - \alpha + \frac{\alpha}{K}\right)x + \log\left(1 + \frac{e^{\Delta^*}(e^x - 1)}{e^{\Delta^*} + (K-1)}\right)$. The derivative of $g(x)$ is:

$$g'(x) = \left(1 - \alpha + \frac{\alpha}{K}\right) + \frac{e^{\Delta^* + x}}{e^{\Delta^*} + (K-1) + e^{\Delta^*}(e^x - 1)} \geq 0.$$

We have $g(x)$ is monotonically increasing. As $\Delta^* - \Delta_T \geq \Delta^* \cdot e^{-\frac{1}{2}\eta p_1 T}$, we have:

$$\mathcal{L}_1(\mathbf{W}_T) - \mathcal{L}_1^* \geq g(\Delta^* \cdot e^{-\frac{1}{2}\eta p_1 T})$$

$$= \left(1 - \alpha + \frac{\alpha}{K}\right)\Delta^* \cdot e^{-\frac{1}{2}\eta p_1 T} + \log\left(1 + \frac{e^{\Delta^*}(\exp(-e^{-\frac{1}{2}\eta p_1 T}\Delta^*) - 1)}{e^{\Delta^*} + (K-1)}\right)$$

$$= \left(1 - \alpha + \frac{\alpha}{K}\right)\Delta^* \cdot e^{-\frac{1}{2}\eta p_1 T} + \log\left(1 + \left(1 - \alpha + \frac{\alpha}{K}\right)(\exp(-e^{-\frac{1}{2}\eta p_1 T}\Delta^*) - 1)\right). \tag{33}$$

In Equation (33), we substitute $e^{\Delta^*} = K \cdot \frac{(1 - \alpha + \frac{\alpha}{K})}{\alpha}$. As $T \to \infty$ and $\Delta^* = \log\left(K \cdot \frac{1 - \alpha + \frac{\alpha}{K}}{\alpha}\right)$, we have:

$$e^{-\frac{1}{2}\eta p_1 T}\Delta^* \to 0.$$

With Taylor expansion of $\exp(x)$ and $\log(1 + x)$ at $x = 0$, the log term can be estimated as:

$$\log\left(1 + \left(1 - \alpha + \frac{\alpha}{K}\right) \cdot (\exp(-\Delta^* \cdot e^{-\frac{1}{2}\eta p_1 T}) - 1)\right)$$

$$= \log\left(1 - \left(1 - \alpha + \frac{\alpha}{K}\right) \cdot \left(\Delta^* \cdot e^{-\frac{1}{2}\eta p_1 T} - \frac{(\Delta^*)^2 \cdot e^{-\eta p_1 T}}{2} + o(e^{-\eta p_1 T})\right)\right)$$

$$= -\left(1 - \alpha + \frac{\alpha}{K}\right) \cdot \left(\Delta^* \cdot e^{-\frac{1}{2}\eta p_1 T} - \frac{(\Delta^*)^2 \cdot e^{-\eta p_1 T}}{2}\right) - \left(1 - \alpha + \frac{\alpha}{K}\right)^2\frac{(\Delta^*)^2 \cdot e^{-\eta p_1 T}}{2} + o(e^{-\eta p_1 T})$$

$$= -\left(1 - \alpha + \frac{\alpha}{K}\right)\Delta^* \cdot e^{-\frac{1}{2}\eta p_1 T} + \left(1 - \alpha + \frac{\alpha}{K}\right)\left(\alpha - \frac{\alpha}{K}\right)\frac{(\Delta^*)^2 \cdot e^{-\eta p_1 T}}{2}.$$

Then the convergence rate of $\mathcal{L}_1(\mathbf{W}_T)$ to $\mathcal{L}_1^*$ can be estimated as:

$$\mathcal{L}_1(\mathbf{W}_T) - \mathcal{L}_1^*$$
$$\geq \left(1 - \alpha + \frac{\alpha}{K}\right)\Delta^* \cdot e^{-\frac{1}{2}\eta p_1 T} - \left(1 - \alpha + \frac{\alpha}{K}\right)\Delta^* \cdot e^{-\frac{1}{2}\eta p_1 T} + \left(1 - \alpha + \frac{\alpha}{K}\right)\left(\alpha - \frac{\alpha}{K}\right)\frac{(\Delta^*)^2 \cdot e^{-\eta p_1 T}}{2}$$
$$= \left(1 - \alpha + \frac{\alpha}{K}\right)\left(\alpha - \frac{\alpha}{K}\right)\frac{(\Delta^*)^2 \cdot e^{-\eta p_1 T}}{2}$$
$$= \Omega\left((\log K)^2 \cdot e^{-\eta p_1 T}\right).$$

For other columns of the weight matrix $\widehat{\mathbf{W}}_t$, as $p_1 \geq p_2 \geq \cdots \geq p_M$, we have $\eta p_i$ also satisfies $\eta p_i \leq 1$ when $\eta p_1 \leq 1$. Then the proof of first column can be directly applied to other columns. We have:

$$\mathcal{L}_i(\mathbf{W}_T) - \mathcal{L}_i^* = \left(1 - \alpha + \frac{\alpha}{K}\right)\left(\alpha - \frac{\alpha}{K}\right)\frac{(\Delta^*)^2 \cdot e^{-\eta p_i T}}{2}, \quad \text{for all } i \in [K].$$

Combining the loss related to all tasks, we have:

$$\mathcal{L}(\mathbf{W}_T) - \mathcal{L}^* = \sum_{i=1}^{K} p_i(\mathcal{L}_i(\mathbf{W}_T) - \mathcal{L}_i^*)$$
$$\geq \sum_{i=1}^{K} p_i \left(1 - \alpha + \frac{\alpha}{K}\right) \cdot \left(\alpha - \frac{\alpha}{K}\right)\frac{(\Delta^*)^2 \cdot e^{-\eta p_i T}}{2}$$
$$= \left(1 - \alpha + \frac{\alpha}{K}\right) \cdot \left(\alpha - \frac{\alpha}{K}\right)\frac{(\Delta^*)^2}{2}\sum_{i=1}^{K} p_i e^{-\eta p_i T}. \tag{34}$$

$\square$

### D.2. Analysis of Muon

Proof of Theorem 5.1 and Theorem 5.2 is as follows. Theorem 5.2 can be viewed as an extension of Theorem 5.1. According to the gradient in the noisy case (Equation (13)), we have:

$$-\mathbf{G}_t = \mathbf{P}' - \widehat{\mathbf{P}'}_t.$$

It can be rewritten as:
$$-\mathbf{G}_t = \mathbf{R}_t^+ - \mathbf{R}_t^-, \quad \text{where}$$

$$\mathbf{R}_t^+ = \begin{pmatrix} p_1(1 - \alpha + \widehat{p}_{t,2|1} - \widehat{p}_{t,1|1}) & \cdots & \mathbf{0} \\ \vdots & \ddots & \vdots \\ \mathbf{0} & \cdots & p_K(1 - \alpha + \widehat{p}_{t,K-1|K} - \widehat{p}_{t,K|K}) \end{pmatrix},$$

with the diagonal entry $(\mathbf{R}_t^+)_{i,i} = p_i(1 - \alpha + \widehat{p}_{t,j|i} - \widehat{p}_{t,i|i})$, where $j \neq i$ and $j$ is in the same group as $i$. $\mathbf{R}_t^-$ is defined as:

- For the diagonal entry:

$$(\mathbf{R}_t^-)_{i,i} = -p_i\left(\frac{\alpha}{K} - \widehat{p}_{t,j|i}\right), \quad j \text{ is the same index chosen for } (\mathbf{D}_t)_{i,i}.$$

- For the off-diagonal entry:

$$(\mathbf{R}_t^-)_{l,i} = -p_i\left(\frac{\alpha}{K} - \widehat{p}_{t,l|i}\right), \quad \forall l \neq i.$$

Since $\mathbf{W}_t$ is initialized as $\mathbf{0}$, $\mathbf{R}_t^+$ is a block-diagonal matrix and $\mathbf{R}_t^-$ is a block-wise constant matrix. According to Proposition B.4, $\text{msgn}(\mathbf{G}_0)$ shares the same block structure, remaining a block-wise constant matrix. We can prove $\mathbf{R}_t^+$

remains a block-diagonal matrix and $\mathbf{R}_t^-$ remains a block-wise constant matrix for all $t$ using induction. The proof is similar to that of Lemma C.2. Thus, we have:

$$\mathbf{R}_t^+ = \begin{pmatrix} p_C(1 - \alpha + \widehat{p}_{t,C-1|C} - \widehat{p}_{t,C|C})\mathbf{I}_C & \cdots & \mathbf{0} \\ \vdots & \ddots & \vdots \\ \mathbf{0} & \cdots & p_K(1 - \alpha + \widehat{p}_{t,K-1|K} - \widehat{p}_{t,K|K})\mathbf{I}_C \end{pmatrix}, \tag{35}$$

$$-\mathbf{R}_t^- = \begin{pmatrix} p_C(\frac{\alpha}{K} - \widehat{p}_{t,C-1|C})\mathbf{J}_C & \cdots & p_C(\frac{\alpha}{K} - \widehat{p}_{t,K|C})\mathbf{J}_C \\ \vdots & \ddots & \vdots \\ p_K(\frac{\alpha}{K} - \widehat{p}_{t,C|K})\mathbf{J}_C & \cdots & p_K(\frac{\alpha}{K} - \widehat{p}_{t,K-1|K})\mathbf{J}_C \end{pmatrix}. \tag{36}$$

Similar to the noiseless case, the weight matrix $\widehat{\mathbf{W}}_t$ maintains the special structure throughout the training process. According to Proposition B.4, $\mathrm{msgn}(\mathbf{G})$ can be estimated as:

$$\mathrm{msgn}(\mathbf{G}_t) \approx \begin{pmatrix} \mathrm{sgn}(1 - \alpha + \widehat{p}_{t,C-1|C} - \widehat{p}_{t,C|C})\mathbf{I}_C & \cdots & \mathbf{0} \\ \vdots & \ddots & \vdots \\ \mathbf{0} & \cdots & \mathrm{sgn}(1 - \alpha + \widehat{p}_{t,K-1|K} - \widehat{p}_{t,K|K})\mathbf{I}_C \end{pmatrix}.$$

When $\widehat{p}_{t,iC|iC} - \widehat{p}_{t,iC-1|iC} < 1 - \alpha$ holds for all $i \in [M]$, $\mathrm{msgn}(\mathbf{G}_t)$ can be estimated as $\mathbf{I}_K$. Otherwise, the sign matrix will have negative entries on the diagonal, which leads to some diagonal blocks of the weight matrix to decrease instead of increase.

For a block where $\widehat{p}_{t,iC|iC} - \widehat{p}_{t,iC-1|iC} > 1 - \alpha$, the corresponding entry in $\mathrm{msgn}(\mathbf{G}_t)$ flips to $-1$. The weight matrix $\widehat{\mathbf{W}}_t$ will decrease at the corresponding block, which further decreases $\widehat{p}_{t,iC|iC} - \widehat{p}_{t,iC-1|iC}$. The decrease can be viewed as approximately returning to the previous state. At the next iteration, the predicted probability corresponding to this block will satisfy $\widehat{p}_{t+1,iC|iC} - \widehat{p}_{t+1,iC-1|iC} < 1 - \alpha$, then the weight matrix $\widehat{\mathbf{W}}_{t+1}$ will increase at the corresponding block. Focusing on this specific block, we observe an oscillation in the weight matrix $\widehat{\mathbf{W}}_t$, where the fluctuations are characterized by a magnitude of roughly $2\eta$. As a result, the predicted probability $\widehat{p}_t$ also oscillates around the optimal value, leading to oscillations in the loss $\mathcal{L}(\mathbf{W}_t)$. The rigorous analysis will be provided below.

Thus, when $\widehat{p}_{t,iC|iC} - \widehat{p}_{t,iC-1|iC} < 1 - \alpha$ does not hold for some $i \in [M]$, at some iterations, parts of the weight matrix $\widehat{\mathbf{W}}_t$ begin to oscillate instead of converging. Therefore, the trajectory of Muon can be divided into three phases. Denote $T'$ as the time Phase 1 ends and $T''$ as the time Phase 2 ends.

- **Phase 1** (1 - $T'$)**:** For all $t$ in this phase, $\widehat{p}_{t,iC|iC} - \widehat{p}_{t,iC-1|iC} < 1 - \alpha$ holds for all $i \in [M]$. In this phase, the weight matrix $\widehat{\mathbf{W}}_t$ increases at all blocks, and the loss $\mathcal{L}(\mathbf{W}_t)$ decreases monotonically. $T'$ is defined as the first timesuch that there exists some $i \in [M]$ satisfying $\widehat{p}_{T',iC|iC} - \widehat{p}_{T',iC-1|iC} \geq 1 - \alpha$.

- **Phase 2** ($T'$ - $T''$)**:** The gap between $\widehat{p}_{t,iC|iC}$ and $\widehat{p}_{t,iC-1|iC}$ continues to widen over iterations until the threshold $1 - \alpha$ is reached. After some time step $T'$, there exists some $i \in [M]$ such that $\widehat{p}_{T',iC|iC} - \widehat{p}_{T',iC-1|iC} \geq 1 - \alpha$. From time $T'$, some blocks of the weight matrix $\widehat{\mathbf{W}}_t$ begin to oscillate, $\widehat{p}_t$ also oscillates around the optimal value, and the loss $\mathcal{L}(\mathbf{W}_t)$ begins to oscillate consequently.

  $T''$ is defined as the first time step such that $\widehat{p}_{t,iC|iC} - \widehat{p}_{t,iC-1|iC} \geq 1 - \alpha$ has emerged at some time step $t < T''$ for all $i \in [M]$.

- **Phase 3** ($T''$ - $\infty$)**:** $\widehat{p}_{t,iC|iC} - \widehat{p}_{t,iC-1|iC} > 1 - \alpha$ has been observed for all $i \in [M]$. In this phase, all blocks of the weight matrix $\widehat{\mathbf{W}}_t$ oscillate, and the loss $\mathcal{L}(\mathbf{W}_t)$ oscillates around the optimal value.

### D.2.1. PHASE TRANSITION ANALYSIS

**Loss dynamics:** The dynamics are similar to those of the noiseless case in Appendix C.2. As the updates of different columns of the weight matrix $\widehat{\mathbf{W}}_t$ are similar, we focus on the update of the first column of $\widehat{\mathbf{W}}_t$ and the loss related to the first task $\mathcal{L}_1(\mathbf{W}_t)$ again, as an example. We use the same notation $s_{t,i|j}^+$, $s_{t,i|j}^-$ as defined in Appendix C.2.

We have:

$$
\begin{cases}
(\widehat{\mathbf{W}}_t)_{1,1} = (\widehat{\mathbf{W}}_{t-1})_{1,1} + \eta\left(1 - \frac{1}{C} + \sum_{l=1}^{M}(\mathbf{u}_{t,l}\mathbf{v}_{t,l}^{\top})_{1,1}\right), \\[2mm]
(\widehat{\mathbf{W}}_t)_{C,1} = (\widehat{\mathbf{W}}_{t-1})_{C,1} + \eta\left(-\frac{1}{C} + \sum_{l=1}^{M}(\mathbf{u}_{t,l}\mathbf{v}_{t,l}^{\top})_{1,1}\right), \\[2mm]
(\widehat{\mathbf{W}}_t)_{iC,1} = (\widehat{\mathbf{W}}_{t-1})_{iC,1} + \eta\sum_{l=1}^{M}(\mathbf{u}_{t,l}\mathbf{v}_{t,l}^{\top})_{iC,1}, \quad i \neq 1.
\end{cases}
$$

As we did in the noiseless case, we can express all scores related to the first column using $s_{t,1|1}^{+}$:

$$
s_{t,1|1}^{-} = s_{t,1|1}^{+} \cdot \exp(-\eta t)
$$

$$
s_{t,i|1}^{-} = s_{t,1|1}^{+} \cdot \exp\left(-\eta t + \frac{\eta t}{C} + \eta\sum_{t'=0}^{t-1}\tilde{s}_{t,i}\right), \quad i \neq 1.
$$

The loss related to the first task is:

$$
\mathcal{L}_1(\mathbf{W}_t) = -\left(1 - \alpha + \frac{\alpha}{K}\right)\log\widehat{p}_{t,1|1} - \frac{\alpha}{K}\sum_{i=1}^{K}\log\widehat{p}_{t,i|1}.
$$

It can be expressed using $s_{t,1|1}^{+}$ as:

$$
\begin{aligned}
\mathcal{L}_1(\mathbf{W}_t) &= -\log\widehat{p}_{t,1|1} - \frac{\alpha}{K}\cdot(C-1)\log\left(\frac{s_{t,1|1}^{-}}{s_{t,1|1}^{+}}\right) - \frac{\alpha}{K}\cdot C\sum_{i=2}^{M}\log\left(\frac{s_{t,i|1}^{-}}{s_{t,1|1}^{+}}\right) \\[2mm]
&= -\log\widehat{p}_{t,1|1} + \frac{\alpha(C-1)}{K}\eta t + \frac{\alpha C}{K}\sum_{i=2}^{M}\left(\eta t - \frac{\eta t}{C} - \eta\sum_{t'=0}^{t-1}\tilde{s}_{t,i}\right) \\[2mm]
&\leq -\log\widehat{p}_{t,1|1} + \frac{\alpha(C-1)}{K}\eta t + \frac{\alpha C}{K}\sum_{i=2}^{M}\left(\eta t - \frac{\eta t}{C} + \frac{2M\eta t}{C}\right) \qquad (37) \\[2mm]
&= -\log\widehat{p}_{t,1|1} + \frac{K + 2M^2 - 3M}{K}\alpha\eta t, \quad \text{for } t < T'. \qquad (38)
\end{aligned}
$$

Equation (37) uses the absolute value bound of $\tilde{s}_{t,i}$ that $|\tilde{s}_{t,i}| \leq \frac{2M}{C}$.

Similar to the noiseless case, we analyze the dynamics of $\widehat{p}_{t,1|1}$ as follows:

$$
\begin{aligned}
\widehat{p}_{t,1|1} &= \frac{s_{t,1|1}^{+}}{s_{t,1|1}^{+} + (C-1)s_{t,2|1}^{-} + C\cdot\sum_{i=2}^{M}s_{t,i|1}^{-}} \\[2mm]
&= \frac{1}{1 + (C-1)\exp(-\eta t) + C\cdot\sum_{i=2}^{M}\exp\left(-\eta t + \frac{\eta t}{C} + \eta\sum_{t'=0}^{t-1}\tilde{s}_{t,i}\right)} \\[2mm]
&\geq \frac{1}{1 + (C-1)\exp(-\eta t) + C\cdot\sum_{i=2}^{M}\exp\left(-\eta t + \frac{\eta t}{C} + \frac{2M\eta t}{C}\right)} \\[2mm]
&\geq \frac{1}{1 + (K-1)\exp\left(-\eta t + \frac{3M^2\eta t}{K}\right)}, \quad \text{for all } t < T'.
\end{aligned}
$$

Substituting the lower bound of $\widehat{p}_{t,1|1}$ into the loss expression Equation (38), we have:

$$
\begin{aligned}
\mathcal{L}_1(\mathbf{W}_t) &\leq \log\left(1 + (K-1)\exp\left(-\eta t + \frac{3M^2\eta t}{K}\right)\right) + \frac{K + 2M^2 - 3M}{K}\alpha\eta t, \\[2mm]
&\lesssim K\exp\left(-\eta t + \frac{3M^2\eta t}{K}\right) + \frac{K + 2M^2 - 3M}{K}\alpha\eta t, \quad \text{for } t < T'.
\end{aligned}
$$

Combining the loss related to all tasks, we have:

$$\mathcal{L}(\mathbf{W}_t) = \sum_{i=1}^{K} p_i \mathcal{L}_i(\mathbf{W}_t)$$

$$\leq K \exp\left(-\eta t + \frac{3M^2 \eta t}{K}\right) + \frac{K + 2M^2 - 3M}{K} \alpha \eta t, \quad \text{for } t < T'. \tag{39}$$

Using the absolute value bound of $\tilde{s}_{t,i}$ that $|\tilde{s}_{t,i}| \leq \frac{2M}{C}$, we can estimate the lower bound of $\mathcal{L}_1(\mathbf{W}_t)$ as:

$$\mathcal{L}_1(\mathbf{W}_t) \geq -\log \widehat{p}_{t,1|1} + \frac{\alpha(C-1)}{K} \eta t + \frac{\alpha C}{K} \sum_{i=2}^{M} \left(\eta t - \frac{\eta t}{C} - \frac{2M\eta t}{C}\right)$$

$$= -\log \widehat{p}_{t,1|1} + \frac{K - 2M^2 + M}{K} \alpha \eta t, \quad \text{for } t < T'.$$

We can also estimate the upper bound of $\widehat{p}_{t,1|1}$ as:

$$\widehat{p}_{t,1|1} \leq \frac{1}{1 + (C-1)\exp(-\eta t) + C \cdot \sum_{i=2}^{M} \exp\left(-\eta t + \frac{\eta t}{C} - \frac{2M\eta t}{C}\right)}$$

$$\leq \frac{1}{1 + (K-1)\exp\left(-\eta t - \frac{2M^2 \eta t}{K}\right)}, \quad \text{for all } t < T'.$$

Thus, the lower bound of $\mathcal{L}_1(\mathbf{W}_t)$ is:

$$\mathcal{L}_1(\mathbf{W}_t) \geq \log\left(1 + (K-1)\exp\left(-\eta t - \frac{2M^2 \eta t}{K}\right)\right) + \frac{K - 2M^2 + M}{K} \alpha \eta t,$$

$$\gtrsim (K-1)\exp\left(-\eta t - \frac{2M^2 \eta t}{K}\right) + \frac{K - 2M^2 + M}{K} \alpha \eta t, \quad \text{for } t < T'.$$

Combining the loss of all sub-tasks, we have:

$$\mathcal{L}(\mathbf{W}_t) \gtrsim (K-1)\exp\left(-\eta t - \frac{2M^2 \eta t}{K}\right) + \frac{K - 2M^2 + M}{K} \alpha \eta t, \quad \text{for } t < T'.$$

**Analysis of $T'$ and $T''$:** As $|\tilde{s}_{t,i}| \leq \frac{2M}{C}$, we have:

$$\exp\left(-\eta t + \frac{\eta t}{C} - \frac{2M\eta t}{C}\right) \leq \frac{s_{t,i|1}^-}{s_{t,1|1}^+} \leq \exp\left(-\eta t + \frac{\eta t}{C} + \frac{2M\eta t}{C}\right).$$

The actual iteration for each task can be bounded by two auxiliary iterations. Denote the $s_{t,i|1}^-$ for these two auxiliary iterations as $\left(s_{t,i|1}^-\right)_1$ and $\left(s_{t,i|1}^-\right)_2$, the number of steps that satisfies $\widehat{p}_{t,1|1} - \widehat{p}_{t,2|1} \leq 1 - \alpha$ as $T_1$ and $T_2$ respectively. The first auxiliary iteration is defined as:

$$\left(s_{t,i|1}^-\right)_1 = s_{t,1|1}^+ \cdot \exp\left(-\eta t + \frac{\eta t}{C} - \frac{2M\eta t}{C}\right),$$

and the second auxiliary iteration is defined as:

$$\left(s_{t,i|1}^-\right)_2 = s_{t,1|1}^+ \cdot \exp\left(-\eta t + \frac{\eta t}{C} + \frac{2M\eta t}{C}\right).$$

For $t < T_1$, predicted probability of task in all groups satisfies $\widehat{p}_{t,iC|iC} - \widehat{p}_{t,iC-1|iC} < 1 - \alpha$, which means phase 1 hasn't ended yet. For $t > T_2$, $\widehat{p}_{t,iC|iC} - \widehat{p}_{t,iC-1|iC} > 1 - \alpha$ has happened for all $i \in [M]$, at some time step before $t$, which means phase 2 must have ended. The first auxiliary iteration gives a lower bound of $T'$, and the second auxiliary iteration gives an upper bound of $T''$.

For the first auxiliary iteration, we have:

$$
\begin{aligned}
\widehat{p}_{t,1|1} - \widehat{p}_{t,2|1} &= \frac{s_{t,1|1}^+ - s_{t,2|1}^-}{s_{t,1|1}^+ + (C-1)s_{t,2|1}^- + C \cdot \sum_{i=2}^M \left(s_{t,i|1}^-\right)_1} \\
&= \frac{1 - e^{-\eta T_1}}{1 + (C-1)e^{-\eta T_1} + (K-C)\exp(-\eta T_1 + \frac{1-2M}{C}\eta T_1)} \\
&\leq \frac{1 - e^{-\eta T_1}}{1 + (K-1)\exp\left(-\eta T_1 + \frac{1-2M}{C}\eta T_1\right)}.
\end{aligned}
$$

Solving the equation $\frac{1-e^{-\eta T_1'}}{1+(K-1)\exp\left(-\eta T_1'+\frac{1-2M}{C}\eta T_1'\right)} = 1 - \alpha$, we have:

$$
1 - e^{-\eta T_1'} = 1 - \alpha + (1-\alpha)(K-1)\exp\left(-\eta T_1' + \frac{1-2M}{C}\eta T_1'\right).
$$

$$
\alpha e^{\eta T_1'} - 1 = (1-\alpha)(K-1)\exp\left(\frac{1-2M}{C}\eta T_1'\right).
$$

When $\alpha e^{\eta T_1'} \gg 1$, the equality can be approximated as:

$$
\alpha e^{\eta T_1'} = (1-\alpha)(K-1)\exp\left(\frac{1-2M}{C}\eta T_1'\right). \tag{40}
$$

Taking logarithm on both sides, we have:

$$
T_1' = \frac{1}{\eta} \cdot \frac{1}{1 + \frac{2M-1}{C}} \cdot \log\left(\frac{(1-\alpha)(K-1)}{\alpha}\right).
$$

As $T_1 > T_1'$, we have:

$$
T_1 \geq \frac{1}{\eta} \cdot \frac{1}{1 + \frac{2M-1}{C}} \cdot \log\left(\frac{(1-\alpha)(K-1)}{\alpha}\right).
$$

For the second auxiliary iteration, the analysis is similar. We have:

$$
\begin{aligned}
\widehat{p}_{t,1|1} - \widehat{p}_{t,2|1} &= \frac{s_{t,1|1}^+ - s_{t,2|1}^-}{s_{t,1|1}^+ + (C-1)s_{t,2|1}^- + C \cdot \sum_{i=2}^M \left(s_{t,i|1}^-\right)_2} \\
&= \frac{1 - e^{-\eta T_2}}{1 + (C-1)e^{-\eta T_2} + (K-C)\exp\left(-\eta T_2 + \frac{1+2M}{C}\eta T_2\right)} \\
&\geq \frac{1 - e^{-\eta T_2}}{1 + (K-1)\exp\left(-\eta T_2 + \frac{1+2M}{C}\eta T_2\right)}.
\end{aligned}
$$

Solving the equation $\frac{1-e^{-\eta T_2'}}{1+(K-1)\exp\left(-\eta T_2'+\frac{1+2M}{C}\eta T_2'\right)} = 1 - \alpha$, we have:

$$
1 - e^{-\eta T_2'} = 1 - \alpha + (1-\alpha)(K-1)\exp\left(-\eta T_2' + \frac{1+2M}{C}\eta T_2'\right).
$$

$$
\alpha e^{\eta T_2'} - 1 = (1-\alpha)(K-1)\exp\left(\frac{1+2M}{C}\eta T_2'\right).
$$

When $\alpha e^{\eta T_2'} \gg 1$, the equality can be approximated as:

$$
\alpha e^{\eta T_2'} = (1-\alpha)(K-1)\exp\left(\frac{1+2M}{C}\eta T_2'\right). \tag{41}
$$

Taking the logarithm on both sides, we have:

$$T_2' = \frac{1}{\eta} \cdot \frac{1}{1 - \frac{2M+1}{C}} \cdot \log\left(\frac{(1-\alpha)(K-1)}{\alpha}\right).$$

As $T_2 < T_2'$, we have:

$$T_2 \leq \frac{1}{\eta} \cdot \frac{1}{1 - \frac{2M+1}{C}} \cdot \log\left(\frac{(1-\alpha)(K-1)}{\alpha}\right).$$

As $T_1 < T' \leq T'' < T_2$, combining the above results, we have:

$$\frac{1}{\eta} \cdot \frac{1}{1 + \frac{2M-1}{C}} \cdot \log\left(\frac{(1-\alpha)(K-1)}{\alpha}\right) \leq T' \leq T'' \leq \frac{1}{\eta} \cdot \frac{1}{1 - \frac{2M+1}{C}} \cdot \log\left(\frac{(1-\alpha)(K-1)}{\alpha}\right). \tag{42}$$

As $K \gg 1$, $C \gg 1$, and $M^2 \ll K$, we have:

$$T' = \Theta\left(\frac{1}{\eta} \log\left(\frac{(1-\alpha)K}{\alpha}\right)\right), \quad T'' = \Theta\left(\frac{1}{\eta} \log\left(\frac{(1-\alpha)K}{\alpha}\right)\right). \tag{43}$$

### D.2.2. THE EXCESS RISK

**Excess Risk at the End of Phase 1:** Recall the auxiliary iteration equation solved in the step-count analysis. According to Equations (40) and (41), as $\alpha e^{\eta T'} \gg 1$ holds, we have:

$$\alpha e^{\eta T'} = (1-\alpha)(K-1) \exp\left(\zeta \eta T'\right), \quad \text{for some } \zeta = \mathcal{O}\left(\frac{M}{C}\right). \tag{44}$$

Equivalently, we have:

$$(K-1)e^{-\eta T'} = \frac{\alpha}{1-\alpha} \exp\left(-\zeta \eta T'\right). \tag{45}$$

Using Equation (39), we obtain an upper bound for the loss at the end of Phase 1. The following lemma is used to analyze this bound.

**Lemma D.3.** *Consider a function* $f : \mathbb{R}^+ \to \mathbb{R}^+$, $f(x) = \log(1 + ae^x)$, *where $a$ is a positive constant. For any $x > 0$, we have:*

$$f(x) \leq \log(1+a) + x.$$

*Proof.* Define an auxiliary function $g : \mathbb{R}^+ \to \mathbb{R}^+$, $g(x) = f(x) - x$. The function $g(x)$ is monotonically decreasing with respect to $x$. As $g(0) = \log(1+a)$, we have $g(x) \leq g(0)$ for any $x > 0$. Thus, we have:

$$f(x) \leq \log(1+a) + x, \quad \text{for any } x > 0.$$

$\square$

The upper bound of the loss (Equation (39)) at $t = T'$ can be bounded with the above lemma as:

$$\mathcal{L}(\mathbf{W}_{T'}) \leq \log\left(1 + (K-1)\exp\left(-\eta T' + \frac{3M^2 \eta T'}{K}\right)\right) + \frac{K + 2M^2 - 3M}{K} \alpha \eta T' \tag{46}$$

$$\leq \log(1 + (K-1)e^{-\eta T'}) + \frac{3M^2 \eta T'}{K} + \frac{K + 2M^2 - 3M}{K} \alpha \eta T' \tag{47}$$

$$= \log\left(1 + \frac{\alpha}{1-\alpha} \exp\left(-\zeta \eta T'\right)\right) + \frac{3M^2 \eta T'}{K} + \frac{K + 2M^2 - 3M}{K} \alpha \eta T' \tag{48}$$

$$\leq \log\left(1 + \frac{\alpha}{1-\alpha}\right) + \alpha \eta T' + \frac{5M^2}{K} \eta T'. \tag{49}$$

In Equation (46) uses the upper bound of $\mathcal{L}$ at $t = T'$. In Equation (47), we use the lemma proved above, with $a = (K-1)e^{-\eta T'}$ and $x = \frac{3M^2 \eta T'}{K}$. We substitute $(K-1)e^{-\eta T'}$ in Equation (48) using Equation (45) and relax it to $\frac{\alpha}{1-\alpha}$.

According to Equation ([42](#)), we have:

$$\eta T' \le \left(1 + \frac{2M}{C - 2M}\right) \cdot \log\left(\frac{(1-\alpha)K}{\alpha}\right)$$
$$\eumlaut \approx \left(1 + \frac{2M^2}{K}\right) \cdot \log\left(\frac{(1-\alpha)K}{\alpha}\right), \quad \text{as } C \gg M.$$

Substituting the above inequality into the loss upper bound at $t = T'$, we have:

$$\mathcal{L}(\mathbf{W}_{T'}) \le -\log(1-\alpha) + \alpha\log\left(\frac{(1-\alpha)K}{\alpha}\right) + \mathcal{O}\left(\frac{M^2\log K}{K}\right)$$
$$\le -(1-\alpha)\log(1-\alpha) + \alpha\log\left(\frac{K}{\alpha}\right) + \mathcal{O}\left(\frac{M^2\log K}{K}\right).$$

The optimal loss is $\mathcal{L}^* = -\left(1 - \alpha + \frac{\alpha}{K}\right)\log\left(1 - \alpha + \frac{\alpha}{K}\right) - \frac{(K-1)\alpha}{K}\log\left(\frac{\alpha}{K}\right)$. As $K \gg 1$, it can be approximated as:

$$\mathcal{L}^* \approx -(1-\alpha)\log(1-\alpha) - \frac{\alpha}{K}\log(1-\alpha) - \alpha\log\left(\frac{\alpha}{K}\right) + \frac{\alpha}{K}\log\left(\frac{\alpha}{K}\right)$$
$$= -(1-\alpha)\log(1-\alpha) - \frac{\alpha}{K}\log\left(\frac{(1-\alpha)K}{\alpha}\right) - \alpha\log\left(\frac{\alpha}{K}\right). \tag{50}$$

Then the excess risk at the end of Phase 1 can be bounded as:

$$\mathcal{L}(\mathbf{W}_{T'}) - \mathcal{L}^* \le \frac{\alpha}{K}\log\left(\frac{(1-\alpha)K}{\alpha}\right) + \mathcal{O}\left(\frac{M^2\log K}{K}\right).$$

**Excess Risk during Phase 3:** After time step $T''$, all blocks of the weight matrix $\widehat{\mathbf{W}}_t$ have entered the oscillation phase. In this subsection, we analyze the upper bound of the excess risk during this oscillation and express it as a function of the learning rate $\eta$.

Since the analysis of excess risk for different sub-tasks is similar, we focus on the first sub-task as an example. The excess risk for the first sub-task can be bounded by assuming that $\widehat{p}_t$ has already reached the optimal value but moves one step further in the $t$-th iteration.

When $(\widehat{\mathbf{W}}_t)_{1,1}$ increases during the $t$-th iteration, the scores related to the first task satisfy:

$$s^+_{t,1|1} \le s^+_{t-1,1|1} \cdot \exp\left(\eta\left(1 - \frac{1}{C} + \frac{M}{C}\right)\right),$$
$$s^-_{t,1|1} = s^+_{t,1|1} \cdot \exp(-\eta),$$
$$s^-_{t,i|1} \ge s^+_{t,1|1} \cdot \exp\left(-\eta + \frac{\eta}{C} - \frac{2M\eta}{C}\right), \quad i \ne 1.$$

To simplify the notation, we define:

$$\delta_1 = \eta\left(1 - \frac{1}{C} + \frac{M}{C}\right),$$
$$\delta_i = \eta\left(-\frac{1}{C} + \frac{M}{C}\right), \quad i = 2, \ldots, C,$$
$$\delta_i = \eta\left(-\frac{M}{C}\right), \quad i = C+1, \ldots, K.$$

Assume $\widehat{p}_{t-1,1|1} = 1 - \alpha + \frac{\alpha}{K}$ and $\widehat{p}_{t-1,i|1} = \frac{\alpha}{K}$ for all $i \ne 1$. This case provides an upper bound for the excess risk. The predicted probability after the update is:

$$\widehat{p}_{t,i|1} = \widehat{p}_{t-1,i|1} \cdot \frac{e^{\delta_i}}{S} = p_{i|1} \cdot \frac{e^{\delta_i}}{S},$$

where $S = \sum_{i=1}^{K} p_{i|1} \cdot e^{\delta_i}$ is the normalization term.

The excess risk for the first sub-task at the $t$-th iteration can be bounded as:

$$
\begin{aligned}
\mathcal{L}_1(\mathbf{W}_t) - \mathcal{L}^* &\leq -\sum_{i=1}^{K} p_{i|1} \log \widehat{p}_{t,i|1} + \sum_{i=1}^{K} p_{i|1} \log p_{i|1} \\
&= \sum_{i=1}^{K} p_{i|1} \log \left( \frac{p_{i|1}}{\widehat{p}_{t,i|1}} \right) \\
&= \sum_{i=1}^{K} p_{i|1} \log \left( \frac{S}{e^{\delta_i}} \right) \\
&= \log S - \sum_{i=1}^{K} p_{i|1} \delta_i.
\end{aligned}
$$

As $\delta_i \ll 1$ for all $i \in [K]$, $p_{i|1} e^{\delta_i}$ can be approximated as follows:

$$
\begin{aligned}
p_{1|1} e^{\delta_1} &\approx \left(1 - \alpha + \frac{\alpha}{K}\right) \cdot \left(1 + \eta\left(1 + \frac{M-1}{C}\right) + \frac{\eta^2}{2}\left(1 + \frac{M-1}{C}\right)^2 + \mathcal{O}(\eta^3)\right), \\
p_{i|1} e^{\delta_i} &\approx \frac{\alpha}{K} \cdot \left(1 + \eta\left(-\frac{1}{C} + \frac{M}{C}\right) + \frac{\eta^2}{2}\left(-\frac{1}{C} + \frac{M}{C}\right)^2 + \mathcal{O}(\eta^3)\right), \quad i = 2, \ldots, C, \\
p_{i|1} e^{\delta_i} &\approx \frac{\alpha}{K} \cdot \left(1 + \eta\left(-\frac{M}{C}\right) + \frac{\eta^2}{2}\left(-\frac{M}{C}\right)^2 + \mathcal{O}(\eta^3)\right), \quad i = C+1, \ldots, K.
\end{aligned}
$$

Summing all terms, $S$ can be approximated as:

$$
\begin{aligned}
S &\approx 1 + \sum_{i=1}^{K} p_{i|1} \delta_i + \frac{\eta^2}{2}\left((1-\alpha)\left(1 + \frac{M}{C}\right)^2 + \frac{\alpha C}{K}\left(\frac{M}{C}\right)^2 + \frac{\alpha(K-C)}{K}\left(-\frac{M}{C}\right)^2\right) + \mathcal{O}(\eta^3) \\
&\approx 1 + \sum_{i=1}^{K} p_{i|1} \delta_i + \frac{\eta^2}{2}\left((1-\alpha) + \mathcal{O}\left(\frac{M}{C}\right)\right) + \mathcal{O}(\eta^3) \\
&\approx 1 + \sum_{i=1}^{K} p_{i|1} \delta_i + \frac{\eta^2}{2}\left((1-\alpha) + \mathcal{O}\left(\frac{M^2}{K}\right)\right) + \mathcal{O}(\eta^3).
\end{aligned}
$$

Using the approximation $\log(1 + x) \lesssim x$ and the assumption $M^2 \ll K$, the excess risk at the $t$-th iteration is bounded by:

$$
\begin{aligned}
\mathcal{L}_1(\mathbf{W}_t) - \mathcal{L}^* &\leq \log S - \sum_{i=1}^{K} p_{i|1} \delta_i \\
&\lesssim \frac{\eta^2}{2}\left((1-\alpha) + \mathcal{O}\left(\frac{M^2}{K}\right)\right) + \mathcal{O}(\eta^3) \\
&\lesssim \eta^2.
\end{aligned}
$$

Combining all sub-tasks, the excess risk at the $t$-th iteration can be bounded by:

$$
\mathcal{L}(\mathbf{W}_t) - \mathcal{L}^* \lesssim \eta^2.
$$

When $(\widehat{\mathbf{W}}_t)_{1,1}$ decreases at the $t$-th iteration, we can analyze similarly and obtain the same excess risk bound.

**Excess Risk during Phase 2:** As discussed before, Phase 2 is a transitional phase between Phase 1 and Phase 3, where some tasks have entered the oscillation phase while others are still in the descent phase. The excess risk during this phase can be analyzed by combining the results from Phase 1 and Phase 3. Define a set $O_t$ that contains the sub-tasks which

have entered the oscillation phase at time step $t$. For sub-tasks in $O_t$, the excess risk is bounded by $\mathcal{O}(\eta^2)$ according to the analysis of Phase 3, while for sub-tasks not in $O_t$, the excess risk follows the bound derived in Phase 1. Thus, the overall excess risk during Phase 2 can be expressed as:

$$\mathcal{L}(\mathbf{W}_t) - \mathcal{L}^* = \sum_{i \in O_t} p_i \eta^2 + \sum_{i \in O_t^c} p_i \cdot \left( K \exp\left( -\eta t + \frac{3M^2 \eta t}{K} \right) + \frac{K + 2M^2 - 3M}{K} \alpha \eta t - \mathcal{L}^* \right), T' \le t < T''.$$

**The Excess Risk of Muon in the Noisy Case:**
Combining the excess risk bounds from all three phases, the excess risk of the $j$-th sub-task can be summarized as:

$$\mathcal{L}_j(\mathbf{W}_t) - \mathcal{L}_j^* \le \begin{cases} K \exp\left( -\eta t + \frac{3M^2 \eta t}{K} \right) + \frac{K + 2M^2 - 3M}{K} \alpha \eta t - \mathcal{L}_j^*, & t < T_j^*. \\ \eta^2, & t \ge T_j^*, \end{cases}$$

where $T_j^*$ is the time step when the $j$-th sub-task enters the oscillation phase. We have $T' \le T_j^* \le T''$, for all $j \in [K]$. As we mentioned in Equation 43, $T' = \Theta\left( \frac{1}{\eta} \log\left( \frac{(1-\alpha)K}{\alpha} \right) \right)$ and $T'' = \Theta\left( \frac{1}{\eta} \log\left( \frac{(1-\alpha)K}{\alpha} \right) \right)$, which means all sub-tasks enter the oscillation phase within a similar time frame.

Combining the excess risk bounds from all three phases, the excess risk of Muon in the noisy case can be summarized as:

$$\mathcal{L}(\mathbf{W}_t) - \mathcal{L}^* \le \begin{cases} K \exp\left( -\eta t + \frac{3M^2 \eta t}{K} \right) + \frac{K + 2M^2 - 3M}{K} \alpha \eta t - \mathcal{L}^*, & t < T', \\ \sum_{i \in O_t} p_i \eta^2 + \sum_{i \in O_t^c} p_i \cdot \left( K \exp\left( -\eta t + \frac{3M^2 \eta t}{K} \right) + \frac{K + 2M^2 - 3M}{K} \alpha \eta t - \mathcal{L}^* \right), & T' \le t < T'', \\ \eta^2, & t \ge T'', \end{cases} \tag{51}$$

where $O_t$ is the set of tasks that have entered the oscillation phase at time step $t$.

### D.3. Comparison Between GD and Muon in the Noisy Case

According to the discussion above, Muon achieves an excess risk of $\mathcal{O}\left( \frac{M^2 \log K}{K} \right)$ at the end of Phase 1. With Equation (43), the number of steps required is

$$T' = \Theta\left( \frac{1}{\eta} \log\left( \frac{(1-\alpha)K}{\alpha} \right) \right).$$

We analyze how long GD needs to achieve the same excess risk. According to Equation (34), the excess risk of GD in the noisy case satisfies:

$$\mathcal{L}(\mathbf{W}_T) - \mathcal{L}^* \ge \left( 1 - \alpha + \frac{\alpha}{K} \right) \cdot \left( \alpha - \frac{\alpha}{K} \right) \frac{(\Delta^*)^2}{2} \sum_{i=1}^{K} p_i e^{-\eta p_i T}$$

$$\ge \left( 1 - \alpha + \frac{\alpha}{K} \right) \cdot \left( \alpha - \frac{\alpha}{K} \right) \frac{(\Delta^*)^2}{2} \cdot e^{-\eta p_1 T}.$$

To achieve an excess risk of $\mathcal{O}\left( \frac{M^2 \log K}{K} \right)$, $T$ must satisfy:

$$T \gtrsim \frac{1}{\eta p_1} \log\left( \frac{K \log K}{M^2} \right).$$

As $p_1 \le \frac{1}{C}$, we have:

$$T \gtrsim \frac{C}{\eta} \log\left( \frac{K \log K}{M^2} \right)$$

$$= \Omega\left( \frac{C}{\eta} \log\left( \frac{K}{M^2} \right) \right).$$

Therefore, when the two optimizers use the same learning rate $\eta$, to achieve an excess risk of $\mathcal{O}\left(\frac{M^2 \log K}{K}\right)$, the number of steps required by GD is at least $C$ times that of Muon, when $M$ is treated as a relatively fixed constant. When the group size $C$ is large, the gap between the two optimizers extends further.

## E. Scaling Law Analysis

In this section, we introduce additional assumptions regarding the data distribution to analyze the scaling laws of Muon and Gradient Descent (GD) in the presence of noise. We consider an asymptotic regime where the number of groups $M$ goes to infinity, and the total number of classes $K$ approaches infinity consequently.

Let $\tilde{p}_i$ denote the aggregate probability of the $i$-th group, defined as:

$$\tilde{p}_i = \sum_{j=(i-1)C+1}^{iC} p_j.$$

The full assumptions on the data distribution are as follows:

1. **Normalization:** The total probability is normalized.

$$\sum_{i=1}^{M} \tilde{p}_i = 1.$$

2. **Power-law Decay:** The group probabilities follow a power-law distribution.

$$\tilde{p}_i \propto i^{-\beta}, \quad \beta > 1.$$

3. **Intra-group Uniformity:** Within each group, the classes are uniformly distributed:

$$p_j = \frac{\tilde{p}_i}{C}, \quad j = (i-1)C+1, \ldots, iC, \quad \forall i \in [M].$$

The restriction of $T$ is :
$$cM^{\beta} \leq T \leq M^{\beta}, \text{ where } c \text{ is a constant satisfying } 0 < c < 1.$$

We also introduce an additional assumption on the relationship between $M$ and $K$:

$$(\log K)^{\frac{1}{\beta}} \leq M \ll K^{\frac{1}{2}}.$$

### E.1. Analysis of GD

The auxiliary functions used in the analysis of GD are presented in the following lemmas.

**Lemma E.1.** *Let $\beta > 1$ be a constant. For integer $M > 0$, we have:*

$$\sum_{i=1}^{M} i^{-\beta} \cdot e^{-i^{-\beta}t} \gtrsim \frac{1}{t^{1-\frac{1}{\beta}}}, \quad for\ 1 \ll t \leq M^{\beta}.$$

*Proof of Lemma E.1.* Define a function $f : \mathbb{R} \to \mathbb{R}$, $f(x) = x^{-\beta}e^{-x^{-\beta}t}$.
Calculate the derivative of $f(x)$:

$$f'(x) = -\beta x^{-\beta-1}e^{-x^{-\beta}t} + x^{-\beta} \cdot (-\beta x^{-\beta-1}t)e^{-x^{-\beta}t}$$
$$= -\beta x^{-\beta-1}e^{-x^{-\beta}t}(1 - x^{-\beta}t).$$

Let $x^*$ be the maximizer of the objective function. We have $x^* = t^{\frac{1}{\beta}}$. As $t \leq M^\beta$, we have $x^* \leq M$. Thus, $f(x)$ is increasing in the interval $[0, x^*]$, and decreasing in the interval $[x^*, M]$. Then we have:

$$\sum_{i=1}^{M} i^{-\beta} \cdot e^{-i^{-\beta}t} \geq \int_0^{\lfloor x^* \rfloor} x^{-\beta} \cdot e^{-x^{-\beta}t}\,\mathrm{d}x + \int_{\lfloor x^* \rfloor+1}^{M+1} x^{-\beta} \cdot e^{-x^{-\beta}t}\,\mathrm{d}x$$

$$\geq \underbrace{\int_0^\infty x^{-\beta} \cdot e^{-x^{-\beta}t}\,\mathrm{d}x}_{I_\infty} - f(x^*) - \underbrace{\int_{M+1}^\infty x^{-\beta} \cdot e^{-x^{-\beta}t}\,\mathrm{d}x}_{I_{\text{tail}}}.$$

The analysis of $I_\infty$ is:

$$I_\infty = \int_0^\infty x^{-\beta} \cdot e^{-x^{-\beta}t}\,\mathrm{d}x$$

$$= \frac{1}{\beta t} \int_0^\infty x^{\beta+1} \cdot x^{-\beta} \cdot e^{-u}\,\mathrm{d}u, \quad u = x^{-\beta}t$$

$$= \frac{1}{\beta t} \int_0^\infty u^{-\frac{\beta+1}{\beta}} e^{-u}\,\mathrm{d}u = \frac{\Gamma(1 - \frac{1}{\beta})}{\beta} \cdot \frac{1}{t^{1-\frac{1}{\beta}}},$$

where $\Gamma(\cdot)$ is the Gamma function.

The analysis of $f(x^*)$ is:

$$f(x^*) = (x^*)^{-\beta} \cdot e^{-(x^*)^{-\beta}t} = t^{-1} \cdot e^{-1} \ll \frac{1}{t^{1-\frac{1}{\beta}}}, \quad \text{as } t \gg 1.$$

The analysis of $I_{\text{tail}}$ is:

$$I_{\text{tail}} = \int_{M+1}^\infty x^{-\beta} \cdot e^{-x^{-\beta}t}\,\mathrm{d}x \leq \int_{M+1}^\infty x^{-\beta}\,\mathrm{d}x = \frac{(M+1)^{1-\beta}}{\beta - 1} \ll \frac{1}{t^{1-\frac{1}{\beta}}}, \quad \text{as } t \leq M^\beta.$$

Combining the above results, we have:

$$\sum_{i=1}^{M} i^{-\beta} \cdot e^{-i^{-\beta}t} \gtrsim \frac{1}{t^{1-\frac{1}{\beta}}}, \quad \text{for } 1 \ll t \leq M^\beta.$$

$\square$

**Lemma E.2.** *Consider a function $g : \mathbb{R} \to \mathbb{R}$, $g(x) = Ax + \log(1 + A(e^{-x} - 1))$, where $0 < A < 1$. Denote $\frac{g(x)}{x^2}$ as $f(x)$. $B$ is a nonnegative constant and $B > \log\left(\frac{A}{1-A}\right)$. The following inequality holds:*

$$g(x) \geq \min\{\lim_{x\to 0} f(x), f(B)\} \cdot x^2, \quad \text{for } 0 < x \leq B.$$

*Proof of Lemma E.2.* First, we prove the limits $\lim_{x\to 0} f(x)$ exists. Using L'Hôpital's rule, we have:

$$\lim_{x\to 0} f(x) = \lim_{x\to 0} \frac{g(x)}{x^2} = \lim_{x\to 0} \frac{g'(x)}{2x} = \lim_{x\to 0} \frac{A - \frac{Ae^{-x}}{1+A(e^{-x}-1)}}{2x}$$

$$= \lim_{x\to 0} \frac{A(1-A)(1-e^{-x})}{2x(1+A(e^{-x}-1))} = \frac{A(1-A)}{2}.$$

Then, we analyze the monotonicity of $f(x)$. Calculate the derivative of $f(x)$:

$$f'(x) = \frac{g'(x)x^2 - 2xg(x)}{x^4}.$$

Define an auxiliary function $h : (0, B] \to \mathbb{R}$, $h(x) = xg'(x) - 2g(x)$. The sign of $h(x)$ is identical with that of $f'(x)$ when $x > 0$. To analyze $h(x)$, we examine the successive derivatives of it:

$$h'(x) = g'(x) + xg''(x) - 2g'(x) = xg''(x) - g'(x),$$
$$h''(x) = g''(x) + xg'''(x) - g''(x) = xg'''(x).$$

Let $g'(x)$, $g''(x)$, and $g'''(x)$ denote the first, second, and third-order derivatives of $g$, respectively. We calculate them as follows:

$$g'(x) = A - \frac{Ae^{-x}}{1 + A(e^{-x} - 1)} = \frac{A(1 - A)(1 - e^{-x})}{1 + A(e^{-x} - 1)}.$$

$$g''(x) = \frac{Ae^{-x}(1 + A(e^{-x} - 1)) - Ae^{-x} \cdot Ae^{-x}}{(1 + A(e^{-x} - 1))^2} = \frac{A(1 - A)e^{-x}}{(1 + A(e^{-x} - 1))^2}.$$

$$g'''(x) = \frac{-e^{-x}(1 + A(e^{-x} - 1))^2 + e^{-x} \cdot 2Ae^{-x}(1 + A(e^{-x} - 1))}{(1 + A(e^{-x} - 1))^4} \cdot A(1 - A)$$
$$= \frac{e^{-x}(1 + A(e^{-x} - 1))}{(1 + A(e^{-x} - 1))^4} \cdot (Ae^{-x} - 1 + A) \cdot A(1 - A).$$

Define another auxiliary function $q : (0, B] \to \mathbb{R}$, $q(x) = Ae^{-x} - 1 + A$. As $e^{-x} - 1 > -1$ and $0 < A < 1$, we have $1 + A(e^{-x} - 1) > 0$ for any $x \in (0, B]$. Thus, the sign of $g'''(x)$ is identical with that of $q(x)$.

The analysis of $q(x)$ is as follows.

- If $A \leq \frac{1}{2}$, we have:
$$q(x) < A - 1 + A = 2A - 1 \leq 0, \quad \text{for any } x \in (0, B].$$

- If $A > \frac{1}{2}$, let $x_1 = -\log\left(\frac{1-A}{A}\right)$ be the maximizer of $g(x)$. We have $q(x)$ is positive in the interval $(0, x_1)$ and negative in the interval $(x_1, B]$.

Before discussing $h(x)$ by cases, we first evaluate some preliminary limits. The limits when $x$ approaches $0$ are as follows:

$$\lim_{x \to 0} g''(x) = A(1 - A), \quad \lim_{x \to 0} g'(x) = 0, \quad \lim_{x \to 0} g(x) = 0.$$
$$\begin{cases} \lim_{x \to 0} h'(x) = \lim_{x \to 0} xg''(x) - g'(x) = 0, \\ \lim_{x \to 0} h(x) = \lim_{x \to 0} xg'(x) - 2g(x) = 0. \end{cases}$$

The limits when $x$ approaches $\infty$ are as follows:

$$\lim_{x \to \infty} xg''(x) = \lim_{x \to \infty} A(1 - A)\frac{xe^{-x}}{(1 + A(e^{-x} - 1))^2} = 0,$$
$$\lim_{x \to \infty} g'(x) = \lim_{x \to \infty} (A - \frac{Ae^{-x}}{1 + A(e^{-x} - 1)}) = A,$$
$$\lim_{x \to \infty} h'(x) = \lim_{x \to \infty} xg''(x) - g'(x) = -A,$$
$$\lim_{x \to \infty} h(x) = \lim_{x \to \infty} (xg'(x) - 2g(x))$$
$$= \lim_{x \to \infty} (Ax - 2(Ax + \log(1 + A(e^{-x} - 1))))$$
$$= \lim_{x \to \infty} -Ax = -\infty.$$

Now we discuss $h(x)$ by cases:

- **Case 1:** If $A \leq \frac{1}{2}$, $g'''(x) < 0$ for any $x \in (0, B]$.

Thus, $h''(x) < 0$ for any $x \in (0, B]$. $h'(x)$ is monotonically decreasing in $(0, B]$. As $\lim_{x \to 0} h'(x) = 0$, we have $h'(x) < 0$ for any $x \in (0, B]$. Thus, $h(x)$ is monotonically decreasing in $(0, B]$. As $\lim_{x \to 0} h(x) = 0$, we have $h(x) < 0$ for any $x \in (0, B]$.

Therefore, $f'(x) < 0$ for any $x \in (0, B]$. $f(x)$ is monotonically decreasing in $(0, B]$. We have:

$$f(x) \geq f(B), \quad \text{for } 0 < x \leq B.$$

$$g(x) \geq f(B) \cdot x^2, \quad \text{for } 0 < x \leq B.$$

- **Case 2:** If $A > \frac{1}{2}$, let $x_1 = \log\left(\frac{A}{1-A}\right)$. $g'''(x) > 0$ for any $x \in (0, x_1)$ and $g'''(x) < 0$ for any $x \in (x_1, B]$.

  Thus, $h''(x) > 0$ for any $x \in (0, x_1)$ and $h''(x) < 0$ for any $x \in (x_1, B]$. $h'(x)$ is monotonically increasing in $(0, x_1)$ and decreasing in $(x_1, B]$.

  As $\lim_{x \to 0} h'(x) = 0$, $h'(x_1) > 0$. As $\lim_{x \to \infty} h'(x) = -A < 0$, there exists a point $x_2 \in (x_1, \infty]$ such that $h'(x_2) = 0$. Thus, $h(x)$ is monotonically increasing in $(0, x_2)$ and decreasing in $(x_2, B]$.

  As $\lim_{x \to 0} h(x) = 0$, we have $h(x) > 0$ for any $x \in (0, x_2)$. As $\lim_{x \to \infty} h(x) = -\infty$, there exists a point $x_3 \in (x_2, \infty)$ such that $h(x_3) = 0$. Thus, $h(x) > 0$ for any $x \in (0, x_3)$ and $h(x) < 0$ for any $x \in (x_3, \infty)$.

  Therefore, $f'(x) > 0$ for any $x \in (0, x_3)$ and $f'(x) < 0$ for any $x \in (x_3, \infty)$. $f(x)$ is monotonically increasing in $(0, x_3)$ and decreasing in $(x_3, \infty)$. We have:

$$f(x) \geq \min\{\lim_{x \to 0} f(x), f(B)\}, \quad \text{for } 0 < x \leq B.$$

$$g(x) \geq \min\{\lim_{x \to 0} f(x), f(B)\} \cdot x^2, \quad \text{for } 0 < x \leq B.$$

Combining the above two cases, we have:

$$g(x) \geq \min\{\lim_{x \to 0} f(x), f(B)\} \cdot x^2, \quad \text{for } 0 < x \leq B.$$

$\square$

Total proof of Theorem 5.7 is as follows:

*Proof of Theorem 5.7.* Back to equation (33), in the previous estimation, we use $\eta p_i T$ goes to infinity for all $i \in [K]$. When the group number $M$ is a finite constant and $T$ goes to infinity, the claim holds.

However, when the group number $M$ goes to infinity and speed $T$ goes to infinity need to satisfies the restriction $T \leq M^\beta$, the claim $\eta p_i T$ goes to infinity does not hold for all $i \in [M]$. In particular, when $i = M$, $\eta p_M T$ does not go to infinity but a constant multiple of $\eta$. Therefore, a new way to estimate the excess risk is needed under the new restriction.

Equivalent to the analysis about GD in noisy case, a large learning rate $\eta$ is preferred to achieve a smaller excess risk. When analyzing the lower bound of excess risk, we need to consider a relatively large learning rate $\eta$. However, according to Proposition 5.4, under the stability condition, $\eta$ cannot be chosen arbitrarily large, that $\eta p_1$ needs to be upper bounded by a constant. To simplify the analysis, we set $\eta p_1 = 1$.

Thus, the following equation holds for all pairs:

$$\eta p_i = j^{-\beta}, \quad i = (j-1)C + 1, (j-1)C + 2, \cdots, jC, \quad \forall j \in [M].$$

We can estimate equation (33) with Lemma E.2, substituting $A = 1 - \alpha + \frac{\alpha}{K}$ and $x = e^{-\frac{1}{2}\eta p_i T}\Delta^*$. As $cM^\beta \leq T \leq M^\beta$, we have $c \leq \eta p_M T \leq 1$. Thus $e^{-\frac{1}{2}\eta p_i T}$ satisfies $0 < e^{-\frac{1}{2}\eta p_i T} \leq e^{-\frac{c}{2}}$ for all $i \in [M]$. Denote $f(x)$ as in Lemma E.2. To simplify the notation, we denote $e^{-\frac{c}{2}}$ as $c'$

$$f(0) = \frac{(1 - \alpha + \frac{\alpha}{K})(\alpha - \frac{\alpha}{K})}{2},$$

$$f(c'\Delta^*) = \frac{(1 - \alpha + \frac{\alpha}{K})c'\Delta^* + \log(1 + (1 - \alpha + \frac{\alpha}{K})(e^{-c'\Delta^*} - 1))}{(c'\Delta^*)^2}$$

$$\approx \frac{(1 - \alpha + \frac{\alpha}{K})c'\Delta^*}{(c'\Delta^*)^2} = \frac{(1 - \alpha + \frac{\alpha}{K})}{c'\Delta^*}.$$

As $\Delta^* \asymp \log(K)$ and $K$ goes to infinity, we have $f(0) > f(c'\Delta^*)$.

Using Lemma E.2, we have:

$$\mathcal{L}_i(\mathbf{W}_T) - \mathcal{L}_i^* \geq \frac{(1 - \alpha + \frac{\alpha}{K})}{c'\Delta^*} \cdot \left(e^{-\frac{1}{2}\eta p_i T}\Delta^*\right)^2$$

$$= \left(1 - \alpha + \frac{\alpha}{K}\right)\frac{\Delta^*}{c'}e^{-\eta p_i T}.$$

Combining the excess risk related to all tasks, we have:

$$\mathcal{L}(\mathbf{W}_T) - \mathcal{L}^* \geq \left(1 - \alpha + \frac{\alpha}{K}\right)\frac{\Delta^*}{c'}\sum_{i=1}^{K} p_i e^{-\eta p_i T}. \tag{52}$$

Then the loss excess risk can be estimated as:

$$\mathcal{L}(\mathbf{W}_T) - \mathcal{L}^* \geq \left(1 - \alpha + \frac{\alpha}{K}\right)\frac{\Delta^*}{c'}\sum_{j=1}^{M}\sum_{i=(j-1)C+1}^{jC}\frac{\tilde{p}_j}{C}e^{-j^{-\beta}T}$$

$$= \left(1 - \alpha + \frac{\alpha}{K}\right)\frac{\Delta^*}{c'}\sum_{j=1}^{M}\tilde{p}_j e^{-j^{-\beta}T}$$

$$= \left(1 - \alpha + \frac{\alpha}{K}\right)\frac{\Delta^*}{c'}\cdot\frac{1}{\sum_{i=1}^{M}i^{-\beta}}\sum_{j=1}^{M}j^{-\beta}e^{-j^{-\beta}T}.$$

Using Lemma E.1, we have:

$$\mathcal{L}(\mathbf{W}_T) - \mathcal{L}^* \gtrsim \left(1 - \alpha + \frac{\alpha}{K}\right)\frac{\Delta^*}{c'}\cdot\frac{1}{\sum_{i=1}^{M}i^{-\beta}}\cdot\frac{1}{T^{1-\frac{1}{\beta}}}.$$

As

$$\sum_{i=1}^{M}i^{-\beta} < 1 + \int_1^\infty x^{-\beta}\,\mathrm{d}x < 1 + \frac{1}{\beta - 1},$$

we have:

$$\mathcal{L}(\mathbf{W}_T) - \mathcal{L}^* \gtrsim \left(1 - \alpha + \frac{\alpha}{K}\right)\frac{\Delta^*}{c'(1 + \frac{1}{\beta-1})}\cdot\frac{1}{T^{1-\frac{1}{\beta}}}$$

$$\asymp \log K \cdot \frac{1}{T^{1-\frac{1}{\beta}}}.$$

The last inequality uses the fact that $\Delta^* = \log\left(K\cdot\frac{1-\alpha+\frac{\alpha}{K}}{\alpha}\right)$. $\qquad\square$

### E.2. Analysis of Muon

*Proof of Theorem 5.8.* According to Equation (51), the excess risk of Muon after $T$ iterations can be bounded. For a fixed $T$, when $\eta$ is not sufficiently large, leading to $T < T'$, the trajectory is still in Phase 1 at the $T$-th iteration. In this case, increase $\eta$ will decrease the excess risk.

When $\eta$ is large enough, leading the trajectory to enter Phase 2 at the $T$-th iteration, the excess risk is bounded by $\mathcal{O}(\eta^2)$. In this case, decrease $\eta$ will decrease the excess risk.

Thus, there is a trade-off in choosing $\eta$ to minimize the excess risk after $T$ iterations. The $\eta$ can be chosen as:

$$\eta = \frac{1}{(1 - \frac{2M+1}{C})T}\log\left(\frac{(1-\alpha)K}{\alpha}\right).$$

According to Equation (42), the above choice of $\eta$ ensures $T \geq T''$. With Equation (51), the excess risk after $T$ iterations can be bounded as:

$$\mathcal{L}(\mathbf{W}_T) - \mathcal{L}^* \lesssim \eta^2 \lesssim \frac{1}{T^2} \left( \log \left( \frac{(1-\alpha)K}{\alpha} \right) \right)^2 \lesssim \frac{(\log K)^2}{T^2}. \tag{53}$$

$\square$

### E.3. Comparison Between Excess Risk of GD and Muon

Equation (52) and Equation (53) provide the excess risk bounds for GD and Muon, respectively. We have:

$$\begin{cases} \mathcal{L}^{\mathrm{GD}}(\mathbf{W}_T) - \mathcal{L}^* \gtrsim \frac{\log K}{T^{1-\frac{1}{\beta}}}, \\ \mathcal{L}^{\mathrm{Muon}}(\mathbf{W}_T) - \mathcal{L}^* \lesssim \frac{(\log K)^2}{T^2}. \end{cases}$$

Regardless of the differences in logarithmic terms, the polynomial decay rates of excess risk for GD and Muon are $T^{-(1-\frac{1}{\beta})}$ and $T^{-2}$, respectively. As $T \geq cM^\beta \gg \log K$, we have:

$$\mathcal{L}^{\mathrm{GD}}(\mathbf{W}_T) - \mathcal{L}^* \gg \mathcal{L}^{\mathrm{Muon}}(\mathbf{W}_T) - \mathcal{L}^*.$$

Muon provides a speedup in the polynomial decay rate from $1 - \frac{1}{\beta}$ to $2$ compared to GD.

## F. Task-Representation Aligned SignGD (TRA-SignGD)

To highlight the performance discrepancies among these optimizers under the specific structural conditions discussed in this paper, we propose an idealized optimizer called Task-Representation Aligned SignGD (TRA-SignGD).

### F.1. Definition

The Task-Representation Aligned SignGD (TRA-SignGD) is defined under the idealized assumption that the embedding matrices $\mathbf{E}$ and $\widetilde{\mathbf{E}}$ are known. It can be viewed as a variant of SignGD, where the sign operation is performed in the task-representation space. The update rule is:

$$\mathbf{W}_{t+1} = \mathbf{W}_t - \eta \, \widetilde{\mathbf{E}} \, \mathrm{sgn} \left( \widetilde{\mathbf{E}}^\top \nabla_{\mathbf{W}_t} \mathcal{L}(\mathbf{W}_t) \mathbf{E} \right) \mathbf{E}^\top.$$

Denote $\mathbf{G}_t = \widetilde{\mathbf{E}}^\top \nabla_{\mathbf{W}_t} \mathcal{L}(\mathbf{W}_t) \mathbf{E}$ as the gradient in the task-representation space, and $\widehat{\mathbf{W}}_t = \widetilde{\mathbf{E}}^\top \mathbf{W}_t \mathbf{E}$ as the weight matrix in the task-representation space. The update rule of TRA-SignGD can be rewritten as:

$$\widehat{\mathbf{W}}_{t+1} = \widehat{\mathbf{W}}_t - \eta \, \mathrm{sgn}(\mathbf{G}_t). \tag{54}$$

### F.2. Analysis in Noiseless Case

Assume the weight matrix is initialized as $\mathbf{W}_0 = \mathbf{0}$ and optimized via TRA-SignGD, consider the update dynamics of the weight matrix $\widehat{\mathbf{W}}_t$ in optimization. According to the update rule of TRA-SignGD (54) and the gradient of loss (11), the update of the weight matrix $\widehat{\mathbf{W}}_t$ is:

$$\widehat{\mathbf{W}}_{t+1} = \widehat{\mathbf{W}}_t - \eta \, \mathrm{sgn}(\mathbf{G}_t) = \widehat{\mathbf{W}}_t + \eta \left( 2\mathbf{I}_K - \mathbf{J}_K \right).$$

In a classification task with softmax, the update matrix $(2\mathbf{I}_K - \mathbf{J}_K)$ is equivalent to $2\mathbf{I}_K$.

As $\eta = 1$, we have:

$$\widehat{\mathbf{W}}_{t+1} = \widehat{\mathbf{W}}_t + 2\mathbf{I}_K - \mathbf{J}_K$$

The loss at time step $T$ can be expressed as:

$$
\begin{aligned}
\mathcal{L}(\mathbf{W}_T) &= \sum_{i=1}^{K} p_i \mathcal{L}_i(\mathbf{W}_T) \\
&= -\sum_{i=1}^{K} p_i \log \left( \frac{\exp((\widehat{\mathbf{W}}_T)_{i,i})}{\exp((\widehat{\mathbf{W}}_T)_{i,i}) + \sum_{l \neq i} \exp((\widehat{\mathbf{W}}_T)_{l,i})} \right) \\
&= -\sum_{i=1}^{K} p_i \log \left( \frac{e^T}{e^T + (K-1)e^{-T}} \right) \\
&= \sum_{i=1}^{K} p_i \log \left( 1 + (K-1)e^{-2T} \right) \\
&\approx K \cdot e^{-2T}, \quad \text{as } T \to \infty, K \text{ is large enough.}
\end{aligned}
$$

Back to the update of Muon (Proposition 6.1), the update dynamics of Muon are essentially equivalent to those of TRA-SignGD, with the only discrepancy being a factor of 2 in the effective learning rate. Despite having no prior knowledge of the embedding structure of queries and answers, Muon achieves a convergence rate comparable to the idealized TRA-SignGD, which is a significant acceleration in loss decay compared to GD.

### F.3. Analysis in Noisy Case

**Theorem F.1.** *Under TRA-SignGD, the excess risk for the $j$-th knowledge learning sub-task satisfies:*

$$
\mathcal{L}_j^{\text{TRA-SignGD}}(t) - \mathcal{L}_j^* \lesssim \begin{cases} Ke^{-2\eta t} + 2\alpha\eta t - \mathcal{L}_j^*, & t \leq T_j^*; \\ \eta^2, & t > T_j^*, \end{cases}
$$

*where $T_j^* = \Theta\left(\frac{\log K}{\eta}\right)$ is the time step when the $j$-th subtask enters the oscillation phase and $\mathcal{L}_j^*$ is the optimal loss for the $j$-th knowledge learning sub-task, which is equivalent to $\mathcal{L}^*$. We have $T_j$ is the same for all $j \in [K]$.*

*The total excess risk satisfies:*

$$
\mathcal{L}^{\text{TRA-SignGD}}(t) - \mathcal{L}^* \lesssim \begin{cases} Ke^{-2\eta t} + 2\alpha\eta t - \mathcal{L}^*, & t \leq T'; \\ \eta^2, & t > T', \end{cases}
$$

*where $T' = \Theta\left(\frac{\log K}{\eta}\right)$ is the time step when all tasks enter the oscillation phase, which is the same as $T_j^*$ for all $j \in [K]$.*

#### F.3.1. DYNAMICS

According to the gradient in noisy case (Equation (13)), we have:

$$
-\mathbf{G}_t = (\mathbf{P}' - \widehat{\mathbf{P}'}_t).
$$

Operate the sign operation on $\mathbf{G}_t$, we have:

$$
-\operatorname{sgn}(\mathbf{G}_t)_{i,j} = \operatorname{sgn}(p_{i|j} - \widehat{p}_{t,i|j}) = \begin{cases} 1, & p_{i|j} > \widehat{p}_{t,i|j} \\ -1, & p_{i|j} < \widehat{p}_{t,i|j} \\ 0, & p_{i|j} = \widehat{p}_{t,i|j} \end{cases}.
$$

At the beginning of optimization, $\widehat{p}_{t,i|j} = \frac{1}{K}$ for all $i, j \in [K]$. As $p_{i|i} = 1 - \alpha + \frac{\alpha}{K} > \frac{1}{K}$ and $p_{i|j} = \frac{\alpha}{K} < \frac{1}{K}$ for all $i \neq j$, the update of the weight matrix $\widehat{\mathbf{W}}_t$ at the beginning of optimization is:

$$
\widehat{\mathbf{W}}_{t+1} = \widehat{\mathbf{W}}_t + \eta \left( 2\mathbf{I}_K - \mathbf{J}_{K,K} \right).
$$

Under a classification task with softmax, the update matrix $(2\mathbf{I}_K - \mathbf{J}_{K,K})$ is equivalent to $2\mathbf{I}_K$. $\widehat{\mathbf{W}}_{t+1} - \widehat{\mathbf{W}}_t$ maintains $2\mathbf{I}_K - \mathbf{J}_{K,K}$ until $\widehat{p}_{t,i|i}$ reaches $1 - \alpha + \frac{\alpha}{K}$ for some $i$ or $\widehat{p}_{t,i|j}$ reaches $\frac{\alpha}{K}$ for some $i \neq j$.

Focusing on the first task, the predicted probabilities of all wrong classes decrease at the same speed, as the update of $\widehat{\mathbf{W}}_t$ is symmetric for all wrong classes. Therefore, $\widehat{p}_{t,1|1} > 1 - \alpha + \frac{\alpha}{K}$ and $\widehat{p}_{t,i|1} < \frac{\alpha}{K}$ for all $i \neq 1$ will be achieved simultaneously. The update of the first column of $\widehat{\mathbf{W}}_t$ transforms from $(1, -1, \ldots, -1)^\top$ to $(-1, 1, \ldots, 1)^\top$ at some step. Notice that the update of all columns of $\widehat{\mathbf{W}}_t$ is the same; thus, the flip times for all columns are identical.

Similar to the analysis in Muon, after the flip time, the predicted probability oscillates around the true distribution with a small margin. The trajectory of TRA-SignGD can be divided into two phases. Denoting the flip time as $T'$, which is the end of Phase 1, we have:

- For $t \leq T'$, the update of $\widehat{\mathbf{W}}_t$ is:
$$\widehat{\mathbf{W}}_{t+1} = \widehat{\mathbf{W}}_t + \eta \left(2\mathbf{I}_K - \mathbf{J}_{K,K}\right).$$

- For $t > T'$, the update of $\widehat{\mathbf{W}}_t$ oscillates between:
$$\widehat{\mathbf{W}}_{t+1} = \widehat{\mathbf{W}}_t - \eta \left(2\mathbf{I}_K - \mathbf{J}_{K,K}\right)$$

and
$$\widehat{\mathbf{W}}_{t+1} = \widehat{\mathbf{W}}_t + \eta \left(2\mathbf{I}_K - \mathbf{J}_{K,K}\right).$$

### F.3.2. NUMBER OF STEPS TO ENTER OSCILLATION PHASE

As all sub-tasks enter the oscillation phase simultaneously, we analyze the number of steps to enter the oscillation phase $T'$ via the first sub-task.

At time step $t \leq T'$, the predicted probability of the first task for the correct class is:

$$\begin{aligned}
\widehat{p}_{1,1} &= \frac{\exp((\widehat{\mathbf{W}}_t)_{1,1})}{\sum_{l=1}^K \exp((\widehat{\mathbf{W}}_t)_{l,1})} \\
&= \frac{\exp(\eta t)}{\exp(\eta t) + (K-1)\exp(-\eta t)}.
\end{aligned}$$

The condition to enter the oscillation phase is $\widehat{p}_{1,1} \geq 1 - \alpha + \frac{\alpha}{K}$. Thus, we have:

$$\frac{\exp(\eta T')}{\exp(\eta T') + (K-1)\exp(-\eta T')} \geq 1 - \alpha + \frac{\alpha}{K},$$

The number of steps to enter the oscillation phase $T'$ satisfies:

$$T' \approx \frac{1}{2\eta} \log\left(\frac{(1-\alpha)K}{\alpha}\right).$$

Compared to Muon, we have $T_{\text{Muon}}$ is roughly a factor of two larger than $T_{\text{TRA-SignGD}}$.

### F.3.3. THE EXCESS RISK

**Risk at $T'$ Steps:** The loss at time step $t \leq T'$ is:

$$\begin{aligned}
\mathcal{L}(\mathbf{W}_t) &= \sum_{i=1}^K p_i \mathcal{L}_i(\mathbf{W}_t) \\
&= -\left(1 - \alpha + \frac{\alpha}{K}\right) \log\left(\frac{e^{\eta t}}{e^{\eta t} + (K-1)e^{-\eta t}}\right) - \frac{(K-1)\alpha}{K} \log\left(\frac{e^{-\eta t}}{e^{\eta t} + (K-1)e^{-\eta t}}\right) \\
&= -\log\left(\frac{e^{2\eta t}}{e^{2\eta t} + (K-1)}\right) + \frac{(K-1)\alpha}{K} \cdot 2\eta t \\
&\leq (K-1)e^{-2\eta t} + 2\alpha\eta t.
\end{aligned} \tag{55}$$

At time step $t = T'$, the loss can be bounded as:

$$\mathcal{L}(\mathbf{W}_{T'}) \leq -\log(1-\alpha) + \frac{(K-1)\alpha}{K} \log\left(\frac{(1-\alpha)K}{\alpha}\right).$$

According to Equation (50), the optimal loss $\mathcal{L}^*$ can be approximated as:

$$\mathcal{L}^* = -(1-\alpha)\log(1-\alpha) - \frac{\alpha}{K}\log\left(\frac{(1-\alpha)K}{\alpha}\right) - \alpha\log\left(\frac{\alpha}{K}\right).$$

Thus, the excess risk at time step $T'$ can be bounded as:

$$\mathcal{L}(\mathbf{W}_{T'}) - \mathcal{L}^* = \mathcal{O}\left(\frac{1}{K}\right).$$

**Excess Risk After Entering Oscillation Phase:** After entering the oscillation phase, the predicted probabilities oscillate around the true distribution. Similar to the analysis of Muon in the noisy case, we consider the worst-case where the predicted probability has achieved the true distribution at some point, but the optimization still continues at time step $t$.

When the diagonal elements of $\widehat{\mathbf{W}}_t$ still increase at time step $t$, the update of $\widehat{\mathbf{W}}_t$ is:

$$\widehat{\mathbf{W}}_t = \widehat{\mathbf{W}}_{t-1} + \eta\left(2\mathbf{I}_K - \mathbf{J}_{K,K}\right).$$

As we did to analyze the excess risk of Muon in the noisy case, we analyze the excess risk of the first task as an example. The score related to the first task at time step $t+1$ is:

$$s_{t,1|1}^+ = s_{t-1,1|1}^+ \cdot e^{\eta},$$
$$s_{t,i|1}^- = s_{t-1,i|1}^- \cdot e^{-\eta}, \quad \forall i \neq 1.$$

The predicted probability at time step $t$ is:

$$\widehat{p}_{t,1|1} = \widehat{p}_{t-1,1|1} \cdot \frac{e^{\eta}}{S} = p_{1|1} \cdot \frac{e^{\eta}}{S},$$
$$\widehat{p}_{t,i|1} = \widehat{p}_{t-1,i|1} \cdot \frac{e^{-\eta}}{S} = p_{i|1}\frac{e^{-\eta}}{S}, \quad \forall i \neq 1,$$

where $S = \widehat{p}_{t-1,1|1}e^{\eta} + \sum_{l\neq 1}\widehat{p}_{t-1,l|1}e^{-\eta}$ is the normalization term.

The excess risk related to the first task is:

$$\mathcal{L}_1(\mathbf{W}_t) - \mathcal{L}^* \leq -\sum_{i=1}^{K} p_{i|1}\log\widehat{p}_{t,i|1} + \sum_{i=1}^{K} p_{i|1}\log p_{i|1}$$
$$= \sum_{i=1}^{K} p_{i|1}\log\left(\frac{p_{i|1}}{\widehat{p}_{t,i|1}}\right)$$
$$= \log S - (1-\alpha+\frac{\alpha}{K})\eta + \frac{(K-1)\alpha}{K}\eta$$
$$= \log S - (1-2\alpha)\eta.$$

For $\eta \ll 1$, $S$ can be approximated as:

$$S = \left(1-\alpha+\frac{\alpha}{K}\right)e^{\eta} + \frac{(K-1)\alpha}{K}e^{-\eta}$$
$$\approx 1 + \left(1-\alpha+\frac{\alpha}{K}\right)\eta + \left(1-\alpha+\frac{\alpha}{K}\right)\frac{\eta^2}{2} - \frac{(K-1)\alpha}{K}\eta + \frac{(K-1)\alpha}{K}\cdot\frac{\eta^2}{2} + \mathcal{O}(\eta^3)$$
$$\approx 1 + (1-2\alpha)\eta + \frac{\eta^2}{2} + \mathcal{O}(\eta^3).$$

Thus, the excess risk related to the first task can be bounded as:

$$\mathcal{L}_1(\mathbf{W}_t) - \mathcal{L}^* = \log S - (1 - 2\alpha)\eta \le S - 1 - (1 - 2\alpha)\eta$$
$$= \frac{\eta^2}{2} + \mathcal{O}(\eta^3) \lesssim \eta^2.$$

Combining the excess risk related to all tasks, we have:

$$\mathcal{L}(\mathbf{W}_t) - \mathcal{L}^* \lesssim \eta^2. \tag{56}$$

When the diagonal elements of $\widehat{\mathbf{W}}_t$ decrease at time step $t$, we can derive the same excess risk bound.

Compared to the excess risk analysis of Muon in the noisy case, the excess risk bound of TRA-SignGD after entering the oscillation phase has the same rate of $\mathcal{O}(\eta^2)$. If the constant is considered, the excess risk bound of TRA-SignGD is roughly four times as large as that of Muon.

### F.3.4. SCALING LAW

**Theorem F.2** (Scaling law for TRA-SignGD). *Let $\eta = \Theta\left(\frac{\log K}{T}\right)$, we have:*

$$\mathcal{L}^{\text{TRA-SignGD}}(T) - \mathcal{L}^* \lesssim \left(\frac{\log K}{T}\right)^2.$$

*Proof of Theorem F.2.* Under the same assumption mentioned in Section E, we analyze the scaling behavior of TRA-SignGD in a large- scale training setup. Combining the excess risk analyzed above (Equation (55) and Equation (56) ), we have:

$$\mathcal{L}(\mathbf{W}_T) - \mathcal{L}^* \le \begin{cases} (K - 1)e^{-2\eta T} + 2\alpha\eta T - \mathcal{L}^*, & T < T'; \\ \eta^2, & T \ge T'. \end{cases}$$

As we analyzed for Muon, there is a trade-off in choosing $\eta$ to minimize the excess risk after $T$ iterations. The $\eta$ can be chosen as:

$$\eta = \frac{1}{2T} \log\left(\frac{(1 - \alpha)K}{\alpha}\right).$$

According to the analysis of $T'$, the above choice of $\eta$ ensures $T \ge T'$. The excess risk after $T$ iterations can be bounded as:

$$\mathcal{L}(\mathbf{W}_T) - \mathcal{L}^* \lesssim \eta^2 \lesssim \frac{1}{T^2}\left(\log\left(\frac{(1 - \alpha)K}{\alpha}\right)\right)^2 \lesssim \frac{(\log K)^2}{T^2}.$$

$\square$

### F.4. Comparison Between Muon and TRA-SignGD

Compared to TRA-SignGD, the update dynamics of Muon in the noisy case are approximately equivalent, with two primary differences:

- The effective learning rate of Muon is roughly half that of TRA-SignGD. Consequently, the number of steps to enter the oscillation phase ($T'$) of Muon is roughly twice that of TRA-SignGD, the excess risk after entering the oscillation phase of Muon is roughly one-fourth that of TRA-SignGD.

- In the noisy case, all tasks enter the oscillation phase at the same time when optimized via TRA-SignGD. When optimized via Muon, different tasks may enter the oscillation phase at different times, but this temporal discrepancy is negligible.

However, TRA-SignGD is an idealized optimizer that assumes prior knowledge of the embedding matrices $\mathbf{E}$ and $\widetilde{\mathbf{E}}$, which is unavailable in practical scenarios. In contrast, Muon does not require any prior knowledge of the embedding structure, yet its performance is comparable to that of TRA-SignGD.

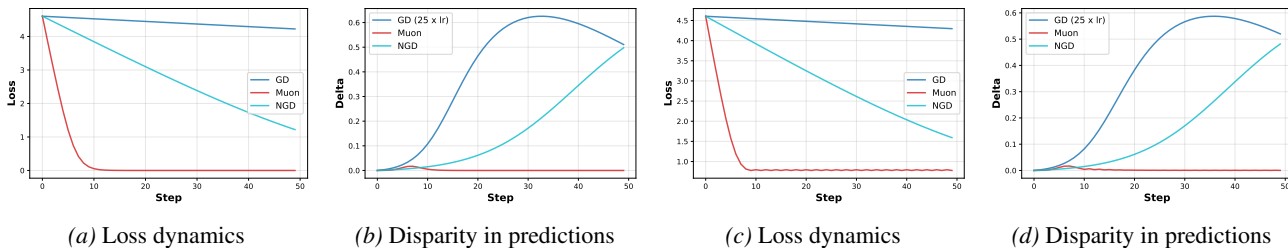

*Figure 3.* (a)(b): The Noiseless case; (c)(d): The Noisy case.

# G. Experiment Details

## G.1. Numerical Simulation

The results are shown in Figures 1a and 1b.

**Experiment Setup.** We generate two orthogonal matrices independently to represent $\mathbf{E}$ and $\widetilde{\mathbf{E}}$. We set the number of pairs $K = 100$, partitioned into $M = 10$ groups of size $C = 10$, with the group probability distribution $\tilde{p} = \{0.15, 0.1, 0.1, \ldots, 0.1, 0.05\}$. The noise level is set as $\alpha = 0.1$. For all experiments, the weight matrix is initialized as $\mathbf{W}_0 = \mathbf{0}$ and the optimization is conducted for $T = 50$ iterations.

**Optimizers and Hyperparameters.** We evaluate the optimization performance of GD, Muon, TRA-SignGD, SignGD and AdamW ($\beta_1 = 0.9$, $\beta_2 = 0.999$, weight decay = 0.01). To ensure a fair comparison, all optimizers share an identical learning rate of $\eta = 0.75$ when comparing the loss. When it comes to Delta(the maximal probability gap, $\Delta_t = \max_{j \in [K]} \widehat{p}_{t,j|j} - \min_{j \in [K]} \widehat{p}_{t,j|j}$), GD employs a larger learning rate of $25\eta(18.75)$ to better visualize the optimization progress within the 50-step budget, while other optimizers maintain the learning rate of $\eta = 0.75$. Furthermore, we also include a case that applies the Muon optimizer with a doubled learning rate ($2\eta$, 1.5) to provide a direct comparison between Muon and TRA-SignGD.

## G.2. Synthetic Imbalanced Classification

The results are shown in Figures 1c and 1d.

**Dataset.** We construct an imbalanced dataset based on the MNIST. Initially, Class 0 is excluded from both the training and the test dataset. We partition the remaining classes into three groups and remove parts of its samples from the training set to simulate an imbalanced probability distribution.

- The Many-shot group (Classes 1-3): retain all original samples in the training set (about 6k samples/class).
- The Medium-shot group (Classes 4-6): retain 50% of the original samples (about 3k samples/class).
- The Few-shot group (Classes 7-9): retain 25% of the original samples (about 1.5k samples/class).

The test set remains at its original size (excluding Class 0) to ensure an unbiased evaluation across all categories.

**Optimizers and Hyperparameters.** We compare the optimization performance of SGD, momentum-free Muon, full Muon ($\beta = 0.9$), AdamW ($\beta_1 = 0.9$, $\beta_2 = 0.999$, weight decay = 0.01). Both optimization processes are conducted for 20 epochs with a learning rate of 0.005. All optimizers share an identical random initialization and a batch size of 128.

**Architecture.** We employ a two-layer MLP with 128 hidden units and ReLU activation, optimized via Cross-Entropy Loss.

**Evaluation.** We monitor the training loss per epoch. To evaluate the imbalance of optimization, we define the maximum group accuracy gap as the difference between the maximum and minimum group average accuracies. Specifically, we first compute the average accuracy across all classes within each group. Then, we calculate the gap between the highest and lowest average accuracies among the three groups.

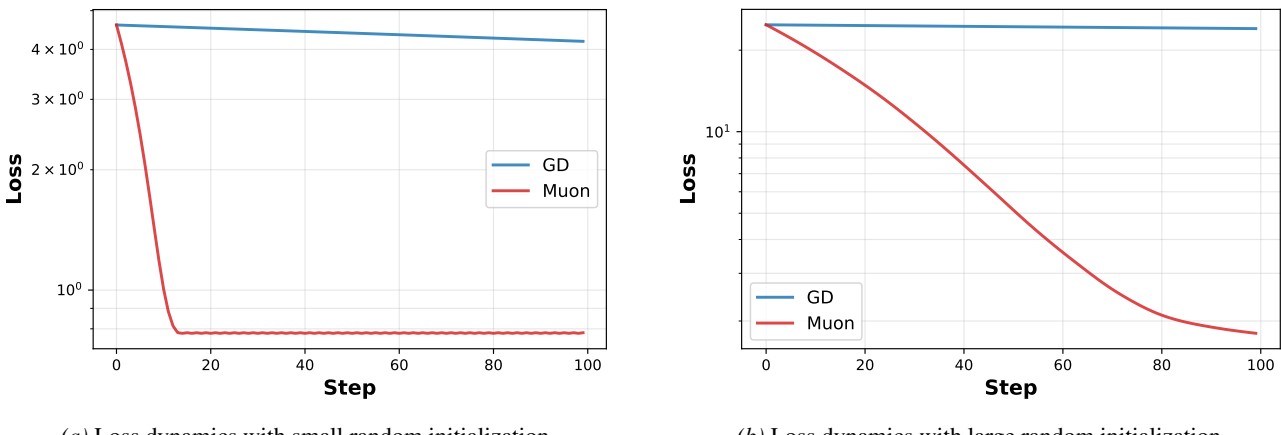

*(a)* Loss dynamics with small random initialization      *(b)* Loss dynamics with large random initialization

*Figure 4.* Loss comparison under different initializations.

### G.3. Numerical Simulations with Normalized GD

We additionally compare Muon with normalized GD (NGD), whose update is $\mathbf{W}_{t+1} = \mathbf{W}_t - \eta \frac{\nabla \mathcal{L}(\mathbf{W}_t)}{\|\nabla \mathcal{L}(\mathbf{W}_t)\|_F}$.

**Setting.** We use the same associative-memory setup as in Appendix G.1. We consider both the noiseless case, with $\alpha = 0$, and the noisy case, with $\alpha = 0.1$. For each setting, we compare the loss dynamics and the maximal probability gap defined at Section 7.1. All optimizers share the same learning rate of $\eta = 0.75$ when comparing the loss, while GD employs a larger learning rate of $25\eta(18.75)$ when comparing the maximal probability gap.

**Results.** The results are shown in Figures 3. NGD improves over GD in optimization speed, but it shows a pronounced increase in the maximal probability gap ($\Delta_t$) in both settings, compared to Muon. In contrast, Muon decreases the loss more rapidly while keeping $\Delta_t$ close to zero, indicating that Muon's advantage is not solely due to update normalization.

### G.4. Random Initialization

**Setting.** The updates in the task-representation space can be written as

$$\widehat{\mathbf{W}}_{t+1} = \widehat{\mathbf{W}}_t - \eta \mathbf{G}_{\text{optimizer},t}$$

, where $\mathbf{G}_{\text{optimizer},t}$ denotes the optimizer-induced update matrix in the task-representation space at step $t$. We visualize $\mathbf{G}_{\text{optimizer},t}$ across different optimizers, initializations, and time steps to provide insights into the optimization dynamics. We use the same associative-memory setup as in Appendix G.1, with $K = 100$, $M = 10$, $C = 10$, and $\alpha = 0.1$, and visualize the update matrices at eight time steps during optimization. We consider two random initializations with different scales, together with zero initialization as a baseline. For the random initializations, we first sample each entry of $\mathbf{W}_0$ from a Gaussian distribution with mean 0 and standard deviation 1. For the small random initialization, we normalize $\|\mathbf{W}_0\|_F$ to 1; for the large random initialization, we multiply the normalized matrix by 1000. We use same learning rates for all optimizers.

**Results.** The loss comparison under different initializations is shown in Figure 4 and the visualization of update matrices is shown in Figure 5. With small random init, Muon remains close to the zero-init case; with larger random init, it is still largely diagonal and balanced but noisier. GD consistently shows clear frequency-dependent disparities.

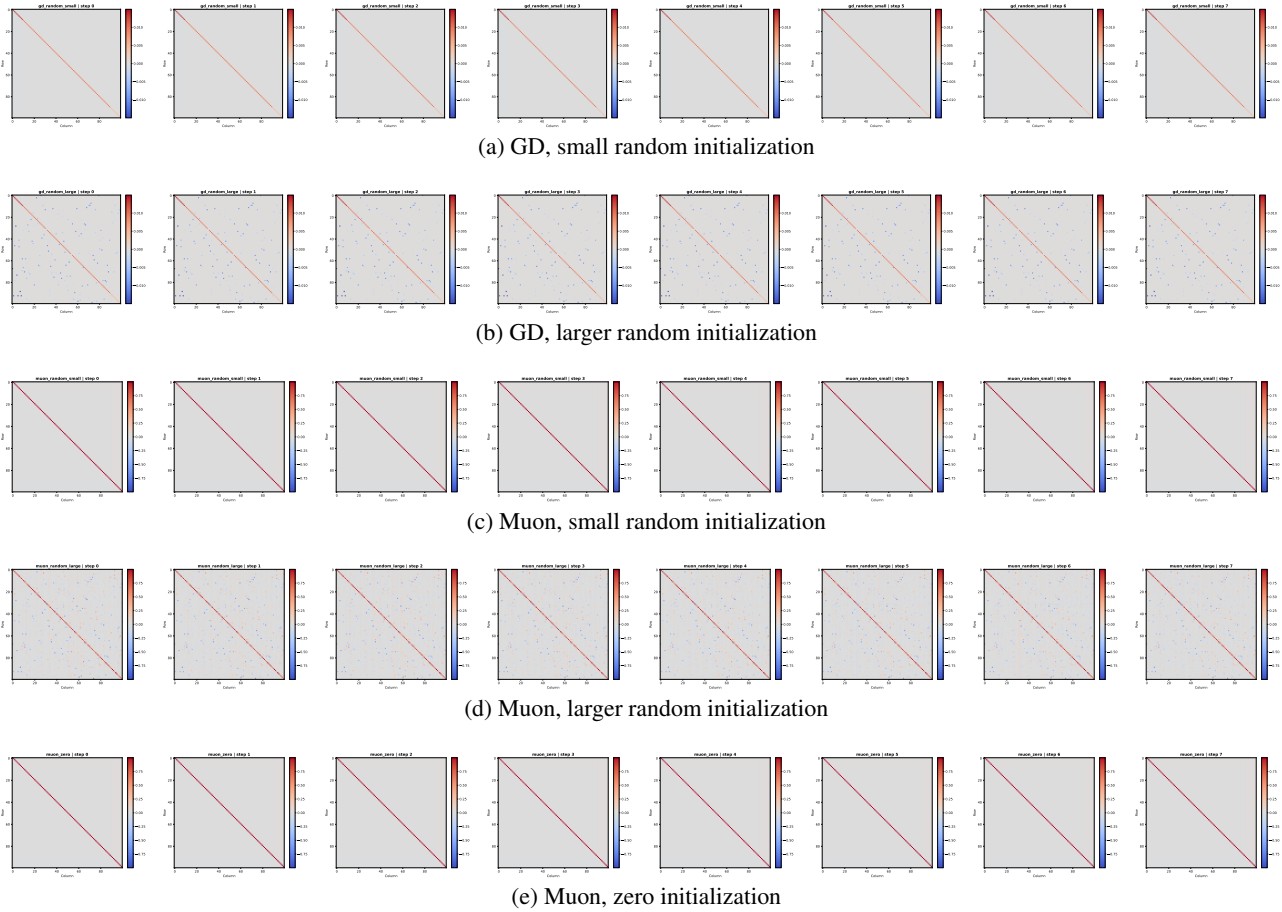

(a) GD, small random initialization

(b) GD, larger random initialization

(c) Muon, small random initialization

(d) Muon, larger random initialization

(e) Muon, zero initialization

*Figure 5.* Visualization of task-space update matrices across different initializations and steps. Each row corresponds to a different setting, showing the evolution at 8 time steps.

