# OpenReview forum: "Muon in Associative Memory Learning: Training Dynamics and Scaling Laws"
_ICML.cc/2026/Conference — ICML 2026 regular_

### Official Review · Reviewer_udDA · 2026-03-11

**Soundness:** 3
**Presentation:** 3
**Significance:** 3
**Originality:** 3
**Overall Recommendation:** 5
**Confidence:** 4

**Summary:**

This work investigates the training dynamics and scaling laws of the Muon optimizer within associative memory models. It characterizes an improved convergence rate and a scaling law in training time compared to GD, as Muon effectively mitigates spectral imbalance.

**Compliance With Llm Reviewing Policy:**

Affirmed.

**Key Questions For Authors:**

- It appears that the main results for GD do not strictly require Assumption 3.2; could the authors confirm if this is correct? If so, I suggest clarifying whether this assumption is specifically for the Muon analysis or if it holds for the entire framework.
- Could the authors provide any intuition or theoretical insight regarding the behavior of SignGD within this specific associative memory setting?
- In the numerical results, SignGD exhibits highly oscillating behavior. Could the authors provide an explanation for this phenomenon?

**Limitations:**

This work does not discuss its limitations and potential societal impact. Although the work is primarily theoretical, I suggest the authors explicitly address the limitations of their theoretical settings, including the simplified Muon setup (referred to as SpecGD in some literature).

**Strengths And Weaknesses:**

### **Strengths**

The paper is well-written, and the overall presentation is clear enough to follow the main arguments easily. The theoretical results effectively establish the contrasting behaviors of GD and Muon. Specifically, in the noiseless case, the authors provide tight convergence rates for both methods. For the noisy case, the study characterizes the training dynamics and scaling laws by establishing an upper bound for Muon and a lower bound for GD. This approach strongly supports their claims, as it provides a more rigorous comparison than some existing literature that relies solely on comparing two upper bounds, which may not definitively prove a performance gap.

### **Weaknesses**

My main concern is the connection between hierarchical frequency (Assumption 3.2) and the main results. It is hard to capture why this assumption is required for the analysis, as the current draft does not include the reasoning behind it before Section 6. I believe addressing this point can improve the manuscript and lead to a better understanding of the underlying mechanism.

---

> ### Author Rebuttal · Authors · 2026-03-31
>
> We are grateful for the reviewer's appreciation of our work and the positive feedback.
>
> **W1&Q1:** Yes, you are right. The analysis of GD in Thms.4.1 and 5.5 do not strictly rely on Assumption 3.2. The frequency hierarchy is a phenomenon observed empirically, which makes the weight matrix block-structured in the task-representation space. Due to this, the SVD of the matrix can be characterized clearly, making the analysis of Muon possible. We consider GD under the same data structure to make the speedup and scaling-law gap explicit. We will clarify this near Assumption 3.2.
>
> **Q2:** Viewed in the task-representation space, the update of SignGD is $\widehat{\mathbf{W}} _{t+1} = \widehat{\mathbf{W}} _t - \eta \widetilde{\mathbf{E}}^{\top}\operatorname{sgn}(\widetilde{\mathbf{E}}\mathbf{G}  _t\mathbf{E}^{\top})\mathbf{E}$, where $\mathbf{G} _t = \widetilde{\mathbf{E}}^{\top} \nabla _{\mathbf{W} _t} \mathcal{L}(\mathbf{W} _t) \mathbf{E}$ is the gradient in the task-representation space. Since the space is unknown, the updates of SignGD can be roughly viewed as applying the aligned gradient in an arbitrary coordinate system, performing the sign operation, and rotating it back. In this way, the behavior of each sub-task cannot be guaranteed. In particular, the performance on low-frequency sub-tasks is poor.
>
> **Q3:** As we mentioned above, SignGD applies coordinate-wise sign in an arbitrary basis, which is misaligned with the task representation most of the time. In the task-representation space,the high-frequency sub-tasks can update in the correct direction because of their large gradients, while the directional signals of the low-frequency sub-tasks are submerged because the gradients of high-frequency sub-tasks dominate the direction after rotation. In the task-representation space, intuitively, the update directions of low-frequency sub-tasks are dominated by noise, which means the optimization on these sub-tasks is ineffective. When the number of low-frequency sub-tasks is large enough that their loss dominates the total loss, the optimization with SignGD will oscillate at a high loss. Additionally, as the sign operator is taken in an arbitrary coordinate system, the effective update norm in the task-representation space might be quite large (it is observed in our numerical experiments). When the high-frequency sub-tasks approach the optimal point and oscillate around it, the amplitude of the loss could be quite large.
>
> **Limitations:** We agree that our theory is developed under a simplified setting. In particular, we study the momentum-free Muon (sometimes referred to as SpecGD) with zero initialization, in a linear associative-memory model with orthogonal embeddings. These assumptions make the dynamics tractable, but they do not fully capture practical large-scale training. Extending the analysis to non-zero initialization, momentum, and more general structures is an important direction for future work. We will state these limitations explicitly in the final version.

---

> > ### Author Rebuttal · Reviewer_udDA · 2026-04-03
> >
> > Thanks to the authors for their detailed rebuttal. It successfully addressed my concerns and questions. I suggest that the authors include this discussion on SignSGD in their revision. I maintain my positive evaluation.

---

> > > ### Author Response · Authors · 2026-04-05
> > >
> > > We sincerely thank the reviewer for the positive assessment and the encouraging feedback. We are grateful for the recognition of our work's contribution and appreciate the time invested in reviewing our manuscript. We will include this discussion on SignSGD in their revision as you suggested.

---

### Official Review · Reviewer_LFGX · 2026-03-15

**Soundness:** 3
**Presentation:** 2
**Significance:** 2
**Originality:** 3
**Overall Recommendation:** 4
**Confidence:** 4

**Summary:**

This work compares Muon and GD when training a linear softmax model for associative memory using population cross-entropy loss. The knowledge items are grouped and have a frequency imbalance. The key claim is that GD learns in an imbalanced manner, with rate proportional to the group frequency, while Muon learns in a balanced way regardless of group frequency. This is studied in both the noiseless case (one-to-one query-to-answer mapping) and the noisy case (soft label distribution over answers). Further, the authors show how Muon's balanced learning translates to faster total loss convergence, and derive scaling laws for the case where group frequencies follow a power law. Finally, they provide a preconditioning view to explain how Muon overcomes the frequency imbalance.

**Compliance With Llm Reviewing Policy:**

Affirmed.

**Final Justification:**

The authors addressed my main concerns in the rebuttal. They acknowledged the looseness of the Phase 1 bounds and the need to clarify the tight end-of-Phase-1 analysis in the main text, agreed to discuss the Muon/SpecGD terminology more carefully, and provided empirical comparisons with NGD. Given the community's current interest in Muon and the timeliness of this work in the linear associative memory framework, which has been heavily studied theoretically in the context of transformers, I believe this paper makes a worthwhile contribution. My main residual concern was whether the promised changes would fit coherently into the paper, but given the authors' detailed and concrete list of planned revisions, I am willing to put faith in them to follow through. It would not make sense to send this through another review cycle because of concerns on presentation and some small additional experiments and discussions imo, so I am raising my score to weak accept.

**Key Questions For Authors:**

Please see strengths and weaknesses.

**Limitations:**

Yes.

**Strengths And Weaknesses:**

**Strengths:**

The paper targets an interesting problem: understanding the benefits of the recently proposed and practically successful optimizer Muon over classical GD in the setting of associative memory learning, which has been heavily studied theoretically in the context of transformers. The use of frequency-imbalanced groups also seems like a natural testbed to demonstrate the benefits of Muon, as recent works have similarly exploited data imbalance to highlight advantages of spectral methods. The claimed results are intuitively sound given Muon's spectral orthogonalization update, which is known to equalize learning across directions. Given recent theoretical work on scaling laws for GD and SignGD, this is a timely paper. The paper is generally written fine that is easy to read, but there's scope for improvement as discussed below.

I am leaving weaknesses and questions together.

**Weaknesses and Questions**:

1. The paper uses the term Muon throughout, when the optimizer actually studied is Spectral GD (momentum/gradient accumulation is disabled, and the update is the matrix sign of the gradient). I would suggest updating the terminology throughout, as Spectral GD (Carlson et al. (2015a), (2015b); Vasudeva et al. 2025; Davis et al. 2025; Fan et al. 2025, etc.) is a common term used in the literature.

2. Proposition 3.3 presents a specific direction $W = \gamma E \tilde{E}^\top$ achieving minimum loss, but this is not the unique such direction. Any $W$ satisfying $\tilde{E}_j^\top W E_j > \tilde{E}_i^\top W E_j$ for all $i \neq j$ achieves the same infimum, defining an open cone of valid directions of which $W = \gamma E \tilde{E}^\top$ is simply the most natural element. I would suggest clarifying that Proposition 3.3 (and 3.4) exhibits one convenient element of this cone rather than characterizing the full solution set.

3. Theorems 4.1 and 4.2 show that Muon (SpecGD) achieves exponential loss convergence, while GD is $O(1/t)$. I believe this is an unfair comparison; the former is normalized steepest descent (under spectral norm), while the latter is standard steepest descent under the $\ell_2$ norm.  A more fair comparison would be normalized GD, which has a much faster $O(\exp(-\sqrt{T}))$ rate (follows from Theorem 8 in Nacson et al. (2019) or Deora et al. (2024) Table 1), at least for linearly separable data in binary classification. I would expect it to have a similarly fast rate here. While I still expect normalized GD’s sub-task loss to be dependent on frequency $p_j$ as it only uses raw gradient information, the rate should similarly be much faster. This should be taken into account for the results in both the noisy and noiseless cases.


4. I have several questions related to Theorems 5.1, 5.2, and 5.5. First, the critical time is denoted by $T_j$ however it is $\Theta(\log K/\eta)$ for all sub-tasks $j$, what does the subscript imply here? Moreover, if we plug this in the first phase expression, the excess risk is $O(\log K)$, while the Phase 2 excess risk is $O(\eta^2)$, are the bounds in the first phase loose, why the sudden shift? Continuing on this, Theorem 5.2 states that there is a second phase where some sub-tasks have been learned while some are in the oscillation phase, however the time duration of this phase seems short-lived as given $T_j = \Theta(\log K/\eta)$, I don't think I fully understand where this is coming from given Theorem 5.2 as all tasks have no dependence on group frequency. Finally, the loss achieved by GD at time $C\log K/\eta$ is $\Omega((\log K/K)^2)$, while for Muon it is $O(\log K)$ as we saw above. How is this comparison fair, am I missing something?

Further, Theorems 5.7 and 5.8 follow via application of bounds derived previously. For Theorem 5.8, $\eta$ is set in terms of training budget as $\eta = \Theta(\log K / T)$, is this a fair assumption given the learning rate you use for GD?


5. Section 6.1 (Lemma 6.1): Starting from zero initialization, each Muon step is essentially aligned with $\tilde{E}E^\top$ and continues to push in that direction, which also connects to the weight configuration in Proposition 3.3. This appears similar to work by Vasudeva et al. (2025) where show that for a linear model with class-imbalanced data, SpecGD starting from zero initialization immediately aligns with the $UV^\top$ direction of the data moment matrices at the first step and subsequently only increments singular values along that direction. Both results share the same structure: zero initialization + matrix-sign update $\rightarrow$ alignment with the correct basis, followed by pure scaling. I think it is worth discussing this connection and how general this is. For instance, I believe initializing $W_0 = \tilde{E}\hat{W}E^\top$ for non-zero $\hat{W}$ would also work. Related to this, how brittle is the proof to the initialization scheme? For random small initialization, what is the likely behavior, would there be a separate phase where the weights first align to the correct basis before scaling? It would be interesting to see this verified numerically.


6. TRA-SignGD: I don't fully understand the purpose of this part. The claim seems to be that the rotated problem has a block-diagonal Hessian, so coordinate-wise updates make sense and Muon inherently does this without access to the basis. However, for coordinate-wise updates to be optimal, would one not need a strictly diagonal structure within each block rather than just block-diagonal?

7. The experiments overall seem rather rushed. Figure 1 is missing captions. Figures 1(c) and 1(d) appear repetitive of Vasudeva et al. (2025), and it is unclear what additional insight they convey. Similarly, the language model pre-training experiments in Section 7.3 largely confirm already known empirical findings Liu et al. (2025) that Muon outperforms SGD in LLM pre-training, and don't really add anything new.

Currently, I have several questions and criticisms regarding the paper that may require a significant update; therefore, I am assigning a reject score. However, I would be happy to discuss these points with the authors during the rebuttal and reconsider.

**References**:

Carlson et al. (2015a). Stochastic spectral descent for restricted boltzmann machines.

Carlson et al. (2015b). Preconditioned spectral descent for deep learning.

Fan et al. (2025). Implicit Bias of Spectral Descent and Muon on Multiclass Separable Data

Nacson et al. (2019). Convergence of gradient descent on separable data.

Deora et al. (2024). Fast test error rates for gradient based algorithms on separable data.

Vasudeva et al. (2025). How Muon's spectral design benefits generalization: A case study on imbalanced data.

Liu et al. (2025). Muon is scalable for LLM training.

---

> ### Author Rebuttal · Authors · 2026-03-31
>
> We thank the reviewer for helpful and insightful comments, as well as for pointing out several relevant works, all of which we will cite and discuss appropriately in the revision. Our point-by-point responses are given below.
>
> **W1:** We agree that Muon is equivalent to SpecGD without momentum, but the use of "Muon" is still reasonable, as we aim to explain Muon's recent empirical success in large-scale training. The momentum term is intentionally removed to focus analysis on the preconditioning mechanism. We also use "Muon" to connect our study to ongoing LLM training discussions, helping practitioners map our analysis to practice. We will clarify the relationship between Muon and SpecGD, explain that our theory applies to SpecGD while aiming to illuminate matrix-sign preconditioners.
>
> **W2:** We agree. In noiseless case, directions approaching the loss infimum are not unique, so Prop.3.3 gives only one example. In Prop.3.4, the minimizer in weight space is not unique, while the induced conditional distribution $p_{i|j}$ is unique. We will clarify this.
>
> **W3:**
> - Current comparison: Our comparison with GD is still informative because our focus is not only convergence speed but also the qualitative optimization dynamics. In particular, GD has frequency-dependent effective step sizes, causing imbalanced subtask progress, whereas Muon exhibits descent-then-oscillation dynamics in task space; For speed, GD allows a much larger learning rate while Muon's learning rate must be small due to oscillation. Comparison under respective optimal learning rates is fair, which we complete in scaling case. We will revise paper to make this emphasis clearer.
> - NGD: Yes, it is a natural baseline for comparing convergence rates, but its theoretical analysis is substantially harder: **normalization couples all columns, leaving $M$ coupled nonlinear systems** and the finite optimum in noisy case causes oscillation. These lead to a different and more complicated optimization geometry compared to the cited works studying binary separable classification.
>
> **W4:** Due to space, we only present key points here and are willing to discuss them further in the following discussion. Details are in Appx.E.2&F.
> - Critical Time: All critical times share a same order since the Phase-1 task-space update is dominated by the $\mathbf{I}_K$ term, while $o_K(1)$ block matrices introduce lower-order differences and lead to a short mixed phase.
> - Excess Risk and Speedup: The tighter end-of-Phase-1 bound is $O(M^2\log K/K)$, requiring a detailed dynamical analysis instead of plugging $t=\Theta(\log K/\eta)$ merely. The $O(\eta^2)$ term is a loose worst-case bound on oscillation amplitude from one extra step at optimum. The claimed $C$-fold speedup is asymptotic for relatively fixed $M$: GD needs $\Omega(C\log(K/M^2)/\eta)$ steps to achieve Muon Phase-1 loss while Muon needs $O(\log K/\eta)$.
> - Fairness in $\eta$ Choice: It is fair as we do not assume any scale but optimize it separately. $\eta=\Theta(\log K/T)$ is the smallest scale ensuring all subtasks enter oscillation phase, giving an upper bound on Muon loss, while $\eta p_1=\Theta(1)$ gives a lower bound on GD loss.
>
> **W5:** Random init merits discussion. Since a full theoretical analysis is challenging, we add empirical analysis here.
> - Theory: Since the singular subspace changes during optimization, we need $\nabla \mathcal{L}(\mathbf{W}_t)$ to retain a highly structured form to control SVD. $W_0$ is required to be zero or satisfy other artificial structure and $W_0 = \tilde{E}\hat{W}E^\top$ alone does not suffice.
> - Empirical Analysis: We examine task-space update matrices in numerical experiments.[https://anonymous.4open.science/r/repository-1922/],Sec 7.1. With small random init, Muon remains close to the zero-init case; with larger random init, it is still largely diagonal and balanced but noisier. GD consistently shows clear frequency-dependent disparities. We use the same learning rate for all optimizers, include zero init as a baseline. A distinct alignment phase before scaling is not observed.
>
> **W6:**
> - Optimality of sign: As the reviewer noted, a block-diagonal Hessian alone is insufficient for optimality; in our setting, optimality instead follows from zero init and symmetry preservation. These reduce each subtask to a strictly convex 1D problem in the logit gap, ensuring sign always moves toward the optimum.
> - Motivation of Sec.6: The aim is to compare two preconditioners. Sign follows the ideal direction given task-representation basis, while msgn adaptively approximates it without such access and achieves comparable performance.
>
> **W7:**
> - The caption is already included at L396.
> - Fig.1(c)&(d): Settings and aims differ. While Vasudeva et al. use Colored-MNIST to study learning behavior, we use standard MNIST to validate our theory under relaxed assumptions empirically. Given the space, please see our reply to Reviewer RBvN(W3&W4) for detailed explanations of Secs.7.2&7.3.

---

> > ### Author Rebuttal · Reviewer_LFGX · 2026-04-04
> >
> > Thanks to the authors for their detailed rebuttal.
> >
> > W1/W2: Thanks for acknowledging. These should be clearly discussed in the main body.
> >
> > W3: The looseness of bounds and the tight end-of-Phase-1 analysis should be discussed more clearly in the main text. On NGD: while I get the possible technical difficulty, at least an empirical comparison with NGD should be included.
> >
> > W5: Thanks for the additional experiments, these are worthwhile and should be added to the paper.
> >
> > Overall, I feel there are several significant additions needed and in its current form I will maintain my weak reject score.

---

> > > ### Author Response · Authors · 2026-04-05
> > >
> > > We sincerely thank the reviewer again for the thoughtful feedback. We are encouraged that our previous responses clarified most points. Below, we address the remaining concerns regarding presentation and empirical support point-by-point.
> > >
> > > **W3:**
> > > - **The Looseness of Bounds:** The apparent looseness comes from the fact that, due to space limitations, we omitted the constant term $\mathcal{L}^{\ast}$, which is of order $O(\log K)$ and independent of $t$. As written, the RHS of Thm. 5.1 and Thm. 5.2 are essentially the upper bound of $\mathcal{L}^{\text{Muon}}(t)$ rather than the excess risk $\big(\mathcal{L}^{\text{Muon}}(t) - \mathcal{L}^{\ast}\big)$.
> > >
> > > We will revise Thm. 5.1 to:
> > > $$
> > > \\mathcal{L}_j^{\\text{Muon}}(t) \\lesssim
> > > \\left\\{
> > > \\begin{array}{ll}
> > > K e^{-\\eta(1+o_K(1))t} + \\eta t, & t \\le T_j^{\ast}, \\\\
> > > \\eta^2, & t > T_j^{\ast}.
> > > \\end{array}
> > > \\right.
> > > $$
> > >
> > > and revise Thm. 5.2 to:
> > > $$
> > > \\mathcal{L}^{\\text{Muon}}(t) \\lesssim
> > > \\left\\{
> > > \\begin{array}{ll}
> > > K e^{-\\eta(1+o_K(1))t} + \\eta t, & t \\lesssim \\tfrac{\\log K}{\\eta}, \\\\
> > > K e^{-\\eta(1+o_K(1))t} + \\eta t, & t \\eqsim \\tfrac{\\log K}{\\eta}, \\\\
> > > \\eta^2, & t \\gtrsim \\tfrac{\\log K}{\\eta}.
> > > \\end{array}
> > > \\right.
> > > $$
> > >     With the optimum of the loss $\big(\mathcal{L}^{\ast}=-(1-\alpha+\tfrac{\alpha}{K})\log(1-\alpha+\tfrac{\alpha}{K})-\tfrac{\alpha(K-1)}{K}\log\tfrac{\alpha}{K}\big)$, the tighter end-of-Phase-1 bound can be obtained by plugging in $t=\Theta(\log K/\eta)$ and carefully controlling the lower-order terms; the corresponding derivation is already included in the appendix.
> > >
> > > - **NGD:** We agree that an empirical comparison with NGD would better isolate the speedup caused by normalization and further strengthen our theory. We provide additional numerical experiments in both the noiseless and noisy case, using the same setup as in the paper (with $\alpha=0$ in the noiseless case). The results are available here: noiseless cases: [https://anonymous.4open.science/r/repository-1922/Sec7.1_NGD/noiseless_case.pdf]; noisy case: [https://anonymous.4open.science/r/repository-1922/Sec7.1_NGD/noisy_case.pdf].
> > >     NGD shows a pronounced increase in the maximal probability gap (Delta) in both settings. In contrast, Muon drives the loss down much faster while keeping the maximal probability gap (Delta) essentially at zero.
> > >     We will add a remark at the end of Sec.4 to discuss the possible influence of normalization and incorporate these empirical results.
> > >
> > > \
> > > The remaining revisions are as follows:
> > >
> > > **W1:** We will clarify the relationship between Muon and SpecGD and explain our main purpose for using "Muon", as we discussed in rebuttal, in Sec. 3.2, near the introduction of momentum-free Muon.
> > >
> > > **W2:** We will further clarify Prop. 3.3 and Prop. 3.4 as noted in the rebuttal.
> > >
> > > **W5:** We will add a new section before the conclusion to discuss the limitations of our theoretical setup. In particular, we will discuss the random-init setting, as in the rebuttal, provide the corresponding empirical observations and add experimental results in the appendix.
> > >
> > > If there are further questions or suggestions, we would be very happy to discuss them and clarify any remaining points. We hope that these revisions and additional experiments help address your concerns, and we would sincerely appreciate your reconsideration.

---

### Official Review · Reviewer_RBvN · 2026-03-20

**Soundness:** 2
**Presentation:** 3
**Significance:** 2
**Originality:** 3
**Overall Recommendation:** 3
**Confidence:** 3

**Summary:**

This paper studies the training dynamics of the Muon optimizer under the associative memory learning framework.
- The authors construct a linear softmax model to analyze Muon's training dynamics.
- The proposed explanation:
    - Gradient Descent (GD) learns different frequency components at imbalanced rates and is bottlenecked by low-frequency components.
    - Muon mitigates this (different frequency components learn at a more balanced rate), so it converges faster.
- The authors derive theoretical results for both noiseless and noisy cases and validate the theory with synthetic experiments.
- A pre-training experiment is also provided, to show that Muon optimized models are more data efficient and follow a power law scaling behavior.

**Compliance With Llm Reviewing Policy:**

Affirmed.

**Final Justification:**

My recommendation is weak reject (3) with a confidence of 3. The rebuttal partially addressed my main concerns, however, the main weakness remains. The added experiments are appreciated, however, it does not convince me that class imbalance can explain Muon's effectiveness.

**Key Questions For Authors:**

I am open to adjusting my scores based on the authors' answers and clarifications.
Especially to assess whether this is a contribution that others are likely to build on.
- In the best case, what is the practical impact of this work? How would this theory change or inform how people train models in the future?
- The proposed linear softmax model is essentially a single classification layer, but in practice, Muon is mainly used in the FFN layers between layers in LLMs. How is the linear softmax model analysis in this paper relevant to FFNs in LLMs?
- AdamW is also a very fast optimizer but is expected to be slower than Muon, can you compare against AdamW? Do the empirical results still hold?

**Limitations:**

No.
The authors should explicitly discuss the limitations of this work in the conclusion section.
For example, there is a gap between the softmax linear model and the feedforward networks.
The authors should show this gap and discuss why the theoretical results would generalize.

**Strengths And Weaknesses:**

## Strengths
- (Significance) Associative memory learning is a recent and relevant framework for understanding both attention mechanism and feedforward networks (FFNs).
Muon is also recent and relevant.
- (Originality) This paper focuses on Muon's training dynamics while most existing work on Muon focuses on convergence bounds under standard stochastic optimization frameworks. The authors claim that this is the first theoretical analysis characterizing the scaling laws of the Muon optimizer.
- (Presentation) The paper is clearly written. Related work is discussed and citations are sufficient.
- (Soundness) The authors provide theoretical derivations and used synthetic experiments on both linear softmax models and FFNs to verify this theory (Section 7.1 and 7.2).

## Weaknesses
- (Significance) This paper confirms that Muon is better than gradient descent, but does not inform future directions. How will this work change how people do things in this field\*?
- (Soundness) There is a fundamental gap between the theory and where Muon is actually used in practice. The theory is built on a linear softmax model, which is essentially a classification layer trained with cross entropy loss. However, in practice, Muon is applied to all FFNs in multi-layer networks. These layers get their gradients from backprop. Section 7.2 uses an MLP but the findings are still based on the training loss and classification accuracy (essentially showing that Muon trains faster and more evenly under class imbalance). This does not bridge the gap.
- (Soundness) Section 7.2 only show that Muon trains faster and that classes converge more evenly in image classification with class imbalance. It does not sufficiently verify the theoretical predictions.
- (Soundness) The Muon pretraining experiment in Section 7.3 does not verify the theory that Muon's success is due to balanced learning across frequency components. Instead, it only shows that Muon makes language models more data efficient and scale better, which is already known from existing large-scale pretraining such as [2].
- (Soundness) Training speed is the key indicator in Section 7, but other fast optimizers (such as AdamW) are not tested.


*: As an example, recent work [1] on associative memory tells us head size affects retrieval precision and thus we should use larger head sizes beyond 128 for very long context.

[1] Understanding Transformer from the Perspective of Associative Memory. https://arxiv.org/pdf/2505.19488

[2] Muon is Scalable for LLM Training. https://arxiv.org/pdf/2502.16982v1

---

> ### Author Rebuttal · Authors · 2026-03-31
>
> We thank the reviewer for the helpful feedback and insightful question, as well as for pointing out several relevant works, all of which we will cite and discuss appropriately in the revision.
>
> **[W1&Q1]:** While our paper is primarily theoretical, it sheds light on a central practical bottleneck in modern LLM training: **inefficient optimization under highly imbalanced data distributions**. Such imbalance stems from several factors: by Zipf’s law, natural language itself is heavy-tailed; modern LLMs are trained on heterogeneous mixtures of data sources which typically have highly uneven mixture weights. Consequently, optimization can be dominated by high-frequency patterns, leaving underrepresented sources to be learned much more slowly, which results in inefficient training.
> - **Practical Impact:** Our work illustrates how Muon lead to more uniform progress across the data distribution and provides a theoretical explanation for this success. It offers general suggestions on optimizer selection: when training data is hight imbalaced or rare frequency data matters, Muon is worth considering.
> - **Future directions:** As our current work focuses on associative memory models, the future work can extend to more realistic deep-network and LLM training settings, finding out how such alignment arises and what structures Muon captures in more realistic settings. Our task-space preconditioning view also uggests a design principle for future optimizers: exploit task structure and apply optimization aligning with it.
>
> **[W2&Q2]:** We appreciate the reviewer's perspective on the gap between theory and practice. We clarify this through two key points:
>
> - **Theoretical Simplification:** To achieve theoretical tractability, our analysis utilizes a linear softmax model. This is a standard and widely adopted abstraction for rigorously analyzing neural network dynamics (c.f. [1][2][3]).
> - **Beyond the Linear Case:** Our MLP experiments in Section 7.2 are specifically designed to bridge this gap. They empirically verify that our theoretical conclusions hold in deep, multi-layer networks: Muon learns imbalanced data more evenly, thereby significantly accelerating loss convergence.
>
> We will explicitly clarify this bridging role of Section 7.2 in the revision, and we remain highly open to any specific empirical suggestions the reviewer might have.
>
> **[W3]:** The purpose of the experiments in Sec. 7.2 is **not merely to show** that Muon is more efficient for image classification. Rather, it is to show **our theoretical conclusions are robust to relaxed assumptions**.
>
> We extend the single linear model to a two-layer MLP with ReLU and relax the strict orthogonal-embedding assumption (viewing the image inputs as the embeddings in our theoretical setting). The experiment is intended to test empirically whether the conclusions of our theory continue to hold beyond the idealized setting, and we will clarify this explicitly.
>
> **[W4]:** Our LLaMA experiment aims to verify our scaling-law analysis in the context of real-world language model pre-training. While the prior work mentioned by the reviewer [4] compares **Muon with AdamW**, our evaluation specifically verifies that the scaling efficiency gap between **Muon and SGD** empirically aligns with our theoretical predictions.
>
> **[W5&Q3]:** We agree that comparing against AdamW would strengthen the practical significance of Sec.7. Our current Sec.7 experiments are designed primarily to validate the theory, which focuses on Muon and treats GD as the baseline. That's why AdamW was not included.
> - **additional experiments:** We add additional experiments for Sec. 7.2 on MNIST, including **AdamW and Full-Muon** [https://anonymous.4open.science/r/repository-1922/Sec7.2.pdf], setting learning rate to 0.001, $\beta=0.9$ for Full-Muon and betas=(0.9, 0.999) with weight_decay=0.01 for AdamW. **Full Muon achieves lower training loss and more balanced optimization across groups.** We will add these results in the revision.
>
> **[Limitations]:** We agree that there is a gap between our theoretical setting and practical training, and we will discuss this limitation explicitly in the conclusion. We have presented several experiments to support the possibility that our theoretical conclusions may generalize, but they do not fully close the gap between theory and practice. We will add this discussion explicitly in the revised paper.
>
> **Reference**
>
> [1] Muon Outperforms Adam in Tail-End Associative Memory Learning.
>
> [2] Scaling Laws for Gradient Descent and Sign Descent for Linear Bigram Models under Zipf's Law.
>
> [3] Heavy-Tailed Class Imbalance and Why Adam Outperforms Gradient Descent on Language Models.
>
> [4] Muon is Scalable for LLM Training.

---

> > ### Author Rebuttal · Reviewer_RBvN · 2026-04-04
> >
> > The authors' response and additional experiments are appreciated and will be factored into the final decision.
> > - From [1], AdamW and Full-Muon have very slight difference in the group accuracy gap, so Muon's success may not be explainable from the theoretical perspective you proposed. This difference is very small, how much exactly is it? Is this really significant, and does it truly support your theory?
> > - You tested AdamW on MNIST for 7.2 , but what about Section 7.1 Numerical Simulations? Does the main claim still hold?
> >
> > [1] https://anonymous.4open.science/r/repository-1922/Sec7.2.pdf

---

> > > ### Author Response · Authors · 2026-04-05
> > >
> > > Thanks for the helpful follow-up feedback. We have conducted additional experiments to further clarify the significance of our results and the alignment with our theoretical claims.
> > >
> > > **Q1:** The difference in the group accuracy is slight.
> > >
> > > - **The difference is meaningful in this regime:** The absolute group-accuracy gap difference is approximately 0.01, which is not marginal when class-wise accuracies are already high for both optimizers in this setting.
> > > In particular, when the high-frequency group is already near saturation (~0.99), Full-Muon consistently maintains the low-frequency group above 0.95, whereas AdamW is less stable and does not consistently reach this level. To make this easier to inspect, we provide a zoomed-in view of the curves [https://anonymous.4open.science/r/repository-1922/Sec7.2_addition/original_experiment.pdf]
> > >
> > > We also report the average accuracy trajectories of the high-frequency and low-frequency groups. It shows that **Full-Muon and AdamW behave similarly on the high-frequency group, while Full-Muon achieves better performance on the low-frequency group**, indicating more balanced learning.
> > >
> > > - **Refined Empirical Results with Reduced Stochasticity:** We additionally evaluate AdamW/Full-Muon in a larger-batch training setting (changing the batch size from 64 to 128), which is better aligned with our full-batch theoretical setting and reduces the randomness in the results. The refined experiment is available at [https://anonymous.4open.science/r/repository-1922/Sec7.2_addition/additional_experiment.pdf]. The same qualitative conclusion holds more clearly: Muon and AdamW have very **similar performance on the high-frequency group**, but Muon achieves **better accuracy on the low-frequency group** and **more balanced learning**, resulting in a smaller group-accuracy gap and a lower training loss.
> > >
> > > **Q2: The Performance of AdamW in Numerical Experiment:** We additionally include AdamW in the numerical simulations, using $\beta_1=0.9,\beta_2=0.999$ and weight_decay = 0.01.[https://anonymous.4open.science/r/repository-1922/Sec7.1_AdamW.pdf] Compared with Muon, AdamW exhibits pronounced oscillations in its loss trajectory. Notably, the maximal probability gap (Delta) of AdamW shows large, non-systematic fluctuations, indicating more imbalanced learning across groups.
> > >
> > >
> > > \
> > > We would also like to clarify the **scope of the paper**. The paper primarily aims to **understand Muon's mechanism through training dynamics, with GD as the main theoretical baseline**. Accordingly, the experiments were originally designed as theory-oriented validations under controlled or relaxed settings, rather than as a full practical benchmark. **Our claim that Muon yields more balanced learning than GD is already supported by the experiments in the paper.**
> > >
> > > Since we do not aim to compare Muon and Adam directly, we did not initially include Adam/AdamW comparisons. That said, we agree that adding Adam/AdamW makes the empirical picture more complete, and we will incorporate these comparisons and clarify the scope of our empirical claims in the revision.
> > >
> > > We sincerely thank the reviewer again for the thoughtful feedback. We are happy to provide any further information or elaboration if there are any remaining concerns.

---

### Official Review · Reviewer_NEke · 2026-03-23

**Soundness:** 3
**Presentation:** 2
**Significance:** 2
**Originality:** 3
**Overall Recommendation:** 4
**Confidence:** 3

**Summary:**

In this paper, the authors study a linear associative-memory model with orthogonal query and answer embeddings, a hierarchical frequency spectrum over knowledge items, softmax retrieval, and potential uniform label noise. In noiseless setting, they argue that gradient descent (GD) learns rare items much more slowly because update magnitudes scale with item frequency, while Muon updates approximately equalizes progress across different groups. In the noisy case, this paper shows Muon has descent, mixed, and oscillation phases, with excess risk eventually controlled at order of square of learning rate, and that GD is slower by an $\Omega(C)$ factor when the group size is $C$. Moreover, the authors derive a scaling-law result under a power-law frequency spectrum, and interpret Muon as an implicit matrix preconditioner in a task-aligned basis.

**Compliance With Llm Reviewing Policy:**

Affirmed.

**Key Questions For Authors:**

1. Is the global minimizer for noisy case unique? When the embedding dimension $d>K$, this minimizer is not unique, right?

2. Below Theorem 4.1, you stated that the total loss converges at a rate of $O(1/t)$, which is consistent with the convergence rate of logistic regression on linearly separable data. Can you explain how is the linear associative-memory model related to the logistic regression?

3. In Theorems 4.1 and 4.2, are you considering $t$ is fixed and taking $K\to \infty$ to get the asymptotic behavior of the loss function? There should be a clear statement of the scaling limit for all theorems. For instance, in Theorems 5.7 and 5.8, do you consider both $K$ and $T$ going to infinity?

4. In your Muon algorithm, why do you use matrix sign function instead of matrix orthogonalization as the conventional Muon algorithm?

5. In the scaling law section, you assumed $M=w_K(1)$. How is the setting of Section 5.2 different from previous sections?

6. You may need explicitly explain the definition of disparity in predictions in Figure 1(b).

7. There is no details of experiments of Language model pre-training in Section 7.3.

**Limitations:**

yes

**Strengths And Weaknesses:**

# Strengths:

1. The paper is well-organized and clear written for machine learning community. The presentation is good and original. The most original part is the Muon-specific scaling-law analysis in the noisy associative-memory setting.

2. The submission is technically sound to me. The problem studied in this paper is significant for deep learning and practice.

# Weaknesses:

1. There are some problems in the statements of theorems and proofs. The assumptions on scalings are not clearly stated in the theorems: Remark D.3 says $1/C = M/K \ll 1/K$, which does not make sense; besides, the authors assumed $K \gg 1$ and $M \ll C$, while in the scaling law section they assumed $M = \omega_K(1)$. What is the assumption on $C, M, K$ in Lemma 6.1? Additionally, in the proof of Lemma 6.1, line 330 concludes with $\lVert \operatorname{msgn}(\hat P_0 - P) - I_K \rVert_\infty$, which should involve $\hat P_t$ not $\hat P_0$. The proof of Theorem 4.2 in Appendix D.2 is also sloppy: at line 1312 and below, the authors need more explanation of how to obtain the bound in (25) from $\lvert (u_{t',\ell} v_{t',\ell})_{iC,1} \rvert \le 1$. And, where is the proof of the "if and only if" direction in Proposition 5.4?

2. One concern I have is the gap of the analysis of this paper and practical application. The practical motivation is “why Muon beats Adam/AdamW,” while the core theory of this paper is mostly Muon vs GD, and additionally a task-aligned sign GD baseline rather than full Adam. The experiments also do not include Adam/AdamW. This may lead to a weakness of the paper as [1] also shows that  the loss of infrequent words decreases more slowly than the loss of frequent ones for GD but Adam and sign-based methods do not.

3. Section 6 should compare the TRA-SignGD with Shampoo [2] and SOAP [3]. The paper also needs to state the limitation of the current results


----------------------------
[1] Heavy-Tailed Class Imbalance and Why Adam Outperforms Gradient Descent on Language Models, 2024

[2] Shampoo: Preconditioned Stochastic Tensor Optimization, 2018

[3] SOAP: Improving and Stabilizing Shampoo using Adam for Language Modeling, 2025

---

> ### Author Rebuttal · Authors · 2026-03-31
>
> We are grateful for the positive feedback.
>
> **W1&Q3&Q5:** Typos fixed: $1/C=M/K\ll1/K$ should be $1/C=M/K\ll1/\sqrt{K}$. The conclusion at L330 should be $\|\text{msgn}(\mathbf{P}-\widehat{\mathbf{P}} _t) -\mathbf{I} _{K}\| _{\infty}=o _{K}(1)$. In Appx.D.2, the proof of Thm.4.2 should be $|(\mathbf{u} _{t',l}\mathbf{v} _{t',l}^{\top}) _{iC,1}|\leq 1/C$, hence $|(\mathbf{u}  _{t',l}\mathbf{v} _{t',l}^{\top}) _{iC,1}-(\mathbf{u} _{t',l}\mathbf{v} _{t',l}^{\top}) _{1,1}|\leq2/C.$ Summing over $l$, $|\tilde{s}  _{t',i}|\le\sum _{l=1}^M|(\mathbf{u} _{t',l}\mathbf{v} _{t',l}^\top) _{iC,1}-(\mathbf{u} _{t',l}\mathbf{v} _{t',l}^\top) _{1,1}|\le2M/C =2M^2/K$. We will correct them.
>
> For assumptions on $K,M,C,$ and $t$, except in Thms.5.7&5.8, we consider $K$ to be a sufficiently large constant s.t. $1\ll K$. $M$ and $C$ are fixed constants with $M \ll C$. We let $t\to\infty$ to get the asymptotic loss while keeping dependence on $K$ explicit to show task-number effects. In Thms.5.7&5.8, $K,M,T\to\infty$ jointly, with $cM^{\beta}\leq T\leq M^{\beta}$ for some $0<c<1$, and $(\log K)^{\frac{1}{\beta}}\leq M\ll K^{\frac{1}{2}}$. Here $M=\omega_K(1)$ means $M\to\infty$ rather than remaining constant(the key difference between scaling law case and others). These assumptions are in Appx.F and will be emphasized in main text.
>
> For Prop.5.4, Appx.E.1(Lem.E.2) establishes the claim in a manner closely related to the only-if direction of Prop.5.4; this is the only part needed in the following proof. We agree, however, that a separate formal argument should be provided. Here is proof sketch.
> > Consider the dynamics of $\widehat{\mathbf{W}} _t$, where $\widehat{\mathbf{W}} _t=\widetilde{\mathbf{E}}^{\top}\mathbf{W}  _t\mathbf{E}$ is the weight matrix in the task-representation space. Since all columns update independently and similarly, it suffices to study column 1. $(\widehat{\mathbf{W}} _{t+1}) _{:,1}=(\widehat{\mathbf{W}} _t) _{:,1}-\eta p _1(\text{softmax}((\widehat{\mathbf{W}} _t) _{:,1})-p _{\cdot|1})$, where $p _{\cdot |1}=(1-\alpha+\frac{\alpha}{K},\frac{\alpha}{K},\cdots,\frac{\alpha}{K})^{\top}$. Let $\widehat{\mathbf{W}}^* _{:,1}$ be a fixed point satisfying $\text{softmax}((\widehat{\mathbf{W}}^*) _{:,1})=p _{\cdot |1}$.
>
> >By translational invariance of softmax, fixed points are non-unique, but the Jacobians are the same: $\mathbf{J^*}=\eta p _1(\text{Diag}(p _{\cdot|1})-p _{\cdot |1}(p _{\cdot |1})^{\top})$.
>
> >Its eigenvalues are $0$, $\eta p _1\alpha(1-\alpha+\alpha/K)$, $\eta p _1\alpha/K$ ($K-2$ times). Linear stability for GD with learning rate $\eta$ is equivalent to the existence of $C>0$ such that $\|(\mathbf{I}-\mathbf{J}^*)^t\| _2\leq C$ for all $t$, and similarly for all columns. As $p _1\geq\cdots\geq p _K$, it is equivalent to $\eta p _1 \lesssim 1$, proving both directions.
>
> **W2:** Our goal is to illustrate Muon's mechanism through dynamics, using GD as a baseline. Focusing on preconditioning, we compare msgn with sign used by Adam. While sign preconditions in canonical coordinates, msgn adapts to task directions. We do not aim at a direct Muon-Adam comparison, but we believe the insight helps to understand both. Besides, simplifying Adam to SignGD is a common approach in theoretical analysis(cf.[1]-[4]).
>
> **W3:** Sec.6 aims to compare the effects of preconditioning on coordinate-wise and matrix-sign operators. Covariance-based preconditioners (e.g., Shampoo, SOAP)fall outside our sign-based scope. We will state this explicitly.
>
> **Q1:** The global minimizer is not unique due to softmax translation invariance, and when $d>K$ there are additional null directions, but the optimal conditional distribution is unique.
>
> **Q2:** In the task-representation space, our model reduces to a weighted multiclass softmax regression. For each sub-task, GD is governed by a single logit-gap dynamic, analogous to classical logistic/softmax regression on separable data, yielding the same rate.
>
> **Q4:** Matrix orthogonalization in conventional Muon(applying SVD, keeping singular directions with nonzero singular values with those values set to 1) is exactly matrix sign(applying sign to the singular value matrix).
>
> **Q6:** The disparity is the maximal probability gap $\Delta _t$ in Sec.7.1 ($\Delta_t:=\max _{j\in[K]}\widehat{p} _{j|j}(W _t)-\min _{j\in[K]}\widehat{p} _{j|j}(W _t)$).
>
> **Q7:** We trained a 100M LLaMA on 10B tokens with constant LR, batch size 512, and sequence length 2048 for SGD and Muon. We searched LRs over {0.000125, 0.00025, 0.0005, 0.001, 0.002, 0.004, 0.008} and plotted final loss at the best learning rate for each data budget. We will add these details.
>
> **Reference**
>
> [1]Scaling Laws for Gradient Descent and Sign Descent for Linear Bigram Models under Zipf's Law.
>
> [2]Surge Phenomenon in Optimal Learning Rate and Batch Size Scaling.
>
> [3]Scaling Laws of SignSGD in Linear Regression: When Does It Outperform SGD?
>
> [4]Adaptive Methods through the Lens of SDEs: Theoretical Insights on the Role of Noise.

---

> > ### Author Rebuttal · Reviewer_NEke · 2026-04-03
> >
> > Thanks the authors for their response. It addressed my concerns and questions. I would like to maintain my positive score.

---

> > > ### Author Response · Authors · 2026-04-05
> > >
> > > We sincerely thank the reviewer for the positive feedback and the thoughtful concerns. We are grateful for the recognition of our work's contribution and appreciate the time invested in reviewing our manuscript.

---

### Decision · Program_Chairs · 2026-04-30

**Decision:**

Accept (regular)

**Comment:**

This paper provides a theoretical analysis of the training dynamics of the Muon (or, spectral GD) optimizer within a linear associative-memory framework with frequency-imbalanced data. The authors show that, unlike GD, which learns different frequency components at imbalanced rates and is bottlenecked by rare items, Muon achieves more balanced learning and faster convergence; these insights are supported by analyses in both noiseless and noisy settings, scaling-law results, and empirical validation.

All four reviewers have largely reached a consensus on the value of the paper (despite one weak reject score), and I believe that the main concerns have been satisfactorily addressed by the authors during the rebuttal. I therefore recommend acceptance.